# GLOWQ: GROUP-SHARED LOW-RANK APPROXIMATION FOR QUANTIZED LLMS

**Selim An**
Department of Artificial Intelligence
DGIST, Korea
phantom06@dgist.ac.kr

**Ilhong Suh**
COGA robotics, Korea
ihsuh@coga-robotics.com

**Yeseong Kim**
Department of Electrical Engineering
POSTECH, Korea
yeseongkim@postech.ac.kr

## ABSTRACT

Quantization techniques such as BitsAndBytes (Dettmers et al., 2022), AWQ (Lin et al., 2024), and GPTQ (Frantar et al., 2022) are widely used as a standard method in deploying large language models but often degrades accuracy when using low-bit representations, e.g., 4 bits. Low-rank correction methods (e.g., LQER (Zhang et al., 2024a), QERA (Zhang et al., 2024b), ASER (Zhao et al., 2025)) has been proposed to mitigate this issue, however, they restore all layers and insert error-correction modules into every decoder block, which increases latency and memory overhead. To address this limitation, we propose GlowQ, a group-shared low-rank approximation for quantized LLMs that caches a single shared right factor per input-sharing group and restores only the groups or layers that yield the highest accuracy benefit. GlowQ computes the high-precision projection once per input-sharing group and reuses it across its modules, reducing parameter and memory overhead, and retaining the expressivity of layer-specific corrections. We also propose a selective variant, GlowQ-S, that applies the cached shared module only where it provides the largest benefit. Compared with strong baselines, our approach reduces TTFB by $5.6\%$ and increases throughput by $9.6\%$ on average, while reducing perplexity on WikiText-2 by $0.17\%$ and increasing downstream accuracy by 0.42 percentage points. The selective model GlowQ-S further reduces latency, cutting TTFB by $23.4\%$ and increasing throughput by $37.4\%$, while maintaining accuracy within 0.2 percentage points on average. Code is available at https://github.com/ahnselim/GlowQ.

## 1 INTRODUCTION

As large language models (LLMs) grow in width and depth, the cost of serving and adapting them becomes a primary bottleneck for real-world use. Compression by post-training quantization (PTQ) alleviates memory and bandwidth pressure without altering the model architecture, and has matured through methods such as GPTQ Frantar et al. (2022), AWQ Lin et al. (2024),BITSANDBYTES Dettmers et al. (2022). A complementary thread augments quantized weights with a small high-precision, low-rank term so that $W \approx W_q + AB$ and the inference output is corrected by adding $A(BX)$ Zhang et al. (2024a;c); Zhao et al. (2025); Zhang et al. (2024b). These lines enable competitive quality at quantized weights across modern transformer stacks.

Most low-rank compensation pipelines attach an independent $(A, B)$ module to each layer or projection and evaluate the high-precision projection $BX$ repeatedly along the network. This design (i) duplicates the same expensive computation for modules that ingest the same input tensor, (ii) increases memory traffic by materializing multiple $BX$ values, and (iii) selects subspaces with objectives that often ignore the strong anisotropy of real activations Ethayarajh (2019); Godey et al. (2024), misallocating limited rank to rarely used directions. As a result, the accuracy-efficiency trade-off is weaker than necessary, especially under strict latency budgets.

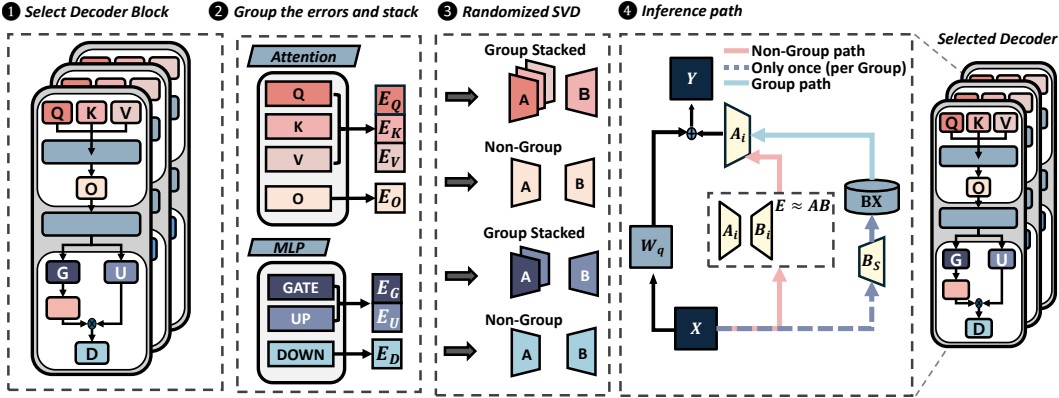

Figure 1: GlowQ Overview

We propose GlowQ, Group-Shared Low-Rank Approximation for Quantized LLMs. As illustrated in Fig. 1, GlowQ treats modules that share the same input as a group Vaswani et al. (2017), learns a single shared right factor $B_{\text{shared}}$ for that group, and keeps module-specific left factors $\{A_i\}$. At inference, it computes $R := B_{\text{shared}}X$ once per group and reuses it via $A_iR$, turning many large $BX$ multiplications into several cheap matrix-vector updates. To align limited rank with how inputs are actually used, we adopt a covariance-aligned objective that emphasizes frequently visited directions. Finally, a Selective Restore policy enables only high-payoff groups or layers under a deployment budget. Molchanov et al. (2019)

When modules share the input dimension, the joint least-squares problem with a single right factor is equivalent to approximating the vertical stack of module-wise error matrices; its minimizers are characterized by the right singular structure of the stacked matrix ("stacked SVD"). We then connect a usage-weighted risk $\min_{\mathbf{A},\mathbf{B}} \|\mathbf{E}_{\text{cat}} - \mathbf{A}\mathbf{B}\|_F^2$ to a right-weighted Frobenius norm, which yields a covariance-aligned objective whose global solution is governed by the SVD of the whitened error, where the whitened errors are those rescaled by the input covariance. This provides both a rationale for a shared $\mathbf{B}$ and a principled way to steer it toward data-preferred axes.

To avoid forming tall whitened matrices, we introduce a QR-reduced randomized SVD routine: a thin QR compresses the stacked error into a $d \times d$ core; randomized SVD with oversampling and power iterations extracts the dominant right subspace; balanced recovery returns $(A^\star, B^\star)$ with improved numerical stability. The solver drops into our grouping and caching runtime with no extra architectural changes.

- **Group-level shared-$B$.** We formalize input-sharing groups and show that one shared right factor per group suffices for the joint least-squares objective, enabling one-shot $BX$ and multi-module reuse (Sec. 3).

- **Data-aware alignment.** We derive a covariance-aligned objective by bridging usage-weighted risk and a right-weighted Frobenius criterion; its global minimizer aligns the shared right subspace with data-preferred directions (Sec. 3.1).

- **QR-reduced RSVD.** We present a practical pipeline that performs QR reduction to a small core and applies randomized SVD with balanced factor recovery, avoiding tall whitened matrices while preserving accuracy (Sec. 3.2).

- **Caching & Selective Restore.** We implement a deployment path that caches $R = B_{\text{shared}}X$ once per group and activates only important groups/layers, translating algorithmic savings into latency/throughput gains (Sec. 3.3).

- **Empirical gains over strong baselines.** Across the evaluated model families and benchmarks, GlowQ consistently improves both efficiency and accuracy: it reduces time-to-first-byte (TTFB) by $5.6\%$ and increases throughput by $9.6\%$ on average, while reducing WikiText-2 perplexity by $0.17\%$ and increasing downstream accuracy by 0.42 percentage points. The selective variant, GlowQ-S, further lowers latency, cutting TTFB by $23.4\%$ and increasing throughput by $37.4\%$, while maintaining accuracy within 0.2 percentage points of full GlowQ.

## 2 RELATED WORK

**Post-training quantization (PTQ).** Today's PTQ methods span a variety of designs that recover accuracy at the quantization stage without changing the model structure or runtime path. GPTQ Frantar et al. (2022) uses second-order information to directly fit quantized weights and preserve layer outputs; AWQ Lin et al. (2024) protects important channels based on activation statistics via rescaling. In the BITSANDBYTES family, LLM.INT8() Dettmers et al. (2022) uses vector-wise quantization with an outlier-aware mixed-precision path where most channels run in INT8. These methods constitute standard baselines for LLM lightweighting.

**Quantization error correction via low-rank compensation.** Prior work shows that post-quantization errors can be effectively reduced by adding a low-rank term to the quantized weights or outputs. LQER approximates the per-layer quantization error as $E \approx AB$ and adds a high-precision correction without changing the inference graph (Zhang et al., 2024a). ZEROQUANT-V2 systematizes low-rank compensation (LoRC) within PTQ pipelines and demonstrates that a small-rank correction can recover accuracy at low bit-widths (Yao et al., 2023). QERA derives a closed-form, output-error-centric formulation that clarifies when low-rank correction benefits PTQ/PEFT (Zhang et al., 2024b). ASER combines a whitened-SVD-style low-rank corrector with activation smoothing to stabilize low-bit regimes (Zhao et al., 2025). While these works justify the $AB$ correction principle, most deploy independent $(A_\ell, B_\ell)$ at every layer and recompute the high-precision product $A_\ell(B_\ell X)$ for all layers and tokens, which increases latency and memory traffic; moreover, attaching a low-rank module to every layer inflates GPU memory usage.

**Stacked/collective SVD for a shared right subspace.** The idea of factorizing multiple matrices with a shared latent factor is established in collective/joint matrix factorization: when several matrices share the same input dimension, one can vertically concatenate their blocks and fit a single right subspace while allowing matrix-specific left factors (Singh & Gordon, 2008). Recent analyses also study the optimal recovery of shared singular subspaces across matrices (Ma & Ma, 2024). We adopt this principle for input-sharing modules in LLMs : we stack group-wise error blocks into $E_{\text{cat}}$ and learn one $B_{\text{shared}}$ per group. At inference, we compute the right projection once per group, $R = B_{\text{shared}}X$, cache it, and let each module apply only the lightweight left multiplication $A_iR$. This reduces high-precision matmuls and the number of resident correction parameters compared to layer-wise independent $AB$.

**Covariance-aligned selective restoration.** Because inputs are anisotropic, a plain stacked objective may learn right subspaces misaligned with data-preferred directions. We therefore adopt a covariance-aligned (whitened) formulation, measuring residual error in the input-covariance metric so that the shared subspace is guided toward meaningful axes (Golub & Van Loan, 2013; Srebro & Jaakkola, 2003). Not all layers require restoration; following pruning-inspired saliency, we activate only the most beneficial groups under a budget, using (i) an SVD energy-capture score ($\|A\|_F^2$ per group), (ii) a normalized error ratio $\|E_g\|_F / \|W_g\|_F$ (Nagel et al., 2020; Banner et al., 2019), and (iii) a layer-order fallback when signals are weak. Coupled with the shared-right-subspace design and cached $R = B_{\text{shared}}X$, this selective restore achieves stronger accuracy-latency-memory trade-offs than per-layer low-rank baselines at the same cost.

## 3 METHOD: GLOWQ

In this section, we introduce our method, Group-Shared Low-Rank Approximation For Quantized LLMs (GlowQ). Prior low-rank restoration often (i) restores all layers and (ii) multiplies a per-layer low-rank module with activations, causing heavy overhead. We address both by (a) learning a **shared** right subspace for modules that share the same input and (b) **caching** the input projection once per group for reuse. We approximate each error matrix and its vertical concatenation by a rank-$r$ factorization: $E_i \approx \mathbf{A}_i\mathbf{B}$ and $E_{\text{cat}} \approx \mathbf{AB}$, where $\mathbf{A} = [\mathbf{A}_1; \ldots; \mathbf{A}_m]$ and $\mathbf{B}$ is shared within a group. At inference, the correction for each module $i$ takes the form $\mathbf{A}_i(\mathbf{BX})$, where the projection $\mathbf{BX}$ is computed once for the entire group.

### 3.1 GROUPING QUANTIZATION-ERROR CORRECTION MODULES

We aim to find the optimal shared low-rank correction module, in particular a shared right factor $\mathbf{B}$. To this end, we first formalize the problem via an unweighted baseline (Sec. 3.1.1), and propose a

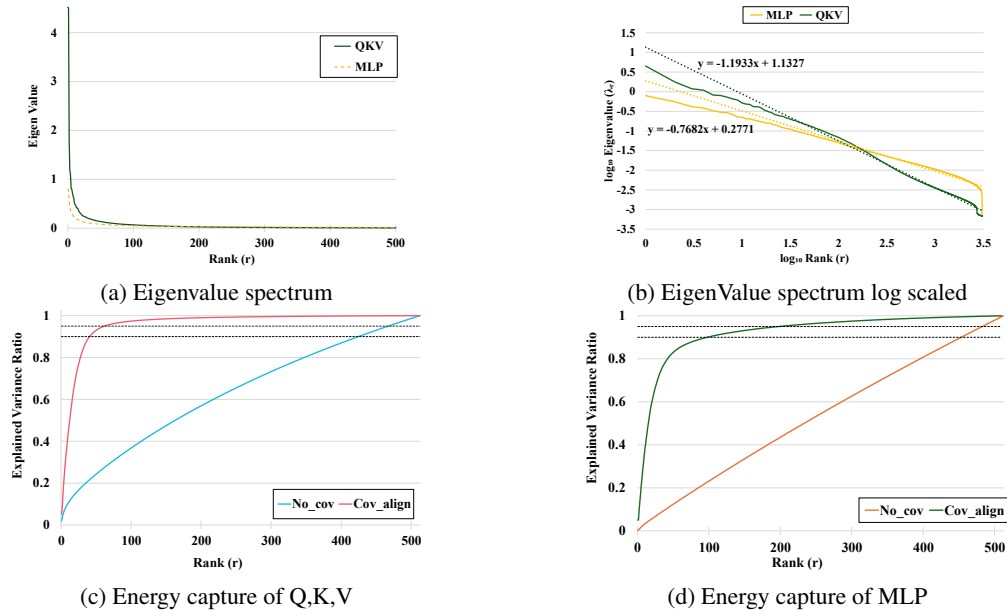

(a) Eigenvalue spectrum

(b) EigenValue spectrum log scaled

(c) Energy capture of Q,K,V

(d) Energy capture of MLP

Figure 2: Input spectrum and energy-capture measurements. (a) We stream calibration samples through the model, collect the input activations at the target layer, and plot the eigenvalue spectrum of the empirical input covariance for the QKV and MLP groups, revealing a heavy-tailed profile. (b) The same spectra plotted in $\log_{10} \lambda_r$–$\log_{10} r$ coordinates; dotted lines show least-squares fits over the approximately linear tail region, indicating power-law decay $\lambda_r \propto r^{-\alpha}$ with exponents $\alpha_{\mathrm{MLP}} \approx 0.77$ and $\alpha_{\mathrm{QKV}} \approx 1.19$. (c-d) For each group, we vertically stack the quantization-error matrices and plot the cumulative fraction of Frobenius energy recovered by the best rank-$r$ approximation. We show both the unweighted baseline (No cov) and the covariance-aligned variant that weights errors by the observed inputs (Cov align). Horizontal dashed lines mark 90% and 95% energy capture.

data-aware objective that incorporates covariance alignment to overcome the limitation induced by input anisotropy (Sec. 3.1.2).

### 3.1.1 UNWEIGHTED BASELINE: STACKED SVD

Let modules $i = 1, \ldots, m$ share the same input dimension $d$. For error matrices $E_i \in \mathbb{R}^{O_i \times d}$, define the vertical concatenation

$$\mathbf{E_{cat}} := \begin{bmatrix} E_1^{\mathsf{T}} & \cdots & E_m^{\mathsf{T}} \end{bmatrix} \in \mathbb{R}^{d \times \left(\sum_i O_i\right)}, \qquad \mathbf{A} := \begin{bmatrix} \mathbf{A}_1^{\mathsf{T}} & \cdots & \mathbf{A}_m^{\mathsf{T}} \end{bmatrix} \in \mathbb{R}^{r \times \left(\sum_i O_i\right)}. \quad (1)$$

We seek a shared right factor $\mathbf{B} \in \mathbb{R}^{r \times d}$ and blocks $\mathbf{A}_i \in \mathbb{R}^{O_i \times r}$ that minimize

$$\min_{\mathbf{A}, \mathbf{B}} \|\mathbf{E_{cat}} - \mathbf{A}\mathbf{B}\|_F^2. \quad (2)$$

**Proposition 1 (Shared-$\mathbf{B}$ is optimal).** For modules that share the same input, jointly fitting with a single right factor $\mathbf{B}$ is equivalent to one low-rank fit of the stacked matrix $E_{\mathrm{cat}}$. By Eckart-Young-Mirsky, an optimal $\mathbf{B}$ spans the top-$r$ right-singular subspace of $E_{\mathrm{cat}}$; allowing per-module $\mathbf{B}_i$ adds no extra expressivity because any differences can be absorbed into invertible reparameterizations of $\mathbf{A}_i$. Hence, a single shared $\mathbf{B}$ is sufficient and optimal for the group. (Proof and identifiability details are deferred to Appendix A.1.)

Real inputs are anisotropic, which can be diagnosed by the eigenvalue spectrum of the covariance $\Sigma_x$. Fig. 2a exhibits a heavy-tailed profile, with an abrupt initial drop followed by a long tail, indicating that the usage of the representation space is strongly concentrated in a small number of axes. Under such a distribution, the relative importance between frequently used directions and the remaining ones diverges markedly. To quantify this behavior, Fig. 2b plots the eigenvalue spectra of the empirical input covariance for the MLP and QKV groups in $\log_{10} \lambda_r$–$\log_{10} r$ scale: for each group we sort the eigenvalues $\{\lambda_r\}$ in descending order and plot $\log_{10} \lambda_r$ versus $\log_{10} r$. The dotted

lines show least-squares linear fits over the approximately linear tail region, revealing power-law decay $\lambda_r \propto r^{-\alpha}$ with exponents $\alpha_{\text{MLP}} \approx 0.77$ and $\alpha_{\text{QKV}} \approx 1.19$, which quantitatively confirms the heavy-tailed, anisotropic input statistics that motivate our covariance-aligned objective.

However, the unweighted cluster SVD selects the shared right subspace purely from the geometry (variance structure) of the error matrices. This can misalign the selected subspace with the axes preferred by the data; at a fixed rank, such a misalignment reduces energy capture and weakens consistency within the group. Alignment arises naturally only in restrictive cases, such as isotropic input or near-simultaneous diagonalization.

Therefore, to treat anisotropy fairly, we should evaluate errors in a coordinate system where all directions carry equal usage. In such a space, frequently used directions are not under-weighted, and rarely used directions are not over-weighted, so the learned shared right subspace aligns better with the data-preferred axes.

### 3.1.2 DATA-AWARE COVARIANCE ALIGNMENT

The evidence in Fig. 2a shows strong input anisotropy; hence reconstruction should account not only for the geometry of error matrices but also for how inputs are actually used. For any factors $(\mathbf{A}, \mathbf{B})$ and residual $\mathbf{M} := \mathbf{E}_{\text{cat}} - \mathbf{A}\mathbf{B}$, the expected loss under the usage distribution is

$$\mathbb{E} \|\mathbf{M}\,\mathbf{x}\|_2^2 = \operatorname{tr}\!\big(\mathbf{M}\,\boldsymbol{\Sigma}_{\mathbf{x}}\,\mathbf{M}^\top\big) = \|\mathbf{M}\,\boldsymbol{\Sigma}_{\mathbf{x}}^{1/2}\|_F^2. \tag{3}$$

which follows from the standard quadratic-form identity together with the Frobenius-trace identity (Petersen & Pedersen, 2006). To balance direction-wise usage, we whiten by $\boldsymbol{\Sigma}_{\mathbf{x}}^{1/2}$ so that the selected shared right subspace is steered toward axes preferred by the data.

Using the definitions from Sec. 3.1.1, we adopt the right-weighted objective

$$\min_{\mathbf{A},\mathbf{B}} \big\| (\mathbf{E}_{\text{cat}} - \mathbf{A}\mathbf{B})\,\boldsymbol{\Sigma}_{\mathbf{x}}^{1/2}\big\|_F^2 \;\equiv\; \min_{\mathbf{A},\mathbf{B}} \big\| \tilde{\mathbf{E}} - \mathbf{A}\mathbf{B}\big\|_F^2, \qquad \tilde{\mathbf{E}} := \mathbf{E}_{\text{cat}}\boldsymbol{\Sigma}_{\mathbf{x}}^{1/2}. \tag{4}$$

In the isotropic case ($\boldsymbol{\Sigma}_{\mathbf{x}} \propto I$), Eq. 4 reduces to the unweighted baseline in Sec. 3.1.1. Empirically, Fig. 2c and 2d shows that, at a fixed rank, whitening yields substantially faster growth of the cumulative energy capture compared to the unweighted variant, indicating better alignment of the learned shared right subspace with data-preferred directions.

**Proposition 2 (Usage-weighted risk equals a right-weighted reconstruction error).** When inputs are centered and have covariance $\boldsymbol{\Sigma}_{\mathbf{x}}$, the model's expected loss equals the residual energy averaged over draws from the input distribution. Equivalently, it is the residual measured after weighting columns according to how frequently and how strongly each input direction is used (as determined by $\boldsymbol{\Sigma}_{\mathbf{x}}$). Therefore, minimizing the usage-weighted risk is exactly the same optimization as minimizing the right-weighted reconstruction error in Eq. 4. A full derivation and the nonzero-mean case are deferred to Appendix A.2.

**Proposition 3 (Covariance-aligned minimizer)** The global minimizers $(\mathbf{A}^\star, \mathbf{B}^\star)$ of Eq. 4 are given by the rank-$r$ SVD of the whitened error matrix $\tilde{\mathbf{E}} = \mathbf{E}_{\text{cat}}\boldsymbol{\Sigma}_{\mathbf{x}}^{1/2}$. In particular, the optimal shared right subspace, $\operatorname{row}(\mathbf{B}^\star)$, is spanned by the top-$r$ right singular vectors of $\tilde{\mathbf{E}}$; this is the standard Eckart-Young-Mirsky solution specialized to the whitened problem (Eckart & Young, 1936) Golub & Van Loan, 2013.

### 3.2 SCALABLE IMPLEMENTATION VIA QR-REDUCED RANDOMIZED SVD

We present an implementation that solves the covariance-aligned objective

$$\min_{\mathbf{A},\mathbf{B}} \big\|(\mathbf{E}_{\text{cat}} - \mathbf{A}\mathbf{B})\,\boldsymbol{\Sigma}_{\mathbf{x}}^{1/2}\big\|_F^2 \tag{5}$$

without forming the tall whitened matrix. The method follows three steps: (i) **QR reduction** to compress the tall matrix into a $d \times d$ core, (ii) **Randomized SVD (RSVD)** on the core to capture the top-$r$ right subspace, and (iii) **balanced recovery** to obtain $(\mathbf{A}^\star, \mathbf{B}^\star)$. This yields practical advantages such as avoiding materialization of huge matrices, lower compute/memory cost, improved numerical stability via balanced factors, and direct compatibility with the caching/Selective-Restore pipeline in Sec. 3.3.

### 3.2.1 ALGORITHM & COMPLEXITY

Using a thin QR of $\mathbf{E}_{\text{cat}}$, we reduce the covariance-aligned objective to a $d \times d$ core (Alg. 1); a full SVD on the core costs $\mathcal{O}(d^3)$, whereas randomized sketching recovers the leading right subspace in $\mathcal{O}\big(d^2(r+p) + q\,d^2(r+p)\big)$ time. Here, $p$ denotes oversampling (extra sketch columns) and $q$ denotes the number of power iterations used to sharpen the subspace.

---

**Algorithm 1** Covariance-aligned QR reduction and randomized SVD on the core

---

**Require:** Stacked error $\mathbf{E}_{\text{cat}} \in \mathbb{R}^{m \times d}$, covariance $\mathbf{\Sigma}_{\mathbf{x}} \succeq 0$, target rank $r$, oversampling $p$, power iters $q$
**Ensure:** Low-rank factors $(\mathbf{A}^\star, \mathbf{B}^\star)$ for the covariance-aligned objective
 1: **Thin QR of $\mathbf{E}_{\text{cat}}$**: compute $\mathbf{Q}_e \mathbf{R}_e = \mathbf{E}_{\text{cat}}$ with $\mathbf{Q}_e^\top \mathbf{Q}_e = \mathbf{I}_d$
 2: **Core construction**: set $\mathbf{M} \leftarrow \mathbf{R}_e \mathbf{\Sigma}_{\mathbf{x}}^{1/2} \in \mathbb{R}^{d \times d}$
 3: **Random sketch / range finding**: draw $\mathbf{\Omega} \sim \mathcal{N}(0,1)^{d \times (r+p)}$, set $\mathbf{Y} \leftarrow \mathbf{M}\mathbf{\Omega}$; optionally do $q$ power steps $\mathbf{Y} \leftarrow \mathbf{M}(\mathbf{M}^\top \mathbf{Y})$
 4: **Orthonormalize**: $\mathbf{Q} \leftarrow \text{orth}(\mathbf{Y}) \in \mathbb{R}^{d \times (r+p)}$
 5: **Compressed SVD**: $\mathbf{B}_{\text{small}} \leftarrow \mathbf{Q}^\top \mathbf{M}$; compute $\mathbf{B}_{\text{small}} = \tilde{\mathbf{U}} \mathbf{\Sigma} \mathbf{V}^\top$
 6: **Lift left factor**: $\mathbf{U} \leftarrow \mathbf{Q}\tilde{\mathbf{U}}$
 7: **Truncate (top-$r$) & balance**: keep $(\mathbf{U}_r, \mathbf{\Sigma}_r, \mathbf{V}_r)$ and set $\widehat{\mathbf{A}}^\star \leftarrow \mathbf{U}_r \mathbf{\Sigma}_r^{1/2}$, $\widehat{\mathbf{B}}^\star \leftarrow \mathbf{\Sigma}_r^{1/2} \mathbf{V}_r^\top$
 8: **Lift to original variables**: $\mathbf{A}^\star \leftarrow \mathbf{Q}_e \widehat{\mathbf{A}}^\star$, $\mathbf{B}^\star \leftarrow \widehat{\mathbf{B}}^\star \mathbf{\Sigma}_{\mathbf{x}}^{-1/2}$ ▷ use a pseudoinverse if $\mathbf{\Sigma}_{\mathbf{x}}$ is singular

---

By left-orthogonal invariance of the Frobenius norm, the QR reduction collapses the tall-$m$ problem to a $d \times d$ core without loss for the covariance-aligned objective (formal proof in Appendix A.3; (Golub & Van Loan, 2013)). Randomized sketching on the core provides an efficient and accurate estimate of the leading right subspace with controllable bias via $(p, q)$; we summarize theoretical guarantees in Appendix A.4 and present empirical runtime-accuracy trade-offs (vs. exact SVD) in Sec. D.3.1 (Halko et al., 2011). Balanced recovery yields

$$\widehat{\mathbf{A}}^\star = \mathbf{U}_r \mathbf{\Sigma}_r^{1/2}, \quad \widehat{\mathbf{B}}^\star = \mathbf{\Sigma}_r^{1/2} \mathbf{V}_r^\top, \big\| (\mathbf{E}_{\text{cat}} - \mathbf{A}^\star \mathbf{B}^\star) \mathbf{\Sigma}_{\mathbf{x}}^{1/2} \big\|_F = \big\| \mathbf{M} - \mathbf{U}_r \mathbf{\Sigma}_r \mathbf{V}_r^\top \big\|_F \quad (6)$$

and the resulting $\mathbf{B}^\star$ serves as the shared right factor used in Sec. 3.3 for once-per-group caching ($R = \mathbf{B}_{\text{shared}}\mathbf{X}$) and Selective Restore.

### 3.3 CACHING AND SELECTIVE RESTORE

The group-shared factorization implies that modules within the same input-sharing group all rely on the *same* right-side projection $\mathbf{X}\,\mathbf{B}_{\ell,\text{shared}}^\top$. Naively evaluating this projection for every module recreates the primary inefficiency of layer-wise correction, i.e., multiple high-precision matrix-vector multiplications along the critical path. To translate the theoretical shared structure into practical inference gains, GlowQ introduces a caching mechanism that computes the right-sided projection *once per group*, and a complementary selective-restore policy that activates correction only at groups offering the largest accuracy benefit under a deployment budget.

For each layer group $G_\ell$ that shares the same input dimension, we compute a single intermediate

$$R_\ell := \mathbf{X}\,\mathbf{B}_{\ell,\text{shared}}^\top \in \mathbb{R}^{B \times T \times r} \quad (7)$$

once per group and reuse it across all modules in the group. Each module $i \in G_\ell$ then applies only the small correction

$$y_i = \mathbf{W}_i^{(q)} \mathbf{X} + \mathbf{A}_{\ell,i} R_\ell, \quad (8)$$

where $\mathbf{A}_{\ell,i} \in \mathbb{R}^{O_i \times r}$ and $\mathbf{B}_{\ell,\text{shared}} \in \mathbb{R}^{r \times I}$. We adopt an anchor policy to materialize $R_\ell$ exactly once and consume it a fixed number of times: in attention, $q$ is the anchor and $(k, v)$ are consumers; in MLP, gate is the anchor and up is the consumer. Solo modules that do not share inputs (e.g., o_proj, down_proj) compute $R_i := \mathbf{X}\,\mathbf{B}_i^\top$ on the fly without reuse.

Given a latency or memory budget, we rank all candidate units (groups or solo layers) by an importance score and activate only the top $k$. Importance is measured using two metrics: a GSVD-based energy-capture score (Eq. 9) after covariance alignment (Paige & Saunders, 1981; Jolliffe &

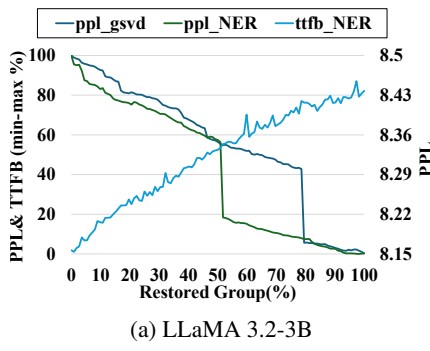 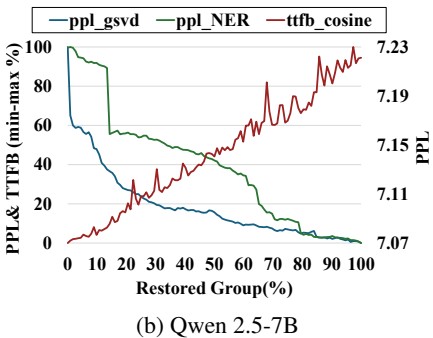

(a) LLaMA 3.2-3B                    (b) Qwen 2.5-7B

Figure 3: Perplexity (PPL) and time-to-first-byte (TTFB) versus the fraction of restored groups.

Cadima, 2016; Halko et al., 2011), and a normalized error ratio (Eq. 10) (Malinovskii et al., 2025; Dong et al., 2019). At runtime, we apply the cached low-rank correction only to the selected units, skipping inactive ones; because the cache is materialized only for active groups, selective restore naturally complements group-shared caching.

$$g_{\text{ec}}(u) = \frac{\sum_{j=1}^{r} \sigma_j(\mathbf{M}_u)^2}{\|\mathbf{M}_u\|_F^2} \qquad (9) \qquad\qquad g_{\text{ner}}(u) = \frac{\|\mathbf{E}_u\|_F^2}{\|\mathbf{W}_u\|_F^2} \qquad (10)$$

# 4 EXPERIMENTS

## 4.1 EXPERIMENTAL SETUP

We evaluate LLaMA 3 (3.2–3B, 3.1–8B) Dubey et al. (2024), LLaMA 2 (7B, 13B) Touvron et al. (2023), Qwen 2.5 (7B, 14B) Qwen et al. (2025), Qwen 3 (8B, 14B) Yang et al. (2025), OPT (1.3B, 6.7B) Zhang et al. (2022), Mistral 7B Jiang et al. (2023), and Qwen1.5-MoE-A2.7B Bai et al. (2023a) with Vicuna reported only in ablations.

All models use W4A16 (int4 weights, fp16 activations) with group size 128; the rank is fixed at 64. Calibration uses 64 sequences of length 2048 shared across methods, with no fine-tuning unless a baseline requires it. We compare GlowQ and GlowQ-S with various state-of-the-art baselines, including PTQ (BitsAndBytes Dettmers et al. (2022), AWQ Lin et al. (2024), GPTQ Frantar et al. (2022)) and error-correction methods in literature (L2QER Zhang et al. (2024a), ZeroQuant-V2 Yao et al. (2023), QERA Zhang et al. (2024b)). All under the same protocol and recommended defaults.

We report perplexity on WikiText-2 Merity et al. (2016) and C4 Raffel et al. (2020), and zero-shot accuracy on ARC-E/ARC-C Clark et al. (2018), PIQA Bisk et al. (2020), HellaSwag Zellers et al. (2019), WinoGrande Sakaguchi et al. (2021), BoolQ Clark et al. (2019), and LAMBADA Paperno et al. (2016) via `lm-eval-harness` (defaults). We run the proposed method on A100 GPUs for covariance/SVD steps while inference is executed on an RTX 4090.

**GlowQ-S Configuration** GlowQ-S applies the cached correction only to a subset of groups, selected according to an importance score. Since different model families exhibit distinct restoration profiles, we adopt a model-specific scoring rule for GlowQ-S. We defer the full characterization of these curves and the selection policy to Section 4.6.

## 4.2 MAIN RESULTS: PERPLEXITY AND ZERO-SHOT ACCURACY

**Perplexity.** Table 1 reports test perplexity (lower is better) for WikiText-2 under a common protocol: W4A16 with int4 weight groups of 128 and a shared calibration set of 64 sequences at length 2048 for all methods. Overall, GlowQ achieves the best or tied-best perplexity on 9 of 11 model variants, including consistent gains on LLaMA 3 (3.2-3B/3.1-8B), Qwen 3 (8B/14B), Qwen 2.5-7B, and Mistral-7B. On Qwen 2.5-14B, GlowQ matches the strongest baselines. Exceptions occur on LLaMA 2-13B (where L2QER slightly leads) and OPT-1.3B (where QERA leads), while OPT 6.7B favors a pure PTQ path. These outcomes indicate that group-shared low-rank correction closes much of the int4 gap to FP16 across diverse architectures without task-specific tuning. Beyond the W4A16 setting, the lower block of Table 1 evaluates mixed-precision weight-activation quantization with W4A4 and W4A8. As expected, W4A4 increases perplexity for all methods, but GlowQ (and

Table 1: WikiText-2 test perplexity (lower is better). GlowQ-S restores 51% of layers for LLaMA 3.2-3B, while all other models use 50% restoration.

| Method | Q config | LLaMA 2 | | LLaMA 3 | | Qwen 2.5 | | Qwen 3 | | Mistral | OPT | |
| | | 7B | 13B | 3.2-3B | 3.1-8B | 7B | 14B | 8B | 14B | 7B | 1.3B | 6.7B |
|---|---|---|---|---|---|---|---|---|---|---|---|---|
| FP16 | - | 5.48 | 4.90 | 7.81 | 6.24 | 6.86 | 5.29 | 9.73 | 8.64 | 5.32 | 14.62 | 10.85 |
| BnB | NF4 | 5.64 | 4.97 | 8.29 | 6.66 | 7.10 | **5.64** | 9.97 | 8.88 | 5.51 | 15.16 | **10.94** |
| AWQ | INT4, g128 | 5.61 | 4.97 | 8.24 | 6.64 | 7.11 | 6.17 | 10.19 | 9.00 | 5.51 | 15.22 | 11.23 |
| GPTQ | INT4, g128 | 5.65 | 5.35 | 9.46 | 6.63 | 7.11 | 5.75 | 9.98 | 8.90 | 5.51 | 15.00 | 11.07 |
| ZeroQuant-V2 | INT4, g128 | 5.72 | 4.99 | 8.44 | 6.79 | 8.41 | 5.75 | 10.19 | 9.04 | 5.53 | 15.10 | 11.14 |
| QERA | INT4, g128 | 5.61 | 4.98 | 8.22 | 6.64 | 8.09 | **5.64** | 10.07 | 8.85 | 5.48 | 14.85 | 11.00 |
| L2QER | INT4, g128 | 5.68 | **4.94** | 8.30 | 6.75 | 8.14 | 5.66 | 10.07 | 8.85 | 5.46 | 15.30 | 11.16 |
| GlowQ | INT4, g128 | **5.58** | 4.96 | **8.16** | **6.59** | 7.07 | **5.64** | **9.90** | **8.80** | **5.42** | **14.84** | 11.00 |
| GlowQ-S | INT4, g128 | 5.60 | 4.96 | 8.22 | 6.62 | 7.09 | 5.68 | 9.97 | 8.89 | 5.45 | 15.00 | 11.00 |
| L2QER | W4A4 | **5.90** | 5.18 | 9.42 | 7.65 | 9.11 | **6.52** | 10.76 | 9.36 | **5.73** | 27.40 | 11.32 |
| L2QER | W4A8 | 5.69 | 4.95 | 8.31 | 6.76 | 8.15 | **5.67** | 10.11 | 8.86 | 5.47 | 14.90 | 11.00 |
| GlowQ | W4A4 | **5.90** | **5.20** | **9.21** | **7.42** | 8.03 | 6.55 | **10.66** | **9.33** | 5.74 | **26.35** | **11.31** |
| GlowQ-S | W4A4 | 5.92 | 5.20 | 9.25 | 7.45 | 8.05 | 6.61 | 10.72 | 9.37 | 5.79 | 27.42 | 11.33 |
| GlowQ | W4A8 | **5.59** | **4.97** | **8.20** | **6.63** | **7.12** | 5.71 | **10.08** | **8.85** | **5.43** | 14.85 | **10.97** |
| GlowQ-S | W4A8 | 5.60 | 4.97 | 8.24 | 6.64 | 7.13 | 5.77 | 10.10 | 8.92 | 5.48 | 14.99 | 10.99 |

GlowQ-S) remain competitive with or better than L2QER on most models, and the W4A8 configuration nearly recovers the W4A16 accuracy, indicating that our covariance-aware low-rank correction continues to be effective even under joint weight-activation quantization.

Table 2: Average accuracy (↑) on seven downstream tasks and C4 perplexity (↓).

| Method | Rank | LLaMA 3.2-3B | | LLaMA 3.1-8B | | Qwen 3-8B | | Qwen 3-14B | |
| | | Acc (↑) | C4 (↓) | Acc (↑) | C4 (↓) | Acc (↑) | C4 (↓) | Acc (↑) | C4 (↓) |
|---|---|---|---|---|---|---|---|---|---|
| FP16 | - | 67.14 | 10.30 | 73.29 | 9.00 | 71.48 | 14.52 | 74.10 | 13.08 |
| ZeroQuant-V2 | | 65.38 | 11.45 | **73.48** | 9.87 | 70.19 | 15.00 | 72.62 | 13.79 |
| QERA | | 65.48 | 11.04 | 72.86 | 9.68 | 69.86 | 14.78 | 73.14 | 13.29 |
| L2QER | 64 | 66.19 | 11.04 | 72.43 | 9.63 | 69.52 | 14.82 | 73.24 | 13.80 |
| GlowQ | | **66.90** | **10.98** | 73.33 | **9.59** | **70.71** | **14.60** | **73.84** | **13.26** |
| GlowQ-S | | 66.33 | 11.07 | 72.62 | 9.78 | 70.29 | 14.77 | 73.24 | 13.48 |

**Overall quality.** Table 2 reports the zero-shot accuracy via *lm-eval-harness* along with the perplexity for the C4 dataset. Across four representative models (LLaMA 3 3.2-3B / 3.1-8B, Qwen 3 8B / 14B), GlowQ attains the lowest C4 perplexity among quantized/error-corrected methods and delivers the strongest average zero-shot accuracy on LLaMA 3.2-3B and Qwen 3-8B/14B (ZeroQuant-V2 leads on LLaMA 3.1-8B). GlowQ improves over the best non-GlowQ baseline in the zero-shot accuracy by average +0.3%; in C4 perplexity, GlowQ improves by -0.2 ppl on average. Relative to FP16, the remaining C4 gap is +0.4 ppl on average, while average accuracy remains close to FP16 across the board. The selective-restore variant (GlowQ-S) shows the expected efficiency trade-off: −0.55% on average accuracy and +0.15 ppl on average in C4 compared to GlowQ.

### 4.3 Latency and Throughput Benefits from Caching and Selective Restore

**Latency on LLaMA 2 models.** Under a common generation protocol (3 prompts, batch=1, max_new_tokens=128, repeats=1, num_beams=1) and custom `CUDA W4A16` kernels, we measure TTFB via a warm-start `generate(max_new_tokens=1)` and per-token decode latency using CUDA events (Table 3). We establish our baseline using a standard Layerwise method, which does not employ caching. This setup ensures a fair comparison, as both the Layerwise baseline and GlowQ utilize the identical custom `CUDA W4A16` kernels compiled with the same optimization

Table 3: Latency comparison on LLaMA 2 models for Layerwise vs. GlowQ, GlowQ-S.

| Models | | Setting | TTFB(ms) ↓ | tok/s ↑ | Prefill(ms) ↓ | Dec(ms/tok) ↓ |
|---|---|---|---|---|---|---|
| | | Layerwise | 88.45 | 15.66 | 95.13 | 63.17 |
| | 7B | GlowQ | 82.66 | 17.12 | 92.23 | 58.32 |
| | | GlowQ-S | 66.68 | 21.16 | 72.35 | 45.90 |
| LLaMA 2 | | Layerwise | 128.70 | 11.22 | 141.76 | 85.91 |
| | 13B | GlowQ | 122.78 | 12.33 | 136.53 | 81.15 |
| | | GlowQ-S | 100.17 | 15.68 | 112.09 | 62.98 |
| | *Avg. Δ BX (%)* | | **-5.57** | **+9.61** | **-3.37** | **-6.61** |
| | *Avg. Δ R50 (%)* | | **-23.39** | **+37.44** | **-22.44** | **-27.01** |

level, isolating the algorithmic impact of our caching strategy. Compared to this Layerwise baseline, GlowQ consistently reduces end-to-end latency across both sizes: on average TTFB drops by $5.51\%$, prefill time by $3.37\%$, and decode latency by $6.61\%$, yielding a $9.61\%$ increase in throughput (tok/s).

**Selective restore efficiency.** The GlowQ-S, which are restoring about half of the units by an importance score, amplifies the gains: average TTFB, prefill, and decode fall by $23.39\%$, $22.44\%$, and $27.01\%$, respectively, and throughput increases by $37.44\%$ over the Layerwise baseline.

### 4.4 MEMORY OVERHEAD AND EFFICIENCY ANALYSIS

On memory, GlowQ consistently uses less additional GPU memory than layer-wise restoration at the same rank $r$. This follows from maintaining a single shared right factor $B_{\text{shared}}$ per input-sharing group and computing $R = B_{\text{shared}}X$ once per group for cache-and-reuse. Applying GlowQ-S further reduces overhead, yielding the flattest growth slope even at higher ranks. On accuracy, under an equal-memory budget in Fig. 4(b), GlowQ attains the lowest PPL, while GlowQ-S preserves PPL close to full GlowQ with substantially lower memory, consistently outperforming than layer-wise methods. Consequently, GlowQ is the preferred choice when maximizing performance within a fixed memory budget, whereas GlowQ-S offers a strong performance-efficiency compromise when memory constraints are tighter or latency minimization is prioritized.

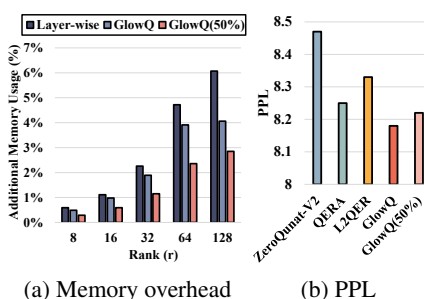

(a) Memory overhead  (b) PPL

Figure 4: Comparison of memory and performance trade-off. (a) Memory overhead of different methods. (b) PPL at equal memory budget.

### 4.5 COMPATIBILITY WITH PTQ METHODS AND GENERALIZATION TO MoE

We further examine the compatibility of GlowQ with diverse LLM configurations, focusing in particular on PTQ baselines and MoE architectures, as summarized in Table 4.

Layering GlowQ on top of PTQ baselines reduces perplexity by -0.59 ppl on average for GPTQ and -0.13 ppl on average for BnB. Improvements hold across both evaluated models in each setting, indicating consistent add-on gains independent of the underlying quantizer. GlowQ acts as an orthogonal, plug-and-play low-rank correction: it exchanges a small set of shared parameters for accuracy gains while remaining compatible with diverse PTQ pipelines.

On this MoE benchmark, GlowQ largely recovers the Wikitext-2 perplexity loss from 4-bit weight quantization and ends up only +0.02 PPL worse

Table 4: Perplexity (↓) on Wikitext-2 with and without GlowQ: dense models (top) and Qwen1.5-MoE-A2.7B (bottom).

| Method | LLaMA 2-7B | LLaMA 3.2-3B |
|---|---|---|
| GPTQ | 5.64 | 9.32 |
| +GlowQ (on GPTQ) | **5.60** | **8.19** |
| BnB | 5.64 | 8.29 |
| +GlowQ (on BnB) | **5.57** | **8.10** |

| | FP16 | Quant only | GlowQ | Layerwise |
|---|---|---|---|---|
| Qwen1.5-MoE-A2.7B | 7.22 | 7.70 | **7.41** | **7.39** |

than the more expensive layer-wise low-rank baseline. The layer-wise variant attaches a separate error-correction module to every expert, whereas GlowQ uses a single shared right factor $B_{\text{shared}}$ per group across experts and the shared MLP. The whitening-based alignment heatmaps in Fig. 7 and Fig. 8 show that expert-specific error subspaces are well aligned with this shared right subspace, explaining why the shared-$B$ design can match layer-wise accuracy while reducing the memory footprint of the low-rank correction by about $63\%$. These results confirm that GlowQ remains effective even on large MoE architectures.

Given the recent trend toward rotation-based saliency-aware PTQ and KV-cache compression, GlowQ can be viewed as a complementary low-rank correction layer that may be attached to strong PTQ baselines such as ROSAQ Yoon et al. (2025) and GuidedQuant Kim et al. (2025), and further extended to KV-cache compression frameworks like CommVQ Li et al. (2025); exploring such combinations remains an interesting direction for future work.

## 4.6 Behavior of Selective Restoration Across Model Families

Fig. 3 plots PPL and TTFB as a function of the restored fraction. On LLaMA 3.2-3B, PPL stays relatively flat and then exhibits an abrupt drop at an elbow point, after which marginal gains saturate quickly. In contrast, Qwen 2.5-7B shows a more gradual, near-monotone PPL decrease with increasing restoration, without a clear knee. Since TTFB generally grows with the restoration fraction, these shapes motivate different selective-restoration budgets. We verify that these family-specific tendencies persist across other sizes within each family in our ablation study (Sec. G).

Guided by the above curves, GlowQ-S restores (i) for LLaMA, the elbow (steep-drop) operating point to capture most PPL gains with limited overhead, and (ii) for Qwen, a fixed **50**% of groups, which offers stable accuracy improvements with moderate TTFB growth. For unit ranking, we follow the importance metrics delineated in Sec. 3.3: covariance-aware error capture is adopted as the default criterion. For the model families, when two alternative metrics are available (covariance-aware error capture vs. normalized error ratio), we evaluate both on the validation split and, per model, adopt the metric that yields the stronger outcome; all reported results use this per-model best choice.

## 4.7 Impact of Covariance Alignment on Accuracy

On C4 (Table 5), the $\Sigma_x$-weighted Whitened SVD consistently outperforms the unweighted Stacked SVD across both layer-wise and group-shared variants of Qwen 3-8B. Because the unweighted objective ignores the input-usage distribution embodied in $\Sigma_x$, it tends to select right subspaces misaligned with the axes most exploited by the data, leading to a marked degradation in perplexity at a fixed rank; whitening, by evaluating errors in a data-aligned coordinate system, improves energy capture and

Table 5: C4 Evaluation of $\Sigma_x$-weighted (Whitened SVD) vs. unweighted (Stacked SVD) on Qwen 3-8B; lower is better ($\downarrow$).

| No-White | | White | |
| --- | --- | --- | --- |
| Layer | Group | Layer | Group |
| 14.97 | 14.60 | **13.85** | **13.40** |

yields lower PPL. Grouped restoration also dominates layer-wise under both weightings, and, taken together, these results identify White + Group as the preferred configuration.

## 5 Conclusions

We introduced GlowQ, a group-shared low-rank approximation for quantized LLMs that replaces per-layer correction with a single right subspace shared among input-sharing modules and a cache-and-reuse runtime. By connecting usage-weighted risk to a right-weighted reconstruction objective, our covariance-aligned (whitened) formulation steers the learned subspace toward data-preferred directions, and a QR-reduced randomized SVD provides an efficient, scalable solver. The deployment path computes one right-side projection per group and reuses it across modules, while a selective policy (GlowQ-S) activates only high-importance units under latency or memory budgets. Across modern model families and PTQ baselines, GlowQ consistently lowers perplexity, reduces time-to-first-byte, increases throughput, and decreases memory overhead relative to layer-wise correction; whitening and grouping combine to yield the strongest results. The approach is architecture-agnostic, drop-in at inference, and complementary to existing PTQ pipelines.

ACKNOWLEDGMENTS

This work was supported by the National Research Foundation of Korea (NRF) grant funded by the Korea government (MSIT) (RS-2025-24803164). This research was also supported by the Ministry of Trade Industry & Energy (MOTIE, Korea), under the Technology Innovation Program titled "Development of Navigation Technology Utilizing Visual Information Based on Vision-Language Models for Understanding Dynamic Environments in Non-Learned Spaces" (Project Number: RS-2024-00445759).

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

## A  APPENDIX

### A.0 NOTATION & SHAPES

Refer to Table 6

Table 6: Unified notation and shapes for stacked errors, low-rank factors, and covariance-weighted core.

| Symbol | Meaning |
|---|---|
| $\mathbf{E}_i \in \mathbb{R}^{O_i \times d}$ | Error matrix of module $i$ with output dimension $O_i$ and shared input dimension $d$. |
| $\mathbf{E}_{\mathrm{cat}} := [\mathbf{E}_1; \ldots; \mathbf{E}_m] \in \mathbb{R}^{m \times d}$ | Row-stacked errors across modules; total rows $m := \sum_i O_i$. |
| $\mathbf{A} := [\mathbf{A}_1; \ldots; \mathbf{A}_m] \in \mathbb{R}^{m \times r}$ | Left factor formed by stacking per-module factors $\mathbf{A}_i$. |
| $\mathbf{B} \in \mathbb{R}^{r \times d}$ | Shared right factor; target rank $r$. |
| $1 \le r \le \min\{m, d\}$ | Admissible rank range. |
| $\mathbf{E}_{\mathrm{cat}} = \mathbf{U}\boldsymbol{\Sigma}\mathbf{V}^\top$ | Thin SVD with $\mathbf{U} \in \mathbb{R}^{m \times d}$, $\boldsymbol{\Sigma} \in \mathbb{R}^{d \times d}$, $\mathbf{V} \in \mathbb{R}^{d \times d}$ orthogonal. |
| $(\mathbf{U}_r, \boldsymbol{\Sigma}_r, \mathbf{V}_r)$ | Top-$r$ SVD blocks: $\mathbf{U}_r \in \mathbb{R}^{m \times r}$, $\boldsymbol{\Sigma}_r \in \mathbb{R}^{r \times r}$, $\mathbf{V}_r \in \mathbb{R}^{d \times r}$. |
| $\boldsymbol{\Sigma}_{\mathbf{x}} \succeq \mathbf{0}$ | Input covariance; $\boldsymbol{\Sigma}_{\mathbf{x}} := \mathbb{E}[\mathbf{x}\mathbf{x}^\top]$ for centered inputs $\mathbb{E}[\mathbf{x}] = \mathbf{0}$. |
| $\boldsymbol{\Sigma}_{\mathbf{x}}^{1/2}$ | (Pseudo-)square root of $\boldsymbol{\Sigma}_{\mathbf{x}}$. |
| $\mathbf{E}_{\mathrm{cat}} = \mathbf{Q}_e \mathbf{R}_e$ | Thin QR with $\mathbf{Q}_e \in \mathbb{R}^{m \times d}$, $\mathbf{Q}_e^\top \mathbf{Q}_e = \mathbf{I}_d$, $\mathbf{R}_e \in \mathbb{R}^{d \times d}$. |
| $\mathbf{M} := \mathbf{R}_e \boldsymbol{\Sigma}_{\mathbf{x}}^{1/2} \in \mathbb{R}^{d \times d}$ | Covariance-weighted SVD core used for randomized SVD on the reduced space. |
| $\widehat{\mathbf{A}} := \mathbf{Q}_e^\top \mathbf{A} \in \mathbb{R}^{d \times r}$ | Variable change (reduced left factor). |
| $\widehat{\mathbf{B}} := \mathbf{B}\boldsymbol{\Sigma}_{\mathbf{x}}^{1/2} \in \mathbb{R}^{r \times d}$ | Variable change (covariance-weighted right factor). |
| Residual $(\mathbf{E}_{\mathrm{cat}} - \mathbf{A}\mathbf{B}) \in \mathbb{R}^{m \times d}$ | Stacked error after factorization (no separate symbol reserved). |

### A.1  STACKED SVD: SHARED RIGHT SUBSPACE AND GLOBAL OPTIMUM (PROOF)

When multiple modules share the same input dimension, we vertically concatenate the module-wise error matrices $\mathbf{E}_i \in \mathbb{R}^{O_i \times d}$ into $\mathbf{E}_{\mathrm{cat}}$. We then choose a shared right subspace (the row space of $\mathbf{B}$) while allowing module-specific left factors $\mathbf{A}_i$, by solving

$$\min_{\mathbf{A},\mathbf{B}} \left\| \mathbf{E}_{\mathrm{cat}} - \mathbf{A}\mathbf{B} \right\|_F^2.$$

This appendix shows that (i) the solution is well-defined, and (ii) the shared $\mathbf{B}$ is also optimal in an energy/projection sense (Ky Fan; cf. Fan (1950); Golub & Van Loan (2013)). Consequently, a single shared $\mathbf{B}$ serves as a strong representative of what one might otherwise try to learn as separate $\mathbf{B}_i$'s per module.

**Problem (Unweighted Frobenius Approximation).**

$$\min_{\mathbf{A},\mathbf{B}} \left\| \mathbf{E}_{\text{cat}} - \mathbf{A}\mathbf{B} \right\|_F^2. \tag{A.1.1}$$

LEMMA A.1.1 - EQUIVALENCE OF SEARCH SETS: $\mathcal{M}_r = \mathcal{R}_r$.

Let

$$\mathcal{M}_r := \{\mathbf{A}\mathbf{B} : \ \mathbf{A} \in \mathbb{R}^{m \times r}, \ \mathbf{B} \in \mathbb{R}^{r \times d}\}, \qquad \mathcal{R}_r := \{\mathbf{X} \in \mathbb{R}^{m \times d} : \ \text{rank}(\mathbf{X}) \le r\}.$$

Then $\mathcal{M}_r = \mathcal{R}_r$.

This is a standard consequence of rank-factorization and the SVD characterization of best rank-$r$ approximants; see, e.g., Eckart & Young (1936); MIRSKY (1960); Golub & Van Loan (2013); Horn & Johnson (1985). We omit the proof.

**Proof.** By Lemma A.1.1, the problem reduces to a rank-$r$ approximation of $\mathbf{E}_{\text{cat}}$. By the Eckart-Young-Mirsky theorem Eckart & Young (1936); MIRSKY (1960), the optimizer is the truncated SVD

$$\mathbf{X}^\star = \mathbf{U}_r \mathbf{\Sigma}_r \mathbf{V}_r^\top,$$

so any global minimizer $(\mathbf{A}, \mathbf{B})$ must satisfy

$$\mathbf{A}\mathbf{B} = \mathbf{X}^\star = \mathbf{U}_r \mathbf{\Sigma}_r \mathbf{V}_r^\top. \tag{A.1.2}$$

If $\sigma_r = \sigma_{r+1}$, the optimizer may be non-unique Golub & Van Loan (2013). $\square$

THEOREM A.1.2 - IDENTIFYING THE SHARED RIGHT SUBSPACE: $\text{row}(\mathbf{B}) = \text{span}(\mathbf{V}_r^\top)$.

We determine the optimal shared right subspace for the factorization $\min_{\mathbf{A},\mathbf{B}} \ \|\mathbf{E}_{\text{cat}} - \mathbf{A}\mathbf{B}\|_F^2$. Let $\mathbf{E}_{\text{cat}} = \mathbf{U}\mathbf{\Sigma}\mathbf{V}^\top$ be a thin SVD, and let $r = \text{rank}(\mathbf{B})$. Denote $\mathbf{S} := \text{row}(\mathbf{B})$ and the orthogonal projector $\mathbf{P_S} := \mathbf{B}^\top (\mathbf{B}\mathbf{B}^\top)^{-1} \mathbf{B}$ (assume $\mathbf{B}$ has full row rank; otherwise use the Moore-Penrose pseudoinverse).

Fixing $\mathbf{B}$, least-squares normal equations yield (see, e.g., (Golub & Van Loan, 2013, §5))

$$\mathbf{A}^* = \mathbf{E}_{\text{cat}} \mathbf{B}^\top (\mathbf{B}\mathbf{B}^\top)^{-1}, \tag{A.1.3a}$$

$$\mathbf{A}^* \mathbf{B} = \mathbf{E}_{\text{cat}} \mathbf{P_S}. \tag{A.1.3b}$$

Hence, with $\mathbf{G} := \mathbf{E}_{\text{cat}}^\top \mathbf{E}_{\text{cat}}$,

$$\|\mathbf{E}_{\text{cat}} - \mathbf{A}^* \mathbf{B}\|_F^2 = \|\mathbf{E}_{\text{cat}}(\mathbf{I} - \mathbf{P_S})\|_F^2 = \|\mathbf{E}_{\text{cat}}\|_F^2 - \text{tr}(\mathbf{P_S}\mathbf{G}), \tag{A.1.4}$$

where the last identity is the usual projection-trace formula (cf. Horn & Johnson (1985)).

Therefore, selecting $\mathbf{S}$ of dimension $r$ is equivalent to

$$\max_{\dim \mathbf{S} = r} \ \text{tr}(\mathbf{P_S}\mathbf{G}). \tag{A.1.5}$$

By Ky Fan's maximum principle Fan (1950), the maximizer $\mathbf{S}$ is the span of the top-$r$ eigenvectors of $\mathbf{G}$. Since $\mathbf{E}_{\text{cat}} = \mathbf{U}\mathbf{\Sigma}\mathbf{V}^\top$ implies $\mathbf{G} = \mathbf{V}\mathbf{\Sigma}^2\mathbf{V}^\top$, its top-$r$ eigenspace equals $\text{span}(\mathbf{V}_r)$. Thus

$$\text{row}(\mathbf{B}) = \text{span}(\mathbf{V}_r^\top). \tag{A.1.6}$$

$\square$

THEOREM A.1.3 - REPRESENTATIVENESS / ENERGY OPTIMALITY: SUM OF PROJECTION ENERGIES.

The shared right subspace $\mathbf{S} = \text{row}(\mathbf{B})$ of dimension $r$ maximizes the total projection energy $\sum_i \|\mathbf{E}_i \mathbf{P_S}\|_F^2$, where $\mathbf{P_S}$ is the orthogonal projector onto $\mathbf{S}$ (e.g., $\mathbf{P_S} = \mathbf{Q}\mathbf{Q}^\top$ for any orthonormal basis $\mathbf{Q}$ of $\mathbf{S}$).

**Proof.** For each module $\mathbf{E}_i$,

$$\|\mathbf{E}_i \mathbf{P_S}\|_F^2 = \mathrm{tr}\big(\mathbf{P_S}\, \mathbf{E}_i^\top \mathbf{E}_i\big), \tag{A.1.7}$$

a standard identity using symmetry/idempotence of $\mathbf{P_S}$ and trace cyclicity (see, e.g., Horn & Johnson (1985); Golub & Van Loan (2013)). Summing over $i$ yields

$$\max_{\dim \mathbf{S}=r} \sum_i \|\mathbf{E}_i \mathbf{P_S}\|_F^2 \;=\; \max_{\dim \mathbf{S}=r} \mathrm{tr}\Big(\mathbf{P_S} \sum_i \mathbf{E}_i^\top \mathbf{E}_i\Big) \;=\; \max_{\dim \mathbf{S}=r} \mathrm{tr}(\mathbf{P_S}\mathbf{G}), \quad \mathbf{G} := \sum_i \mathbf{E}_i^\top \mathbf{E}_i = \mathbf{E}_{\mathrm{cat}}^\top \mathbf{E}_{\mathrm{cat}}. \tag{A.1.8}$$

By Ky Fan's maximum principle Fan (1950) (cf. Eq. A.1.5), the maximizer is the span of the top-$r$ eigenvectors of $\mathbf{G}$. Since $\mathbf{E}_{\mathrm{cat}} = \mathbf{U}\boldsymbol{\Sigma}\mathbf{V}^\top$ implies $\mathbf{G} = \mathbf{V}\boldsymbol{\Sigma}^2\mathbf{V}^\top$, it follows that

$$\mathbf{S}^* = \mathrm{span}(\mathbf{V}_r) \quad \Longleftrightarrow \quad \mathrm{row}(\mathbf{B}) = \mathrm{span}(\mathbf{V}_r^\top). $$

$\square$

LEMMA A.1.4 - IDENTIFIABILITY AND "BALANCED" FACTORIZATION.

Although the pair $(\mathbf{A}, \mathbf{B})$ is non-unique up to invertible reparameterizations, the right subspace $\mathrm{row}(\mathbf{B})$ is identifiable; choosing the SVD half-split $\boldsymbol{\Sigma}_r^{1/2}$ yields a numerically stable balanced factorization Golub & Van Loan (2013).

**Non-uniqueness.** For any invertible $\mathbf{R} \in \mathbb{R}^{r \times r}$,

$$(\mathbf{A}, \mathbf{B}) \mapsto (\mathbf{A}\mathbf{R},\ \mathbf{R}^{-1}\mathbf{B}) \quad \Rightarrow \quad \mathbf{A}\mathbf{B}\ \text{invariant}.$$

Hence factors are not unique, while the projector onto $\mathrm{row}(\mathbf{B})$ is unique (right singular subspace; cf. Theorem. A.1.2 and Golub & Van Loan (2013)).

**Balanced factorization.** Let $\mathbf{E}_{\mathrm{cat}} = \mathbf{U}\boldsymbol{\Sigma}\mathbf{V}^\top$ and denote by $\mathbf{U}_r, \boldsymbol{\Sigma}_r, \mathbf{V}_r$ the top-$r$ blocks. The half-split

$$\mathbf{A}^* = \mathbf{U}_r \boldsymbol{\Sigma}_r^{1/2}, \tag{A.1.9a}$$

$$\mathbf{B}^* = \boldsymbol{\Sigma}_r^{1/2}\mathbf{V}_r^\top, \tag{A.1.9b}$$

$$\mathbf{A}^*\mathbf{B}^* = \mathbf{U}_r\boldsymbol{\Sigma}_r\mathbf{V}_r^\top \tag{A.1.9c}$$

satisfies

$$\mathbf{A}^{*\top}\mathbf{A}^* = \boldsymbol{\Sigma}_r, \qquad \mathbf{B}^*\mathbf{B}^{*\top} = \boldsymbol{\Sigma}_r,$$

which avoids squaring condition numbers in normal equations and minimizes combined factor norms among reparameterizations:

$$\frac{1}{2}\big(\|\mathbf{A}\|_F^2 + \|\mathbf{B}\|_F^2\big) \;\geq\; \|\mathbf{U}_r\boldsymbol{\Sigma}_r\mathbf{V}_r^\top\|_*,$$

with equality at $(\mathbf{A}^*, \mathbf{B}^*)$ Recht et al. (2010). (Standard facts; see Golub & Van Loan (2013); Recht et al. (2010).)

**Block Recovery and the Pseudoinverse** Given the shared right factor $\mathbf{B}^*$, each module-specific left factor $\mathbf{A}_i$ is obtained by a single least-squares solve. Using the Moore-Penrose pseudoinverse provides the minimum-norm solution and remains valid under rank deficiency Penrose (1955); Ben-Israel & Greville (2010); Golub & Van Loan (2013):

$$\mathbf{A}_i^* = \mathbf{E}_i\, \mathbf{B}^{*\top}\big(\mathbf{B}^*\mathbf{B}^{*\top}\big)^\dagger. \tag{A.1.10}$$

It suggests that (i) when $\mathbf{B}^*$ has full row rank, $(\cdot)^\dagger$ reduces to the inverse and Eq. A.1.10 coincides with the normal-equations solution; (ii) in general, $(\cdot)^\dagger$ yields the unique minimum-norm LS solution and is numerically stable under near-singularity Ben-Israel & Greville (2010); Golub & Van Loan (2013).

## A.2 COVARIANCE-ALIGNED OBJECTIVE: BRIDGE EQUIVALENCE AND GLOBAL MINIMIZER (PROOF)

Sec. 3.1.2 formulates the covariance-aligned objective

$$\min_{\mathbf{A},\mathbf{B}} \left\| (\mathbf{E}_{\mathrm{cat}} - \mathbf{A}\mathbf{B})\, \boldsymbol{\Sigma}_{\mathbf{x}}^{1/2} \right\|_F^2,$$

which weights errors by the input usage encoded in the covariance $\boldsymbol{\Sigma}_{\mathbf{x}}$ Anderson (1984); Bishop (2006). This appendix provides a complete mathematical justification: (i) a bridge equivalence that converts $\mathbb{E}\big[\|(\mathbf{E}_{\mathrm{cat}} - \mathbf{A}\mathbf{B})\mathbf{x}\|_2^2\big]$ into a Frobenius form via the trace identity $\mathbb{E}[\mathbf{x}^\top \mathbf{M}\mathbf{x}] = \mathrm{tr}(\mathbf{M}\boldsymbol{\Sigma}_{\mathbf{x}})$ Petersen & Pedersen (2006); (ii) a whitening reduction to a standard low-rank approximation by the change of variables $\widetilde{\mathbf{B}} := \mathbf{B}\boldsymbol{\Sigma}_{\mathbf{x}}^{1/2}$ (and $\mathbf{E}_{\mathrm{cat}}\boldsymbol{\Sigma}_{\mathbf{x}}^{1/2}$ on the right) Golub & Van Loan (2013); (iii) a closed-form global minimizer given by the truncated SVD of $\mathbf{E}_{\mathrm{cat}}\boldsymbol{\Sigma}_{\mathbf{x}}^{1/2}$ with balanced factors and the identity of the shared right subspace; and (iv) extensions to nonzero-mean inputs (centering) and singular $\boldsymbol{\Sigma}_{\mathbf{x}}$ via pseudoinverse whitening Ben-Israel & Greville (2010); Penrose (1955).

In our case, the (distribution-weighted) risk is the expected squared output error under the input law:

$$\mathcal{R}(\mathbf{A},\mathbf{B}) := \mathbb{E}\,\|\mathbf{M}\mathbf{x}\|_2^2.$$

Directions used more frequently or with larger magnitude (large variance) are weighted more heavily by $\boldsymbol{\Sigma}_{\mathbf{x}}$, which motivates a right-weighted objective via $\boldsymbol{\Sigma}_{\mathbf{x}}^{1/2}$ Bishop (2006); Anderson (1984). An empirical counterpart uses samples $\{\mathbf{x}_n\}_{n=1}^N$:

$$\widehat{\mathcal{R}}(\mathbf{A},\mathbf{B}) := \frac{1}{N}\sum_{n=1}^N \|\mathbf{M}\,\mathbf{x}_n\|_2^2, \qquad \widehat{\boldsymbol{\Sigma}}_{\mathbf{x}} := \frac{1}{N}\sum_{n=1}^N \mathbf{x}_n\mathbf{x}_n^\top.$$

THEOREM A.2.1 (BRIDGE EQUIVALENCE).

In this subsection, we prove the bridge identity $\mathbb{E}\,\|\mathbf{M}\mathbf{x}\|_2^2 = \mathrm{tr}(\mathbf{M}\boldsymbol{\Sigma}_{\mathbf{x}}\mathbf{M}^\top) = \|\mathbf{M}\boldsymbol{\Sigma}_{\mathbf{x}}^{1/2}\|_F^2$, which converts the distribution-weighted risk into a Frobenius norm amenable to SVD analysis (see the trace/expectation identities in Petersen & Pedersen (2006)).

For zero-mean inputs with covariance $\boldsymbol{\Sigma}_{\mathbf{x}} \succeq \mathbf{0}$,

$$\mathbb{E}\,\|\mathbf{M}\mathbf{x}\|_2^2 = \mathrm{tr}(\mathbf{M}\boldsymbol{\Sigma}_{\mathbf{x}}\mathbf{M}^\top) = \left\|\mathbf{M}\boldsymbol{\Sigma}_{\mathbf{x}}^{1/2}\right\|_F^2. \tag{A.2.1}$$

**Proof.** (Vector norm $\to$ trace). Since $\|\mathbf{y}\|_2^2 = \mathrm{tr}(\mathbf{y}\mathbf{y}^\top)$ and trace is linear,

$$\mathbb{E}\,\|\mathbf{M}\mathbf{x}\|_2^2 = \mathbb{E}\,\mathrm{tr}(\mathbf{M}\mathbf{x}\mathbf{x}^\top\mathbf{M}^\top) = \mathrm{tr}\big(\mathbf{M}\,\mathbb{E}[\mathbf{x}\mathbf{x}^\top]\,\mathbf{M}^\top\big) = \mathrm{tr}(\mathbf{M}\boldsymbol{\Sigma}_{\mathbf{x}}\mathbf{M}^\top).$$

(Trace $\to$ Frobenius). Because $\|\mathbf{Z}\|_F^2 = \mathrm{tr}(\mathbf{Z}\mathbf{Z}^\top)$ and $\boldsymbol{\Sigma}_{\mathbf{x}}^{1/2}\boldsymbol{\Sigma}_{\mathbf{x}}^{1/2} = \boldsymbol{\Sigma}_{\mathbf{x}}$,

$$\|\mathbf{M}\boldsymbol{\Sigma}_{\mathbf{x}}^{1/2}\|_F^2 = \mathrm{tr}\big((\mathbf{M}\boldsymbol{\Sigma}_{\mathbf{x}}^{1/2})(\mathbf{M}\boldsymbol{\Sigma}_{\mathbf{x}}^{1/2})^\top\big) = \mathrm{tr}(\mathbf{M}\boldsymbol{\Sigma}_{\mathbf{x}}\mathbf{M}^\top). \quad \square$$

Distribution-weighted risk equals the Frobenius norm of the right-whitened residual $\mathbf{M}\boldsymbol{\Sigma}_{\mathbf{x}}^{1/2}$.

LEMMA A.2.2 (NONZERO-MEAN INPUTS).

In this subsection, we decompose the risk for $\mathbb{E}[\mathbf{x}] \neq \mathbf{0}$ into a covariance term and a deterministic mean term, showing $\mathbb{E}\|\mathbf{M}\mathbf{x}\|_2^2 = \mathrm{tr}(\mathbf{M}\,\mathrm{Cov}(\mathbf{x})\,\mathbf{M}^\top) + \|\mathbf{M}\boldsymbol{\mu}\|_2^2$ (cf. Anderson (1984); Bishop (2006)).

Let $\boldsymbol{\mu} := \mathbb{E}[\mathbf{x}]$ and $\mathrm{Cov}(\mathbf{x}) := \mathbb{E}[(\mathbf{x} - \boldsymbol{\mu})(\mathbf{x} - \boldsymbol{\mu})^\top]$. Then

$$\mathbb{E}\,\|\mathbf{M}\mathbf{x}\|_2^2 = \mathrm{tr}\big(\mathbf{M}\,\mathrm{Cov}(\mathbf{x})\,\mathbf{M}^\top\big) + \|\mathbf{M}\boldsymbol{\mu}\|_2^2. \tag{A.2.2}$$

**Proof.** Write $\mathbf{x} = (\mathbf{x} - \boldsymbol{\mu}) + \boldsymbol{\mu}$ and expand:

$$\|\mathbf{M}\mathbf{x}\|_2^2 = \|\mathbf{M}(\mathbf{x} - \boldsymbol{\mu})\|_2^2 + 2\langle \mathbf{M}(\mathbf{x} - \boldsymbol{\mu}), \mathbf{M}\boldsymbol{\mu}\rangle + \|\mathbf{M}\boldsymbol{\mu}\|_2^2.$$

Taking expectations annihilates the cross term since $\mathbb{E}[\mathbf{x} - \boldsymbol{\mu}] = \mathbf{0}$, yielding the claim. $\square$

Risk decomposes into a covariance term plus a mean-induced term.

THEOREM A.2.3 (VARIABLE CHANGE AND WHITENING).

In this subsection, we show that right-whitening reduces the covariance-aligned objective to a standard Frobenius low-rank approximation by proving $\left\|(\mathbf{E}_{\text{cat}} - \mathbf{AB})\boldsymbol{\Sigma}_{\mathbf{x}}^{1/2}\right\|_F^2 = \|\widetilde{\mathbf{E}} - \widehat{\mathbf{A}}\widehat{\mathbf{B}}\|_F^2$ with $\widetilde{\mathbf{E}} = \mathbf{E}_{\text{cat}}\boldsymbol{\Sigma}_{\mathbf{x}}^{1/2}$, $\widehat{\mathbf{B}} = \mathbf{B}\boldsymbol{\Sigma}_{\mathbf{x}}^{1/2}$ (standard whitening trick; cf. Golub & Van Loan (2013)).

Define

$$\widetilde{\mathbf{E}} := \mathbf{E}_{\text{cat}}\boldsymbol{\Sigma}_{\mathbf{x}}^{1/2}, \qquad \widehat{\mathbf{A}} := \mathbf{A}, \qquad \widehat{\mathbf{B}} := \mathbf{B}\,\boldsymbol{\Sigma}_{\mathbf{x}}^{1/2}.$$

Then

$$\left\|(\mathbf{E}_{\text{cat}} - \mathbf{AB})\boldsymbol{\Sigma}_{\mathbf{x}}^{1/2}\right\|_F^2 = \|\widetilde{\mathbf{E}} - \widehat{\mathbf{A}}\,\widehat{\mathbf{B}}\|_F^2. \tag{A.2.3}$$

**Proof.** Direct substitution:

$$\widetilde{\mathbf{E}} - \widehat{\mathbf{A}}\widehat{\mathbf{B}} = \mathbf{E}_{\text{cat}}\boldsymbol{\Sigma}_{\mathbf{x}}^{1/2} - \mathbf{A}(\mathbf{B}\boldsymbol{\Sigma}_{\mathbf{x}}^{1/2}) = (\mathbf{E}_{\text{cat}} - \mathbf{AB})\boldsymbol{\Sigma}_{\mathbf{x}}^{1/2}.$$

Taking Frobenius norms yields the identity. $\square$

Whitening converts risk minimization into a plain Frobenius factorization.

LEMMA A.2.4 (WEIGHTED LEAST SQUARES FOR $A$ GIVEN $B$).

In this subsection, we derive the closed-form weighted least-squares minimizer $\mathbf{A}^* = \mathbf{E}\,\boldsymbol{\Sigma}_{\mathbf{x}}\,\mathbf{B}^\top(\mathbf{B}\,\boldsymbol{\Sigma}_{\mathbf{x}}\,\mathbf{B}^\top)^{-1}$ for fixed $\mathbf{B}$, and interpret the residual as a $\boldsymbol{\Sigma}_{\mathbf{x}}$-weighted right projection ((Golub & Van Loan, 2013, Ch. 5), Björck (1996); matrix derivatives in Petersen & Pedersen (2006)).

Consider

$$f(\mathbf{A}) := \left\|(\mathbf{E}_{\text{cat}} - \mathbf{AB})\,\boldsymbol{\Sigma}_{\mathbf{x}}^{1/2}\right\|_F^2.$$

Let $\mathbf{E} := \mathbf{E}_{\text{cat}}$, $\mathbf{E}_\sim := \mathbf{E}\boldsymbol{\Sigma}_{\mathbf{x}}^{1/2}$, and $\widehat{\mathbf{B}} := \mathbf{B}\boldsymbol{\Sigma}_{\mathbf{x}}^{1/2}$. Then the unique least-squares minimizer is

$$\mathbf{A}^* = \mathbf{E}_\sim \widehat{\mathbf{B}}^\top(\widehat{\mathbf{B}}\,\widehat{\mathbf{B}}^\top)^{-1} = \mathbf{E}\,\boldsymbol{\Sigma}_{\mathbf{x}}\,\mathbf{B}^\top\,(\mathbf{B}\,\boldsymbol{\Sigma}_{\mathbf{x}}\,\mathbf{B}^\top)^{-1}. \tag{A.2.4}$$

**Proof.** In whitened variables,

$$f(\mathbf{A}) = \|\mathbf{E}_\sim - \mathbf{A}\widehat{\mathbf{B}}\|_F^2 = \text{tr}(\mathbf{E}_\sim \mathbf{E}_\sim^\top) - 2\,\text{tr}(\mathbf{A}\widehat{\mathbf{B}}\mathbf{E}_\sim^\top) + \text{tr}\!\left(\mathbf{A}(\widehat{\mathbf{B}}\widehat{\mathbf{B}}^\top)\mathbf{A}^\top\right).$$

Using $\frac{\partial}{\partial \mathbf{A}}\text{tr}(\mathbf{ACA}^\top) = 2\mathbf{A}\,\mathbf{C}$ for symmetric $\mathbf{C}$ and $\frac{\partial}{\partial \mathbf{A}}\text{tr}(\mathbf{AM}) = \mathbf{M}^\top$ Petersen & Pedersen (2006),

$$\nabla_{\mathbf{A}}f(\mathbf{A}) = -2\mathbf{E}_\sim \widehat{\mathbf{B}}^\top + 2\mathbf{A}(\widehat{\mathbf{B}}\widehat{\mathbf{B}}^\top) = 0 \;\Rightarrow\; \mathbf{A}^* = \mathbf{E}_\sim \widehat{\mathbf{B}}^\top(\widehat{\mathbf{B}}\widehat{\mathbf{B}}^\top)^{-1}.$$

Substituting $\mathbf{E}_\sim = \mathbf{E}\boldsymbol{\Sigma}_{\mathbf{x}}^{1/2}$ and $\widehat{\mathbf{B}} = \mathbf{B}\boldsymbol{\Sigma}_{\mathbf{x}}^{1/2}$ gives the second form. $\square$

In whitened variables, $\mathbf{E}_\sim - \mathbf{A}^*\widehat{\mathbf{B}} = \mathbf{E}_\sim(\mathbf{I} - \mathbf{P}_{\widehat{\mathbf{S}}})$ with $\mathbf{P}_{\widehat{\mathbf{S}}} := \widehat{\mathbf{B}}^\top(\widehat{\mathbf{B}}\widehat{\mathbf{B}}^\top)^{-1}\widehat{\mathbf{B}}$, the orthogonal projector onto $\text{row}(\widehat{\mathbf{B}})$ in the Euclidean metric. In original variables, $\mathbf{A}^*\mathbf{B} = \mathbf{E}\,\mathbf{P}_{\boldsymbol{\Sigma}}$ with

$$\mathbf{P}_{\boldsymbol{\Sigma}} := \boldsymbol{\Sigma}_{\mathbf{x}}\,\mathbf{B}^\top\,(\mathbf{B}\,\boldsymbol{\Sigma}_{\mathbf{x}}\,\mathbf{B}^\top)^{-1}\mathbf{B},$$

the right projection under the $\boldsymbol{\Sigma}_{\mathbf{x}}$-weighted inner product (a standard form of weighted/oblique projection;( cf. Golub & Van Loan (2013), Ben-Israel & Greville (2010); Björck (1996)).

For fixed $\mathbf{B}$, the optimal $\mathbf{A}$ is a weighted LS solution; the residual is a $\boldsymbol{\Sigma}_{\mathbf{x}}$-weighted right projection.

THEOREM A.2.5 (GLOBAL MINIMIZER; BALANCED FACTORS; RIGHT SUBSPACE).

In this subsection, we obtain the global solution via the Eckart–Young–Mirsky theorem Eckart & Young (1936); MIRSKY (1960), choose balanced factors $\widehat{\mathbf{A}}^\star = \mathbf{U}_r \boldsymbol{\Sigma}_r^{1/2}$, $\widehat{\mathbf{B}}^\star = \boldsymbol{\Sigma}_r^{1/2} \mathbf{V}_r^\top$, and identify the optimal shared right subspace as $\mathrm{row}(\mathbf{B}^\star) = \mathrm{row}(\mathbf{V}_r^\top \boldsymbol{\Sigma}_\mathbf{x}^{-1/2})$ (cf. Ky Fan's principle and the subspace discussion in Fan (1950); Golub & Van Loan (2013)).

Let $\widetilde{\mathbf{E}} = \mathbf{U}\boldsymbol{\Sigma}\mathbf{V}^\top$ be an SVD and $(\mathbf{U}_r, \boldsymbol{\Sigma}_r, \mathbf{V}_r)$ the top-$r$ blocks. Then

$$\widehat{\mathbf{A}}^\star = \mathbf{U}_r \boldsymbol{\Sigma}_r^{1/2}, \qquad \widehat{\mathbf{B}}^\star = \boldsymbol{\Sigma}_r^{1/2} \mathbf{V}_r^\top$$

achieve the global optimum of $\min_{\widehat{\mathbf{A}}, \widehat{\mathbf{B}}} \|\widetilde{\mathbf{E}} - \widehat{\mathbf{A}}\widehat{\mathbf{B}}\|_F^2$, with minimum value $\sum_{i>r} \sigma_i(\widetilde{\mathbf{E}})^2$ Eckart & Young (1936); MIRSKY (1960); Golub & Van Loan (2013). In original variables,

$$\mathbf{A}^\star = \widehat{\mathbf{A}}^\star = \mathbf{U}_r \boldsymbol{\Sigma}_r^{1/2}, \qquad \mathbf{B}^\star = \widehat{\mathbf{B}}^\star \boldsymbol{\Sigma}_\mathbf{x}^{-1/2} = \boldsymbol{\Sigma}_r^{1/2} \mathbf{V}_r^\top \boldsymbol{\Sigma}_\mathbf{x}^{-1/2},$$

and

$$\mathrm{row}(\mathbf{B}^\star) = \mathrm{row}(\mathbf{V}_r^\top \boldsymbol{\Sigma}_\mathbf{x}^{-1/2}). \tag{A.2.5}$$

**Proof.**  By Theorem A.2.3,

$$\min_{\mathbf{A}, \mathbf{B}} \|(\mathbf{E}_{\mathrm{cat}} - \mathbf{A}\mathbf{B})\boldsymbol{\Sigma}_\mathbf{x}^{1/2}\|_F^2 = \min_{\widehat{\mathbf{A}}, \widehat{\mathbf{B}}} \|\widetilde{\mathbf{E}} - \widehat{\mathbf{A}}\widehat{\mathbf{B}}\|_F^2.$$

Left/right orthogonal invariance of the Frobenius norm reduces the problem to $\min_{\mathrm{rank}(\mathbf{Y}) \leq r} \|\boldsymbol{\Sigma} - \mathbf{Y}\|_F^2$, solved by the truncated SVD $\mathbf{Y}^\star = \boldsymbol{\Sigma}_r \oplus \mathbf{0}$; hence $\mathbf{X}^\star = \mathbf{U}_r \boldsymbol{\Sigma}_r \mathbf{V}_r^\top$ Eckart & Young (1936); MIRSKY (1960). Choosing $\widehat{\mathbf{A}}^\star = \mathbf{U}_r \boldsymbol{\Sigma}_r^{1/2}$ and $\widehat{\mathbf{B}}^\star = \boldsymbol{\Sigma}_r^{1/2} \mathbf{V}_r^\top$ produces $\mathbf{X}^\star = \widehat{\mathbf{A}}^\star \widehat{\mathbf{B}}^\star$. Returning to original variables gives the stated $(\mathbf{A}^\star, \mathbf{B}^\star)$ and the row-space identity (cf. Fan (1950); Golub & Van Loan (2013)). $\square$

In whitened variables: $(\widehat{\mathbf{A}}^\star)^\top \widehat{\mathbf{A}}^\star = \boldsymbol{\Sigma}_r$ and $\widehat{\mathbf{B}}^\star(\widehat{\mathbf{B}}^\star)^\top = \boldsymbol{\Sigma}_r$. In original variables: $(\mathbf{A}^\star)^\top \mathbf{A}^\star = \boldsymbol{\Sigma}_r$ and $\mathbf{B}^\star \boldsymbol{\Sigma}_\mathbf{x}(\mathbf{B}^\star)^\top = \boldsymbol{\Sigma}_r$ Golub & Van Loan (2013). For any orthogonal $\mathbf{R} \in \mathbb{R}^{r \times r}$, $(\mathbf{A}\mathbf{R}, \mathbf{R}^\top \mathbf{B})$ attains the same objective value Golub & Van Loan (2013).

The truncated SVD is globally optimal; the balanced factorization is well-conditioned, and the optimal shared right subspace is $\mathrm{row}(\mathbf{V}_r^\top \boldsymbol{\Sigma}_\mathbf{x}^{-1/2})$.

LEMMA A.2.6 (SINGULAR $\boldsymbol{\Sigma}_\mathbf{x}$ AND PSEUDOINVERSE WHITENING).

In this subsection, we extend all results to rank-deficient $\boldsymbol{\Sigma}_\mathbf{x}$ by showing the objective depends only on $\mathrm{Range}(\boldsymbol{\Sigma}_\mathbf{x})$ and that pseudoinverse whitening preserves the conclusions on that subspace Ben-Israel & Greville (2010); Penrose (1955).

Let $\boldsymbol{\Sigma}_\mathbf{x} = \mathbf{Q}\boldsymbol{\Lambda}\mathbf{Q}^\top$ with $\boldsymbol{\Lambda} = \mathrm{diag}(\lambda_1, \ldots, \lambda_{r_+}, 0, \ldots, 0)$. Define

$$\boldsymbol{\Sigma}_\mathbf{x}^{1/2} = \mathbf{Q}\boldsymbol{\Lambda}^{1/2}\mathbf{Q}^\top, \qquad \boldsymbol{\Sigma}_\mathbf{x}^{-1/2} = \mathbf{Q}\boldsymbol{\Lambda}^{\dagger/2}\mathbf{Q}^\top,$$

where $\boldsymbol{\Lambda}^{\dagger/2}$ applies $\lambda_i^{-1/2}$ to $\lambda_i > 0$ and 0 otherwise. Then the objective $\|(\mathbf{E}_{\mathrm{cat}} - \mathbf{A}\mathbf{B})\boldsymbol{\Sigma}_\mathbf{x}^{1/2}\|_F^2$ depends only on $\mathrm{Range}(\boldsymbol{\Sigma}_\mathbf{x})$, and Theorems A.2.1–A.2.5 hold unchanged on that subspace.

**Proof.**  Let $\mathbf{Q} = [\mathbf{Q}_r\ \mathbf{Q}_0]$ with $\mathbf{Q}_r$ spanning $\mathrm{Range}(\boldsymbol{\Sigma}_\mathbf{x})$ and $\boldsymbol{\Sigma}_\mathbf{x}^{1/2} = \mathbf{Q}_r \boldsymbol{\Lambda}_r^{1/2} \mathbf{Q}_r^\top$. Then

$$\|(\mathbf{E} - \mathbf{A}\mathbf{B})\boldsymbol{\Sigma}_\mathbf{x}^{1/2}\|_F^2 = \|(\mathbf{E}\mathbf{Q}_r - \mathbf{A}(\mathbf{B}\mathbf{Q}_r))\boldsymbol{\Lambda}_r^{1/2}\|_F^2,$$

which is the same Frobenius objective restricted to $\mathrm{Range}(\boldsymbol{\Sigma}_\mathbf{x})$. Components along $\mathbf{Q}_0$ vanish under $\boldsymbol{\Sigma}_\mathbf{x}^{1/2}$ and contribute nothing. $\square$

Pseudoinverse whitening discards the nullspace; all conclusions hold on $\mathrm{Range}(\boldsymbol{\Sigma}_\mathbf{x})$.

In our implementation, to estimate and stabilize $\boldsymbol{\Sigma}_\mathbf{x}$, we perform ridge/shrinkage regularization ($\widehat{\boldsymbol{\Sigma}}_\mathbf{x} \leftarrow \widehat{\boldsymbol{\Sigma}}_\mathbf{x} + \varepsilon\mathbf{I}$) while using diagonal approximations (cf. Bishop (2006); Anderson (1984); Ledoit & Wolf (2004); Hoerl & Kennard (2000)) with mini-batch and sliding-window since computing full covariances are costly.

A.3   QR REDUCTION: SMALL-CORE EQUIVALENCE AND GLOBAL SOLUTION (PROOF)

The covariance-aligned objective

$$\min_{\mathbf{A}\in\mathbb{R}^{m\times r},\ \mathbf{B}\in\mathbb{R}^{r\times d}}\big\|\big(\mathbf{E}_{\mathrm{cat}}-\mathbf{A}\mathbf{B}\big)\mathbf{\Sigma}_{\mathbf{x}}^{1/2}\big\|_F^2 \tag{A.3.1}$$

can be solved without ever forming the tall whitened matrix $\widetilde{\mathbf{E}} := \mathbf{E}_{\mathrm{cat}}\mathbf{\Sigma}_{\mathbf{x}}^{1/2} \in \mathbb{R}^{m\times d}$. A thin QR $\mathbf{E}_{\mathrm{cat}} = \mathbf{Q}_e\mathbf{R}_e$ (with $\mathbf{Q}_e^\top\mathbf{Q}_e = \mathbf{I}_d$) collects all the information relevant to Eq. A.3.1 into the $d\times d$ core $\mathbf{M} := \mathbf{R}_e\mathbf{\Sigma}_{\mathbf{x}}^{1/2}$ because $\widetilde{\mathbf{E}} = \mathbf{Q}_e\mathbf{M}$ and the Frobenius norm is left-orthogonally invariant ($\|\mathbf{Q}\mathbf{Z}\|_F = \|\mathbf{Z}\|_F$ when $\mathbf{Q}^\top\mathbf{Q} = \mathbf{I}$) Golub & Van Loan (2013); Trefethen & Bau (1997). Thus we can reduce the large problem to an equivalent $d \times d$ problem, apply standard SVD/EYM analysis on the core, and lift the solution back (QR reduction to a core matrix; see also Halko et al. (2011); Martinsson & Tropp (2020) for randomized variants).

LEMMA A.3.1 (OPTIMAL $\boldsymbol{A}$ LIES IN $\mathrm{col}(\mathbf{Q}_e)$).

For any $\mathbf{A}$, decompose $\mathbf{A} = \mathbf{Q}_e\widehat{\mathbf{A}} + \mathbf{A}_\perp$ with $\mathbf{Q}_e^\top\mathbf{A}_\perp = \mathbf{0}$ and set $\widehat{\mathbf{B}} := \mathbf{B}\mathbf{\Sigma}_{\mathbf{x}}^{1/2}$. Then

$$\|\widetilde{\mathbf{E}} - \mathbf{A}\widehat{\mathbf{B}}\|_F^2 = \|\mathbf{Q}_e(\mathbf{M} - \widehat{\mathbf{A}}\widehat{\mathbf{B}})\|_F^2 + \|\mathbf{A}_\perp\widehat{\mathbf{B}}\|_F^2 \ \geq\ \|\mathbf{Q}_e(\mathbf{M} - \widehat{\mathbf{A}}\widehat{\mathbf{B}})\|_F^2,$$

where $\widetilde{\mathbf{E}} = \mathbf{Q}_e\mathbf{M}$ and $\mathbf{M} := \mathbf{R}_e\mathbf{\Sigma}_{\mathbf{x}}^{1/2}$. Hence any global minimizer satisfies $\mathbf{A}_\perp = \mathbf{0}$, i.e., $\mathbf{A}^\star = \mathbf{Q}_e\widehat{\mathbf{A}}^\star$. It shrinks the search space for $\mathbf{A}$ to the $d$-dimensional column space of $\mathbf{Q}_e$; any component orthogonal to $\mathrm{col}(\mathbf{Q}_e)$ only increases the loss. (Orthogonal decomposition/Pythagorean property of the Frobenius inner product; cf. Golub & Van Loan (2013); Trefethen & Bau (1997).)

**Proof.**   Use $\widetilde{\mathbf{E}} = \mathbf{Q}_e\mathbf{M}$ and orthogonality: $\mathbf{Q}_e^\top(\mathbf{Q}_e(\cdot)) = (\cdot)$ and $\mathbf{Q}_e^\top(\mathbf{A}_\perp\widehat{\mathbf{B}}) = \mathbf{0}$, so the two terms are orthogonal in the Frobenius inner product and the squared norm splits. The minimum occurs at $\mathbf{A}_\perp = \mathbf{0}$.   □

THEOREM A.3.2 (CORE EQUIVALENCE).

By Lemma A.3.1 and left-orthogonal invariance of $\|\cdot\|_F$ (i.e., $\|\mathbf{Q}\mathbf{Z}\|_F = \|\mathbf{Z}\|_F$ for orthogonal $\mathbf{Q}$; Golub & Van Loan (2013)),

$$\min_{\mathbf{A},\mathbf{B}}\big\|(\mathbf{E}_{\mathrm{cat}} - \mathbf{A}\mathbf{B})\mathbf{\Sigma}_{\mathbf{x}}^{1/2}\big\|_F^2 \ =\ \min_{\widehat{\mathbf{A}},\widehat{\mathbf{B}}}\|\mathbf{M} - \widehat{\mathbf{A}}\widehat{\mathbf{B}}\|_F^2, \qquad \mathbf{M} = \mathbf{R}_e\mathbf{\Sigma}_{\mathbf{x}}^{1/2}. \tag{A.3.2}$$

Any minimizer $(\widehat{\mathbf{A}}^\star, \widehat{\mathbf{B}}^\star)$ lifts to a minimizer of the original problem via

$$\mathbf{A}^\star = \mathbf{Q}_e\widehat{\mathbf{A}}^\star, \qquad \mathbf{B}^\star = \widehat{\mathbf{B}}^\star\mathbf{\Sigma}_{\mathbf{x}}^{-1/2}, \tag{A.3.3}$$

where $\mathbf{\Sigma}_{\mathbf{x}}^{-1/2}$ denotes a (pseudo-)inverse square root when $\mathbf{\Sigma}_{\mathbf{x}}$ is singular Ben-Israel & Greville (2010).

**Proof.**   Restrict to $\mathbf{A} = \mathbf{Q}_e\widehat{\mathbf{A}}$ (Eq. A.3.3). Then $\|(\mathbf{E}_{\mathrm{cat}} - \mathbf{A}\mathbf{B})\mathbf{\Sigma}_{\mathbf{x}}^{1/2}\|_F = \|\mathbf{Q}_e(\mathbf{M} - \widehat{\mathbf{A}}\widehat{\mathbf{B}})\|_F = \|\mathbf{M} - \widehat{\mathbf{A}}\widehat{\mathbf{B}}\|_F$. The lifting follows by inverting the change $\widehat{\mathbf{B}} = \mathbf{B}\mathbf{\Sigma}_{\mathbf{x}}^{1/2}$.   □

COROLLARY A.3.3 (PRESERVATION OF NONZERO SINGULAR VALUES AND RIGHT SINGULAR VECTORS).

Since $(\mathbf{Q}_e\mathbf{M})^\top(\mathbf{Q}_e\mathbf{M}) = \mathbf{M}^\top\mathbf{M}$, $\widetilde{\mathbf{E}} = \mathbf{Q}_e\mathbf{M}$ and $\mathbf{M}$ share the same nonzero singular values and the same right singular vectors. Hence the SVD of $\mathbf{M}$ directly yields the optimal shared right subspace for the covariance-aligned objective (orthogonal invariance of SVD; e.g., Golub & Van Loan (2013); Trefethen & Bau (1997)).

**Proof.**   Immediate from $\mathbf{Q}_e^\top\mathbf{Q}_e = \mathbf{I}_d$.   □

THEOREM A.3.4 (BALANCED FACTORS, GLOBAL MINIMIZER, AND LIFTING).

Let $\mathbf{M} = \mathbf{U\Sigma V}^\top$ be an SVD and $(\mathbf{U}_r, \mathbf{\Sigma}_r, \mathbf{V}_r)$ the top-$r$ blocks. Then

$$\widehat{\mathbf{A}}^\star = \mathbf{U}_r\mathbf{\Sigma}_r^{1/2}, \qquad \widehat{\mathbf{B}}^\star = \mathbf{\Sigma}_r^{1/2}\mathbf{V}_r^\top \tag{A.3.4}$$

achieve the global minimum of $\|\mathbf{M} - \widehat{\mathbf{A}}\widehat{\mathbf{B}}\|_F^2$ by the Eckart–Young–Mirsky theorem Eckart & Young (1936); MIRSKY (1960); Golub & Van Loan (2013). Lifting to the original variables gives

$$\mathbf{A}^\star = \mathbf{Q}_e\mathbf{U}_r\mathbf{\Sigma}_r^{1/2}, \qquad \mathbf{B}^\star = \mathbf{\Sigma}_r^{1/2}\mathbf{V}_r^\top\,\mathbf{\Sigma}_\mathbf{x}^{-1/2}. \tag{A.3.5}$$

The minimum value is $\|\mathbf{M} - \mathbf{U}_r\mathbf{\Sigma}_r\mathbf{V}_r^\top\|_F^2$, and the shared right subspace is $\mathrm{row}(\mathbf{B}^\star) = \mathrm{span}(\mathbf{V}_r^\top\mathbf{\Sigma}_\mathbf{x}^{-1/2})$ (cf. Ky Fan Fan (1950)).

It provides a closed-form global minimizer and a numerically well-conditioned (balanced) factorization.

Truncated SVD is optimal; balancing $(\mathbf{\Sigma}_r^{1/2})$ improves conditioning and scale regularity Golub & Van Loan (2013).

**Proof.** Apply EYM to the core problem from Eq. A.3.2; choose balanced factors so that $\mathbf{U}_r\mathbf{\Sigma}_r\mathbf{V}_r^\top = \widehat{\mathbf{A}}^\star\widehat{\mathbf{B}}^\star$. Use Eq. A.3.3 to obtain $(\mathbf{A}^\star, \mathbf{B}^\star)$. $\square$

This process makes the thin QR cost $\mathcal{O}(md^2)$, while forming/using the core costs $\mathcal{O}(d^3)$ (or $\mathcal{O}(d^2)$ if $\mathbf{\Sigma}_\mathbf{x}^{1/2}$ is precomputed/structured). All subsequent optimization is on the $d \times d$ core Golub & Van Loan (2013); Trefethen & Bau (1997). After computing $\mathbf{M}$, we do not materialize $\mathbf{M}$; instead we keep $\mathbf{z} \mapsto \mathbf{Mz} = \mathbf{R}_e(\mathbf{\Sigma}_\mathbf{x}^{1/2}\mathbf{z})$ and $\mathbf{y} \mapsto \mathbf{M}^\top\mathbf{y} = \mathbf{\Sigma}_\mathbf{x}^{1/2}(\mathbf{R}_e^\top\mathbf{y})$, and pass these to RSVD Halko et al. (2011); Martinsson & Tropp (2020).

From Eq. A.3.5, $\mathrm{row}(\mathbf{B}^\star) = \mathrm{span}(\mathbf{V}_r^\top\mathbf{\Sigma}_\mathbf{x}^{-1/2})$ defines the shared right subspace. In GLOWQ, this subspace is exactly the group-shared projection used to compute and cache $\mathbf{R} = \mathbf{B}_{\mathrm{shared}}\mathbf{X}$ once per input-sharing group, thereby enabling efficient $\mathbf{A}_i\mathbf{R}$ reuse during inference while preserving expressivity via module-specific $\mathbf{A}_i$ (cf. Sec. 3.3 and the Ky Fan view in Theorem A.1.3).

## A.4 RSVD ACCURACY GUARANTEES

Let the core matrix be $\mathbf{M} := \mathbf{R}_e\mathbf{\Sigma}_\mathbf{x}^{1/2} \in \mathbb{R}^{d \times d}$ as defined by the QR reduction in Appendix A.3. We target rank $r \leq d$ with oversampling $p \geq 2$ and power iterations $q \geq 0$. By the core equivalence and preservation results, accuracy on $\mathbf{M}$ transfers verbatim to the covariance-aligned objective.

ALGORITHM A.4.1 - RSVD ON THE CORE $\mathbf{M}$.

It computes the dominant right subspace (which defines the shared right factor) on the small $d \times d$ core without ever materializing the tall whitened matrix (standard RSVD; (Halko et al., 2011; Martinsson & Tropp, 2020)).

**Procedure.**

(i) $\mathbf{\Omega} \sim \mathcal{N}(0,1)^{d \times (r+p)}, \quad \mathbf{Y} \leftarrow \mathbf{M\Omega}$;

(ii) Power iterations: repeat $q$ times $\{\mathbf{Y} \leftarrow \mathbf{M}(\mathbf{M}^\top\mathbf{Y})\}$ with re-orthonormalization;

(iii) $\mathbf{Q} \leftarrow \mathrm{orth}(\mathbf{Y}), \quad \mathbf{B} \leftarrow \mathbf{Q}^\top\mathbf{M}$;

(iv) $\mathbf{B} = \widetilde{\mathbf{U}}\,\mathbf{\Sigma}\,\mathbf{V}^\top, \quad \mathbf{U} \leftarrow \mathbf{Q}\widetilde{\mathbf{U}}$; truncate to $(\mathbf{U}_r, \mathbf{\Sigma}_r, \mathbf{V}_r)$;

(v) Balanced core factors: $\widehat{\mathbf{A}}^\star = \mathbf{U}_r\mathbf{\Sigma}_r^{1/2}, \widehat{\mathbf{B}}^\star = \mathbf{\Sigma}_r^{1/2}\mathbf{V}_r^\top$.

Find a good range $\mathbf{Q}$ via randomized sketching (with optional power iterations), then refine by a small SVD on $\mathbf{Q}^\top\mathbf{M}$. *Justification.* Within the subspace $\mathcal{R}(\mathbf{Q})$, the best rank-$r$ approximation is the truncated SVD of $\mathbf{Q}^\top\mathbf{M}$; lifting by $\mathbf{Q}$ yields $\mathbf{U}_r\mathbf{\Sigma}_r\mathbf{V}_r^\top$ as the optimal restricted approximation (Golub & Van Loan, 2013, Ch. 2). The randomized sketch ensures (in expectation or with high probability) that $\mathcal{R}(\mathbf{Q})$ captures the dominant right subspace of $\mathbf{M}$ (Halko et al., 2011; Martinsson & Tropp, 2020). $\square$

THEOREM A.4.2 (FROBENIUS ERROR, EXPECTATION).

Let $\mathbf{M} = \mathbf{U}\boldsymbol{\Sigma}\mathbf{V}^\top$ with singular values $\sigma_1 \geq \cdots \geq \sigma_d$. For $p \geq 2$ and $q = 0$,

$$\mathbb{E}\, \|\mathbf{M} - \mathbf{Q}\mathbf{Q}^\top\mathbf{M}\|_F \;\leq\; \left(1 + \frac{r}{p-1}\right)^{1/2} \left(\sum_{j>r} \sigma_j^2\right)^{1/2}. \tag{A.4.1}$$

(Halko–Martinsson–Tropp; e.g., (Halko et al., 2011, Thm. 10.5))

It quantifies that RSVD matches the optimal tail energy up to a mild factor depending only on $(r, p)$.

**Proof.** Write $\mathbf{M} = \mathbf{U}\begin{bmatrix}\boldsymbol{\Sigma}_1 & \mathbf{0} \\ \mathbf{0} & \boldsymbol{\Sigma}_2\end{bmatrix}\mathbf{V}^\top$ with $\boldsymbol{\Sigma}_1 \in \mathbb{R}^{r\times r}$ and $\boldsymbol{\Sigma}_2$ the tail. Let $\mathbf{V}^\top\boldsymbol{\Omega} = \begin{bmatrix}\boldsymbol{\Omega}_1 \\ \boldsymbol{\Omega}_2\end{bmatrix}$ and $\mathbf{Y} = \mathbf{M}\boldsymbol{\Omega}$. Standard analysis of Gaussian sketches gives $\|(\mathbf{I} - \mathbf{P_Q})\mathbf{M}\|_F \leq \|\boldsymbol{\Sigma}_2\|_F \|\boldsymbol{\Omega}_2\boldsymbol{\Omega}_1^\dagger\|_F$, and $\mathbb{E}\|\boldsymbol{\Omega}_2\boldsymbol{\Omega}_1^\dagger\|_F^2 \leq r/(p-1)$ for $p \geq 2$ (Halko et al., 2011). Taking square roots and expectations yields Eq. A.4.1. $\qquad\square$

THEOREM A.4.3 (SPECTRAL ERROR WITH $q$ POWER ITERATIONS).

For $q \geq 0$ and a modest constant $\mathbf{C}_{r,p}$ (depending gently on $r, p$),

$$\|\mathbf{M} - \mathbf{U}_r\boldsymbol{\Sigma}_r\mathbf{V}_r^\top\|_2 \;\lesssim\; \mathbf{C}_{r,p}^{1/(2q+1)}\, \sigma_{r+1}. \tag{A.4.2}$$

(Cf. (Halko et al., 2011; Martinsson & Tropp, 2020; Musco & Musco, 2015).)

Power iterations shrink the subspace-angle gap geometrically toward the optimal $\sigma_{r+1}$ bound.

Each power iteration reduces the gap factor roughly by a $(\cdot)^{1/(2q+1)}$ exponent toward $\sigma_{r+1}$.

**Proof.** After $q$ power steps, $\mathbf{Y} = (\mathbf{M}\mathbf{M}^\top)^q\mathbf{M}\boldsymbol{\Omega} = \mathbf{U}\,\boldsymbol{\Sigma}^{2q+1}(\mathbf{V}^\top\boldsymbol{\Omega})$. Block-partitioning $\mathbf{V}^\top\boldsymbol{\Omega} = \begin{bmatrix}\boldsymbol{\Omega}_1 \\ \boldsymbol{\Omega}_2\end{bmatrix}$ and analyzing principal angles between the exact and sketched right subspaces gives

$$\|(\mathbf{I} - \mathbf{P_Q})\mathbf{M}\|_2 \;\leq\; \|\boldsymbol{\Sigma}_2\|_2 \left\|\boldsymbol{\Sigma}_2^{2q}\boldsymbol{\Omega}_2(\boldsymbol{\Omega}_1)^\dagger\boldsymbol{\Sigma}_1^{-2q}\right\|_2^{1/(2q+1)}.$$

Bounding the Gaussian pseudo-inverse term by $\mathbf{C}_{r,p}$ and using $\|\boldsymbol{\Sigma}_2^{2q}\boldsymbol{\Sigma}_1^{-2q}\|_2 = (\sigma_{r+1}/\sigma_r)^{2q}$ yields Eq. A.4.2. $\qquad\square$

COROLLARY A.4.4 (TRANSFER TO THE COVARIANCE-ALIGNED OBJECTIVE).

By Theorem A.3.2 and Corollary A.3.3.

$$\left\|(\mathbf{E}_{\mathrm{cat}} - \mathbf{A}^\star\mathbf{B}^\star)\boldsymbol{\Sigma}_{\mathbf{x}}^{1/2}\right\|_F \;=\; \|\mathbf{M} - \mathbf{U}_r\boldsymbol{\Sigma}_r\mathbf{V}_r^\top\|_F. \tag{A.4.3}$$

It links RSVD accuracy on the core directly to the original covariance-aligned objective.

Core RSVD error bounds become the error bounds for the original problem, verbatim.

**Proof.** We have $\widetilde{\mathbf{E}} = \mathbf{E}_{\mathrm{cat}}\boldsymbol{\Sigma}_{\mathbf{x}}^{1/2} = \mathbf{Q}_e\mathbf{M}$ and Frobenius norms are left-orthogonally invariant; the optimal truncated approximation on $\mathbf{M}$ corresponds under lifting to the optimal approximation of $(\mathbf{E}_{\mathrm{cat}} - \mathbf{A}\mathbf{B})\boldsymbol{\Sigma}_{\mathbf{x}}^{1/2}$, yielding Eq. A.4.3. $\qquad\square$

PROPOSITION (Q-LESS LIFTING: BLOCKWISE RECOVERY OF $\mathbf{A}_i^\star$).

Write $\mathbf{E}_{\mathrm{cat}} = [\mathbf{E}_1; \ldots; \mathbf{E}_m]$ and $\mathbf{A}^\star = [\mathbf{A}_1^\star; \ldots; \mathbf{A}_m^\star]$ conformably. At fixed $\mathbf{B}^\star$, each block admits the closed form

$$\mathbf{A}_i^\star \;=\; \mathbf{E}_i(\mathbf{B}^\star)^\top\big(\mathbf{B}^\star(\mathbf{B}^\star)^\top\big)^\dagger, \qquad i = 1, \ldots, m, \tag{A.4.4}$$

so the tall orthonormal factor $\mathbf{Q}_e$ need not be stored (least-squares with pseudoinverse; cf. (Björck, 1996; Ben-Israel & Greville, 2010; Golub & Van Loan, 2013)).

It economizes memory: per-block factors are recovered directly from $(\mathbf{E}_i, \mathbf{B}^\star)$ without retaining $\mathbf{Q}_e$.

Per-block least-squares with a pseudoinverse yields $\mathbf{A}_i^\star$ using only $(\mathbf{E}_i, \mathbf{B}^\star)$.

**Proof.** For each block, minimize $\|\mathbf{E}_i - \mathbf{A}_i \mathbf{B}^\star\|_F^2$. The first-order optimality condition is $\mathbf{A}_i^\star \mathbf{B}^\star (\mathbf{B}^\star)^\top = \mathbf{E}_i (\mathbf{B}^\star)^\top$. Multiplying on the right by the Moore–Penrose pseudoinverse gives the minimal-norm solution $\mathbf{A}_i^\star = \mathbf{E}_i (\mathbf{B}^\star)^\top \big(\mathbf{B}^\star (\mathbf{B}^\star)^\top\big)^\dagger$, which is precisely Eq. A.4.4. $\qquad\square$

## B EFFECT OF RIGHT-WEIGHTED SHARED B

In this section, we analyze the effect of the right-weighted shared-B on the GlowQ's error correction with the following procedure.

**Procedure.** (1) Using calibration inputs $\{\mathbf{x}_n\}_{n=1}^N \subset \mathbb{R}^d$, estimate the layer input covariance

$$\widehat{\boldsymbol{\Sigma}}_\mathbf{x} = \tfrac{1}{N} \sum_n \mathbf{x}_n \mathbf{x}_n^\top \quad (\text{optionally: } \widehat{\boldsymbol{\Sigma}}_\mathbf{x} \leftarrow \widehat{\boldsymbol{\Sigma}}_\mathbf{x} + \varepsilon\,\mathbf{I}).$$

(2) For each module $i \in \{\mathrm{q}, \mathrm{k}, \mathrm{v}, \mathrm{gate}, \mathrm{up}\}$, form the quantization-error matrix $\mathbf{E}_i \in \mathbb{R}^{O_i \times d}$ and the row-stack $\mathbf{E}_{\mathrm{cat}} = [\mathbf{E}_1; \ldots; \mathbf{E}_m] \in \mathbb{R}^{m \times d}$.

(3) **Cov-aligned (whitened):** compute SVDs of $\widetilde{\mathbf{E}}_i := \mathbf{E}_i \widehat{\boldsymbol{\Sigma}}_\mathbf{x}^{1/2}$ and $\widetilde{\mathbf{E}}_{\mathrm{cat}} := \mathbf{E}_{\mathrm{cat}} \widehat{\boldsymbol{\Sigma}}_\mathbf{x}^{1/2}$, and take the top-$r$ right bases $\mathbf{V}_{i,r}$ and $\mathbf{V}_r$.

(4) **Unweighted (no-cov):** repeat the same without whitening to obtain $\mathbf{V}_{i,r}^{(\text{no-cov})}$ and $\mathbf{V}_r^{(\text{no-cov})}$.

(5) For each module, form the absolute cross-basis cosine matrix

$$\mathbf{C}_i = \big|\mathbf{V}_r^\top \mathbf{V}_{i,r}\big| \in \mathbb{R}^{r \times r},$$

Hungarian-reorder it to maximize the diagonal sum, and visualize as heatmaps.

**Impact of optimization with the right weighted objective.** As illustrated in Fig. 5 and 6, the whitened condition produces a bright near-diagonal across all groups (Q/K/V and MLP gate/up), indicating a one-to-one alignment between the shared right subspace $\mathrm{row}(B_{\mathrm{shared}})$ and each module's right subspace (up to sign/permutation). The effect is strongest for Q/K, and slightly more diffuse for V and for MLP (gate/up), but remains concentrated on the leading axes. In contrast, the unweighted condition yields noise-like patterns with no diagonal structure.

Right-side covariance weighting is crucial for estimating a shared $B$ under anisotropic inputs: it exposes a common right subspace across modules that ingest the same input tensor. This validates the shared-$B$ assumption and directly motivates our ABx caching strategy, i.e., computing $R = B_{\mathrm{shared}} X$ once per group and reusing $A_i R$ across modules. Unweighted stacked SVD fails to reveal this alignment, weakening both the shared-$B$ premise and the practical caching benefit.

### B.1 GROUP-CACHED (WEIGHTED STACKED RSVD) VS. LAYER-WISE (WEIGHTED RSVD)

Table 7: Perplexity (lower is better) across model families on WikiText-2. Layer-wise applies layer-wise SVD correction, whereas GLOWQ applies group-wise SVD with a shared right factor $B$; GLOWQ (Selective restore) denotes selective group restoration.

| Method | LLaMA 2 | | LLaMA 3 | | Qwen 2.5 | | Qwen 3 | | OPT | | Vicuna | | Mistral |
|---|---|---|---|---|---|---|---|---|---|---|---|---|---|
| | 7B | 13B | 3.2-3B | 3.1-8B | 7B | 14B | 8B | 14B | 1.3B | 6.7B | 7B | 13B | 7B |
| LAYERWISE | 5.58 | 4.96 | 8.15 | 6.59 | 7.06 | 5.64 | 9.92 | 8.80 | 15.05 | 11.00 | 6.89 | 6.03 | 5.42 |
| GlowQ | 5.58 | 4.96 | 8.16 | 6.59 | 7.07 | 5.64 | 9.90 | 8.80 | 15.06 | 11.00 | 6.90 | 6.02 | 5.42 |
| GlowQ-S | 5.60 | 4.96 | 8.22 | 6.62 | 7.09 | 5.68 | 9.97 | 8.89 | 15.19 | 11.00 | 6.90 | 6.04 | 5.45 |

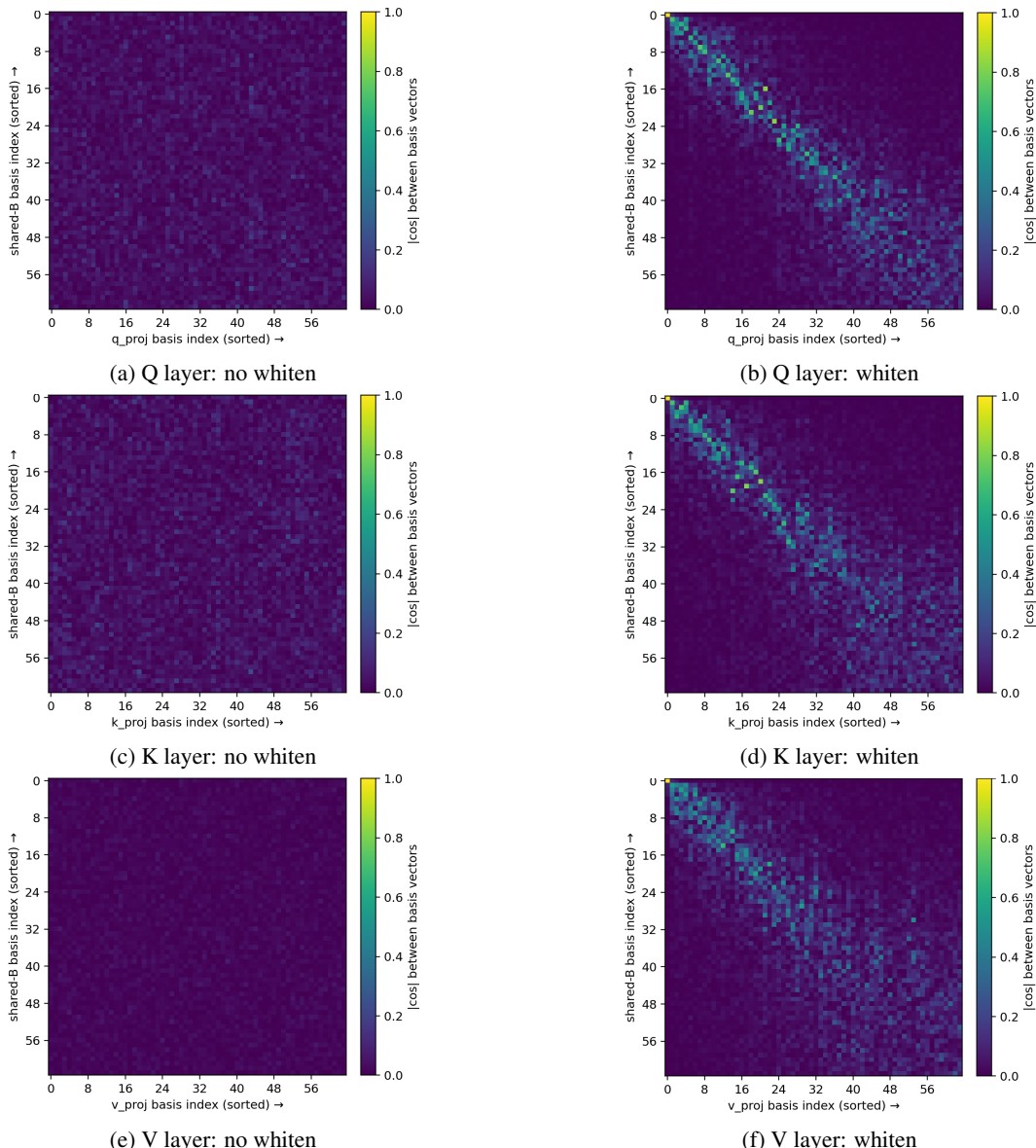

Figure 5: Whitening vs. non-whitening alignment matrices. For LLaMA 3.2-3B, we estimate a shared right basis $B_{\text{shared}}$ from the stacked error either without covariance weighting ($E_{\text{cat}}$, left panels) or with covariance-aware whitening ($E_{\text{cat}} \Sigma_x^{1/2}$, right panels). Each heatmap shows the absolute basis alignment between $\text{row}(B_{\text{shared}})$ and the per-module right subspace for Q, K, V; brighter values denote larger absolute inner products. DiagScore and Affinity summaries are reported in the main text.

**Results on Table 7.** Across 13 model-size combinations, GlowQ and Layer-wise yield essentially identical perplexity: the mean gap is +0.001 ppl on average, with per-family fluctuations confined to ±0.02 ppl. By design, GlowQ-S (Selective restore) trades a bit of accuracy for efficiency, trailing Layer-wise by +0.04 ppl on average. In short, the full shared-**B** configuration matches layer-wise $(\mathbf{A}_i, \mathbf{B}_i)$ on WikiText-2 without systematic degradation, while the selective variant incurs a small, consistent increase in ppl.

**Observation on Fig. 5, 6.** The covariance-aligned cross-basis heatmaps exhibit an almost perfectly diagonal structure after Hungarian matching, indicating a near one-to-one correspondence between the shared right subspace and each module's top-$r$ directions. Whitening aligns input usage so that the shared **B** spans (practically) the same right-singular space that the individual $\mathbf{B}_i$

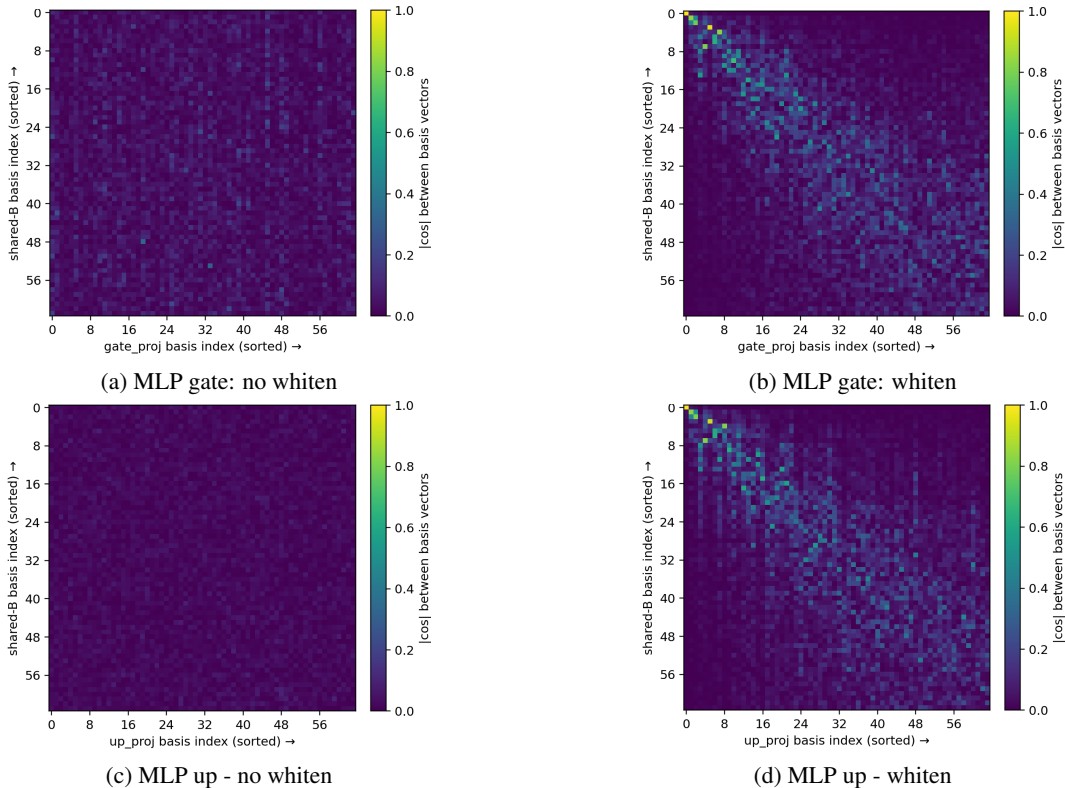

(a) MLP gate: no whiten

(b) MLP gate: whiten

(c) MLP up - no whiten

(d) MLP up - whiten

Figure 6: Whitening vs. non-whitening alignment matrices. MLP (up/gate).

would select, explaining why GlowQ's perplexity tracks layerwise so closely, and why GlowQ-S, restoring only a subset, shows the small upward shift in ppl.

**Observation on Fig. 7, 8, 9.** For the MoE FFN of Qwen1.5-MoE-A2.7B, the covariance-aligned cross-basis heatmaps show the same qualitative behavior as in the dense models once whitening is enabled. Without whitening, all panels (expert gate/up, shared gate/up, and MoE attention) look almost uniformly dark, indicating that the shared right subspace and each expert's local top-$r$ directions are essentially uncorrelated. After whitening and Hungarian matching, the heatmaps become sharply diagonal for both the representative expert (e.g., `expert59_gate_proj` / `expert59_up_proj`) and the shared-$B$ MLP/attention blocks, revealing a near one-to-one alignment between the shared basis and each expert's own error subspace. This confirms that, once inputs are whitened, the grouped MoE FFNs and the shared MLP effectively live in the same right-singular space, so a single shared $\mathbf{B}_{\text{shared}}$ can serve all experts with only small residual mismatch. Consequently, GlowQ can compress all experts and the shared MLP with one shared right-hand matrix while closely tracking the layerwise baseline in perplexity, explaining the tiny +0.02 PPL gap we observe on Qwen1.5-MoE-A2.7B.

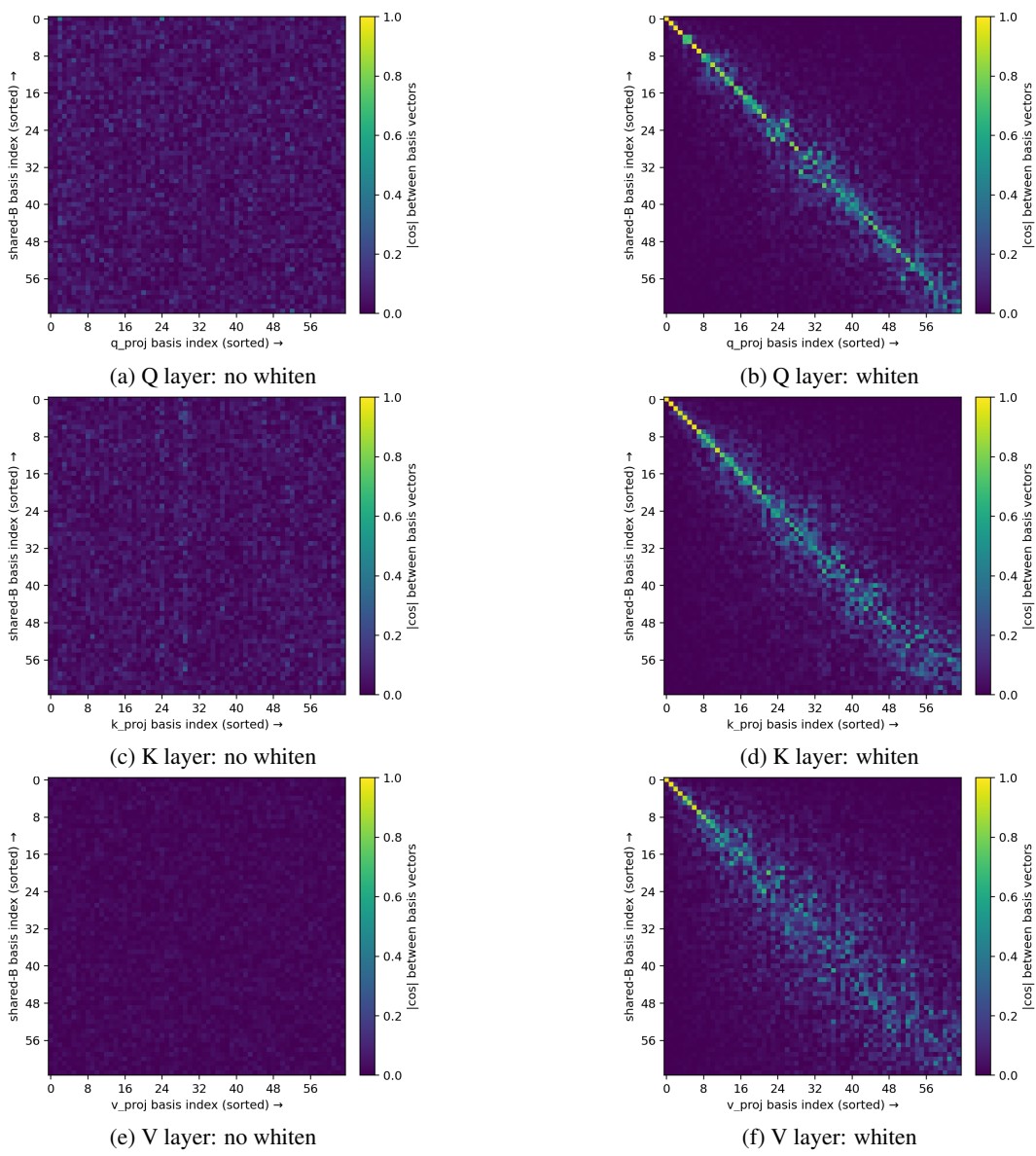

(a) Q layer: no whiten

(b) Q layer: whiten

(c) K layer: no whiten

(d) K layer: whiten

(e) V layer: no whiten

(f) V layer: whiten

Figure 7: Whitening vs. non-whitening alignment matrices for Q/K/V in Qwen1.5-MoE-A2.7B. As in Fig. 5, we estimate a shared right basis $B_{\text{shared}}$ from the stacked attention-projection error, either from the raw error ("no whiten", left panels) or after covariance-aware whitening $E_{\text{cat}}\Sigma_x^{1/2}$ ("whiten", right panels). Each heatmap shows the absolute basis alignment between $\text{row}(B_{\text{shared}})$ and the per-module right subspace for Q, K, and V; brighter values denote larger absolute inner products. Whitening again yields a sharply diagonally dominant structure, indicating that a single covariance-aligned basis captures the dominant error directions across Q/K/V.

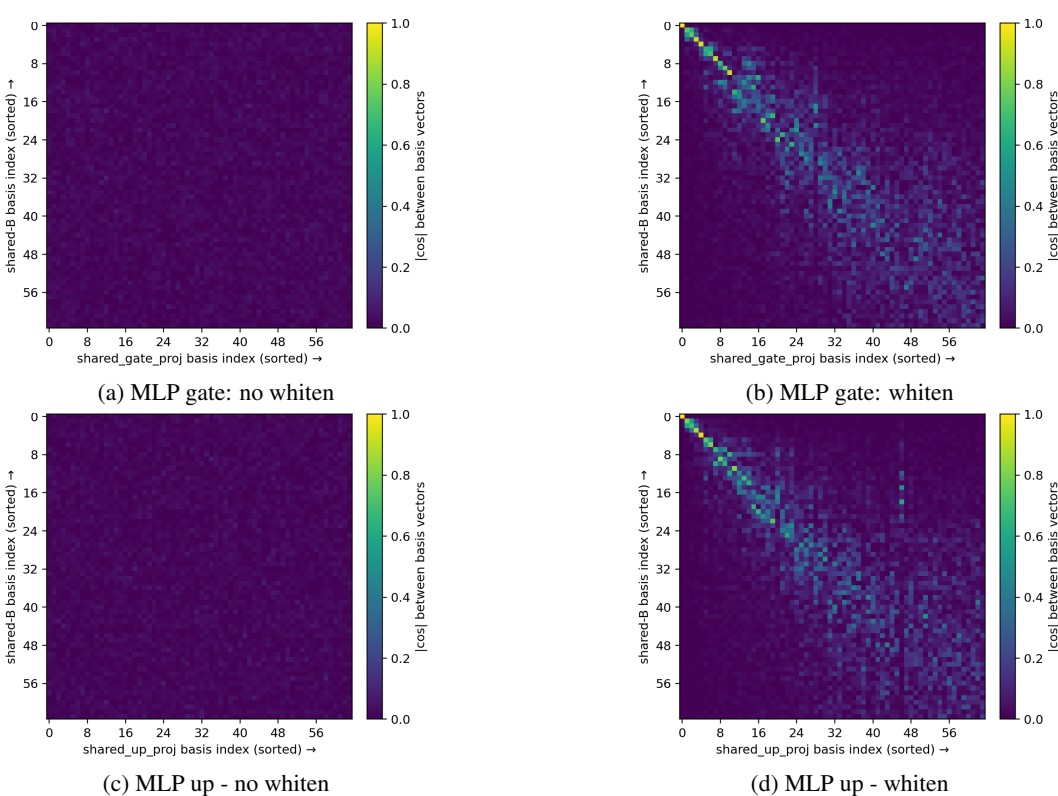

(a) MLP gate: no whiten

(b) MLP gate: whiten

(c) MLP up - no whiten

(d) MLP up - whiten

Figure 8: Whitening vs. non-whitening alignment matrices for MLP (gate and up) in Qwen1.5-MoE-A2.7B. The construction is identical to Fig. 7, but applied to the MLP gate and up projections aggregated over all experts. Whitening produces a diagonally dominant alignment, indicating that a shared covariance-aligned basis also captures the principal error directions of the MLP blocks.

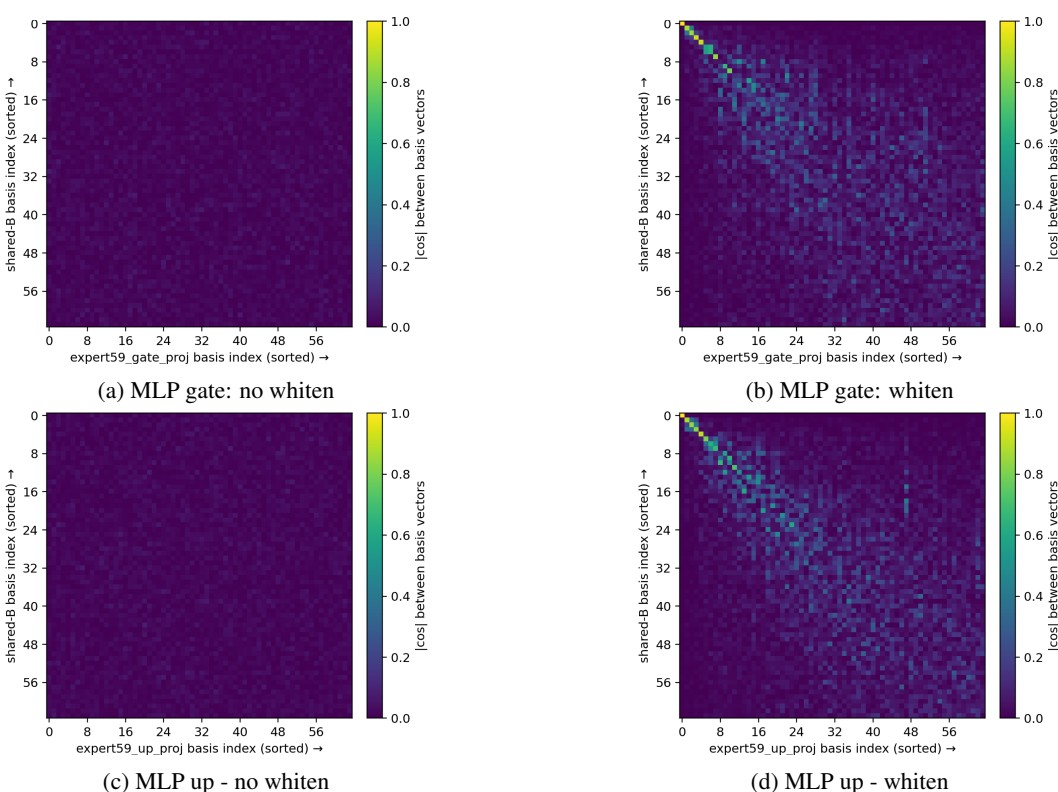

(a) MLP gate: no whiten

(b) MLP gate: whiten

(c) MLP up - no whiten

(d) MLP up - whiten

Figure 9: Whitening vs. non-whitening alignment matrices for the MLP (gate and up) of a single expert (Expert 59) in Qwen1.5-MoE-A2.7B. We apply the same construction as in Fig.8, but restrict the stacked error and shared basis $B_{\text{shared}}$ to Expert 59 only. The diagonally dominant structure under whitening shows that the covariance-aligned basis remains meaningful even at the per-expert level.

## C   TTFB & THROUGHPUT AROUND OTHER MODELS

Table 8: Latency comparison on LLaMA 3 models for Layerwise vs. GlowQ, GlowQ-S.

| Models | | Setting | TTFB ↓ (ms) | tok/s ↑ | Prefill ↓ (ms) | Dec ↓ (ms/tok) |
|---|---|---|---|---|---|---|
| LLaMA 3 | 3.2-3B | Layerwise | 70.83 | 17.46 | 71.07 | 58.22 |
| | | GlowQ | 64.92 | 18.94 | 66.20 | 52.04 |
| | | GlowQ-S | 53.17 | 21.37 | 60.69 | 44.35 |
| | 3.1-8B | Layerwise | 96.50 | 14.24 | 95.72 | 69.26 |
| | | GlowQ | 86.44 | 15.31 | 90.01 | 64.47 |
| | | GlowQ-S | 71.70 | 18.89 | 73.50 | 52.34 |
| | *Avg. Δ BX (%)* | | **-9.38** | **+8.00** | **-6.41** | **-8.77** |
| | *Avg. Δ R50 (%)* | | **-25.32** | **+27.52** | **-18.91** | **-24.13** |

**Results on Table 8.**   Table 8 mirrors the LLaMA 2 evaluation under an identical runtime and measurement protocol. Two consistent trends emerge: (i) GlowQ reduces all latency components, with the largest relative gains on per-token decode; and (ii) GlowQ-S further amplifies these benefits. On LLaMA 3 (3.2-3B, 3.1-8B), GlowQ with BX caching improves serving latency over Layerwise: TTFB $-9.38\%$, tok/s $+8.00\%$, Prefill $-6.41\%$, and Dec $-8.77\%$ on average. GlowQ-S (selective restore) amplifies these gains: TTFB $-25.32\%$, tok/s $+27.52\%$, Prefill $-18.91\%$, and Dec $-24.13\%$ on average. Improvements are consistent across both model sizes, with the largest reductions appearing in the per-token Dec phase and end-to-end TTFB, reflecting reduced compute on the critical path. In practice, BX caching provides drop-in speedups without modifying weights, while the selective policy (GlowQ-S) offers a simple accuracy-latency knob by reducing the number of $\mathbf{A}_i\mathbf{R}$ applications (Sec. 3.3).

**Observation on Table 8.**   BX caching removes redundant right-projection work by reusing the shared subspace, so each decode step primarily executes lightweight $\mathbf{A}_i\mathbf{R}$ updates; this directly lowers Dec and TTFB. The selective-restore strategy further trims the executed paths across decoder blocks, yielding additional latency drops with a commensurate increase in throughput (tok/s). These mechanisms explain the near-linear percentage gains in the RSVD-driven core cost: caching reduces repeated right-side multiplies, while selective restoration shortens the active compute graph along the decoding trajectory.

## D   HYPERPARAMETER CHANGE

### D.1   CALIBRATION DIFFERENCE

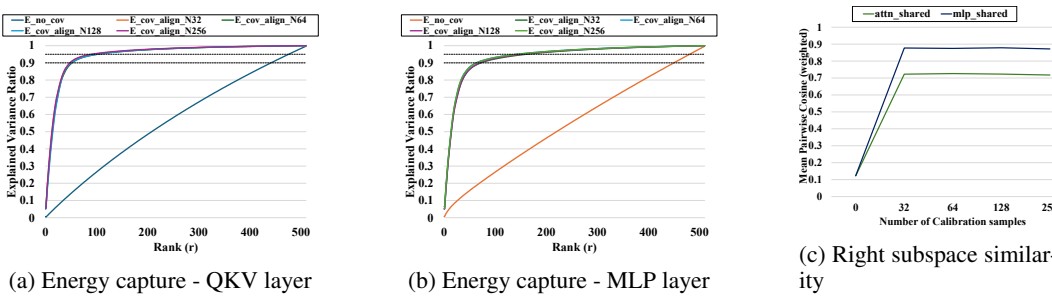

(a) Energy capture - QKV layer     (b) Energy capture - MLP layer     (c) Right subspace similarity

Figure 10: Energy Capture and Cosine similarity of Rightspace over number of calibration samples

Fig. 10a, 10b plot energy-capture curves versus rank for different numbers of calibration samples, while Fig. 10c reports the mean pairwise cosine similarity (weighted) between the shared right subspace and the per-layer right subspaces as the calibration size varies.

Varying the number of calibration samples $N \in \{32, 64, 128, 256\}$ leaves the energy-capture curves in Fig. 10a, 10b nearly indistinguishable, especially for practical ranks $r \leq 128$. In Fig. 10c, the weighted cosine similarity between the shared right subspace and layer-wise right subspaces is already high at $N = 32$ and saturates for $N \geq 64$. These results indicate that a small calibration set suffices to recover a stable, data-aligned right subspace, consistent with PCA stability under a clear spectral gap (Jolliffe & Cadima, 2016; Horn & Johnson, 1985).

We attribute the observed stability under a relatively small calibration set, e.g., $N = 32$ to the following four reasons: (i) *Spectral-gap effect:* The input covariance $\boldsymbol{\Sigma}_{\mathbf{x}}$ is heavy-tailed, so the top directions are separated by a clear eigenvalue gap; the dominant $r$-dimensional right subspace stabilizes quickly with modest $N$ (Jolliffe & Cadima, 2016; Horn & Johnson, 1985). (ii) *Robust weighted objective.* We optimize a right-weighted criterion,

$$\min_{\mathbf{A}, \mathbf{B}} \left\| \left( \mathbf{E}_{\text{cat}} - \mathbf{AB} \right) \boldsymbol{\Sigma}_{\mathbf{x}}^{1/2} \right\|_F^2,$$

so small perturbations in the estimate $\widehat{\boldsymbol{\Sigma}}_{\mathbf{x}}$ have limited effect: large-eigenvalue axes dominate and lead to the same top $r$-subspace (see also weighted low-rank formulations (Srebro & Jaakkola, 2003)). *Numerical regularization.* Shrinkage/normalization of $\widehat{\boldsymbol{\Sigma}}_{\mathbf{x}}$ reduces small-sample noise and improves conditioning (Ledoit & Wolf, 2004; Hoerl & Kennard, 2000; Bishop, 2006). *Benefit of group stacking.* Building the SVD core from vertically stacked errors increases the effective sample support along rows, which smooths estimation of the shared right subspace (Paige & Saunders, 1981; Golub & Van Loan, 2013).

To conclude, calibration sizes as small as $N \approx 32$–$64$ already place the system in a saturated mode since energy capture at a fixed $r$ and the similarity between the shared and layer-wise right subspaces change only marginally beyond this point. Thus, our covariance-aligned, group-shared $\mathbf{B}$ achieves stable performance with low calibration cost.

### D.1.1 SHRINK ALPHA DIFFERENCE

Table 9: Perplexity on WikiText-2 while sweeping calibration samples and shrink $\alpha$ (lower is better).

| Calibration Samples | Shrink $\alpha$ | LLaMA 3 | | Qwen 3 | |
|---|---|---|---|---|---|
| | | 3.2-3B | 8B | 3.1-8B | 14B |
| | 0 | 8.16 | 6.59 | 9.89 | 8.82 |
| 32 | 0.02 | 8.16 | 6.59 | 9.86 | 8.82 |
| | 0.05 | 8.16 | 6.59 | 9.88 | 8.82 |
| | 0 | 8.15 | 6.59 | 9.90 | 8.81 |
| 64 | 0.02 | 8.15 | 6.59 | 9.88 | 8.78 |
| | 0.05 | 8.16 | 6.59 | 9.87 | 8.79 |
| | 0 | 8.16 | 6.59 | 9.92 | 8.80 |
| 128 | 0.02 | 8.16 | 6.58 | 9.91 | 8.80 |
| | 0.05 | 8.15 | 6.58 | 9.90 | 8.81 |
| | 0 | 8.16 | 6.58 | 9.93 | 8.81 |
| 256 | 0.02 | 8.15 | 6.59 | 9.92 | 8.80 |
| | 0.05 | 8.16 | 6.58 | 9.92 | 8.82 |

We apply a standard covariance shrinkage when forming the input statistic used for covariance-aligned subspace estimation. Let $\widehat{\boldsymbol{\Sigma}}_{\mathbf{x}}$ be the sample covariance from $N$ calibration sequences and $d$ the input dimension. We construct

$$\widehat{\boldsymbol{\Sigma}}_{\mathbf{x}}^{(\alpha)} = (1 - \alpha) \, \widehat{\boldsymbol{\Sigma}}_{\mathbf{x}} + \alpha \, \frac{\text{tr}(\widehat{\boldsymbol{\Sigma}}_{\mathbf{x}})}{d} \, \mathbf{I}, \qquad \alpha \in [0, 1],$$

i.e., a convex combination of the sample covariance and an isotropic target (scaled identity); small $\alpha$ reduces small-sample noise and improves conditioning without altering the dominant axes learned from data (Ledoit & Wolf, 2004; Bishop, 2006; Anderson, 1984).

**Results on Table 9.** Across calibration sizes $N \in \{32, 64, 128, 256\}$ and shrink $\alpha \in \{0, 0.02, 0.05\}$, perplexity remains essentially flat for LLaMA 3: for 3.2-3B and 8B, the sweep changes values by +0.01 ppl on average. Qwen 3 shows the same qualitative behavior, with a mild benefit from shrinkage: $\alpha \in [0.02, 0.05]$ yields -0.02 ppl on average for 3.1-8B and -0.01 ppl on average for 14B (relative to $\alpha$=0 at the same $N$). Aggregating all models, $\alpha$=0.02 improves by -0.01 ppl average, and increasing $N$ beyond 64 produces only marginal changes ($\leq$ +0.01-+0.02 ppl on average depending on the family). In short, both the calibration size and a small shrink factor have only second-order effect on WikiText-2 perplexity, consistent with the stability suggested by the energy and cosine-similarity panels (Jolliffe & Cadima, 2016).

**Observation on Table 9.** The right subspace stabilizes quickly because (i) the input covariance exhibits a pronounced spectral gap, so the dominant $r$-dimensional space is identified with few samples (Jolliffe & Cadima, 2016; Horn & Johnson, 1985); (ii) the right-weighted objective emphasizes large-variance directions, making the solution insensitive to small perturbations in $\widehat{\Sigma}_\mathbf{x}$; (iii) mild shrinkage damps small-sample noise (Ledoit & Wolf, 2004; Bishop, 2006); and (iv) stacking modules to form the core increases effective sample support along rows (Paige & Saunders, 1981; Golub & Van Loan, 2013). Consequently, small calibration sets ($N \approx 32$–64) already recover a data-aligned shared right subspace, explaining the near-constant perplexity across the sweep and the slight, consistent gains from $\alpha \in [0.02, 0.05]$ on Qwen 3.

### D.1.2 MEMORY USAGE

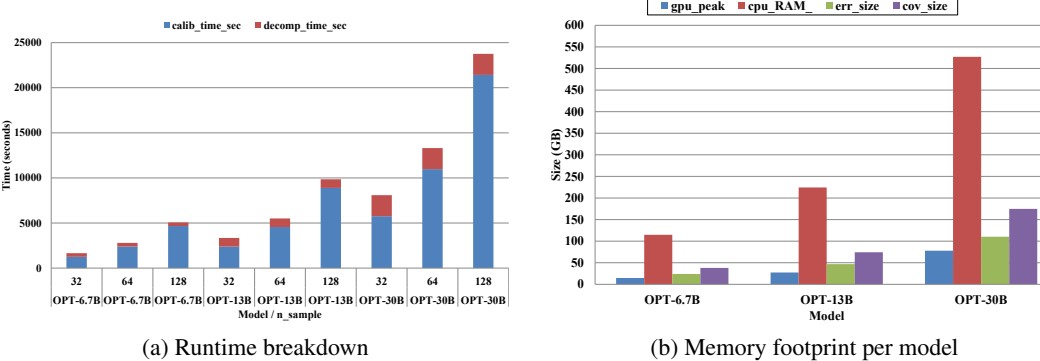

(a) Runtime breakdown        (b) Memory footprint per model

Figure 11: Calibration runtime and memory footprint as a function of model size and the number of calibration samples $N$. (a) Stacked bars show the runtime breakdown into calibration and decomposition for each (model, $N$) configuration; calibration dominates the total cost and grows nearly linearly with $N$, while decomposition time remains almost constant. (b) Memory footprint of the error tensor, covariance tensor, and peak GPU/CPU usage for each OPT model; the error and covariance tensors account for most of the memory and grow steeply with model size.

**Results on Fig. 11.** We profile calibration on a single A100 80GB GPU for three OPT models (6.7B, 13B, 30B) and calibration sizes $N \in \{32, 64, 128\}$, using SlimPajama-6B as the calibration corpus. Fig. 11a shows that the total wall-clock time is dominated by the calibration pass: for every model, the blue bars (forward passes used to estimate $\widehat{\Sigma}_\mathbf{x}$ and collect error tensors) account for most of the runtime and grow almost linearly with $N$, whereas the red bars (randomized GSVD / decomposition) contribute a relatively small and nearly constant overhead. Even for OPT-30B, increasing $N$ from 32 to 128 scales the runtime by roughly the same factor, indicating that the cost is predictable and controlled by the choice of calibration size. Fig. 11b breaks down the memory footprint. Peak GPU memory (blue) grows moderately with model size and remains well below the CPU footprint, since we keep the model and activations on GPU but store error and covariance tensors on host memory. The green and purple bars show that these two tensors dominate the CPU

usage and scale with model size: moving from OPT-6.7B to OPT-30B increases both `err_size` and `cov_size` by several times, and the peak CPU RAM closely tracks their sum.

**Observation on Fig. 11.** Overall, the results indicate that the main cost of our method comes from a one-time, embarrassingly parallel calibration phase whose runtime scales linearly with $N$ and roughly with model size, while the decomposition step has almost fixed cost. Memory-wise, the GPU footprint is modest and does not require larger-than-standard accelerators; the heavy objects are the error and covariance tensors on CPU, which can be streamed, sharded, or discarded immediately after decomposition. Since Sections D.1–D.1.1 show that small calibration sets ($N \approx 32$–$64$) already yield stable energy capture, right-subspace similarity, and perplexity, practitioners can operate in this low-$N$ regime. In practice, this keeps the calibration overhead to a few GPU hours even for 30B models and confines the CPU memory requirement to a one-off offline preprocessing step, directly addressing concerns about prohibitive calibration time and memory pressure for large LLMs.

## D.2 RANK DIFFERENCE

Table 10: Perplexity on WikiText-2 by rank and method, formatted like the calibration-sweep table.(Lower is better.)

| Rank | Method | LLaMA 3 | | Qwen 3 | |
|---|---|---|---|---|---|
| | | 3.2 3B | 3.1 8B | 8B | 14B |
| 8 | GlowQ | 8.22 | 6.64 | 9.95 | 8.84 |
| | Layerwise | 8.22 | 6.64 | 9.96 | 8.84 |
| 16 | GlowQ | 8.20 | 6.63 | 9.95 | 8.81 |
| | Layerwise | 8.20 | 6.62 | 9.94 | 8.80 |
| 32 | GlowQ | 8.18 | 6.61 | 9.91 | 8.81 |
| | Layerwise | 8.18 | 6.61 | 9.93 | 8.80 |
| 64 | GlowQ | 8.16 | 6.59 | 9.87 | 8.80 |
| | Layerwise | 8.15 | 6.58 | 9.88 | 8.80 |
| 128 | GlowQ | 8.12 | 6.56 | 9.83 | 8.79 |
| | Layerwise | 8.11 | 6.55 | 9.87 | 8.79 |

**Results on Table 10.** Sweeping the rank $r$, GlowQ matches layer-wise restoration in perplexity: the gap is +0.02 ppl average across models and ranks (never exceeding +0.04 ppl). Returns diminish beyond moderate ranks: from $r{=}8$ to $r{=}128$, the change is -0.09 ppl average across families. Most of the gain is realized by $r \in \{32, 64\}$; increases beyond this window yield only marginal improvements (e.g., $r{=}64 \rightarrow 128$ shifts by just a few hundredths of a ppl).

**Observation on Table 10.** The rank-accuracy curve exhibits family-specific shapes: LLaMA shows a knee around $r \approx 32$–$64$ (initially flat, then a brief drop), whereas Qwen decreases more gradually without a sharp elbow. In practice, this suggests using $r{=}64$ for LLaMA and $r{=}32$ for Qwen as strong defaults; GlowQ remains interchangeable with layer-wise restoration in accuracy at fixed $r$, while retaining the runtime advantages established elsewhere.

## D.3 RANDOMIZED SVD PARAMETERS

### D.3.1 PROOF OF QR REDUCTION & RANDOMIZED SVD

**Discussion.** Table 11 shows that Exact SVD on the $d \times d$ core **M** takes 42.86 s in total (0.76 s per layer on average), whereas Randomized SVD (RSVD) completes in 5.16–5.22 s (0.09 s per layer). This $\approx 8.2$–$8.3\times$ wall-clock speedup is consistent with the complexity gap between $\mathcal{O}(d^3)$ and $\mathcal{O}\big((q{+}1) d^2 (r{+}p) + d(r{+}p)^2\big)$ when $d \gg r{+}p$ (Golub & Van Loan, 2013; Halko et al.,

Table 11: SVD runtime (s) and perplexity on LLaMA 3.2-3B (WikiText-2). Exact = torch.linalg.svd on the GSVD core $M$; Randomized = Halko R-SVD with oversampling $p$ and power iterations $q$. *SVD-only* times factorization on $M$ (CUDA-synced), excluding the core QR used to build $M$. Total sums over layers; Layer(mean) averages across layers.

| Method | $q$ | $p$ | SVD time (s) $\downarrow$ | | Perplexity $\downarrow$ |
|---|---|---|---|---|---|
| | | | Total | Layer(mean) | |
| Exact SVD | – | – | 42.86 | 0.76 | 8.16 |
| Randomized SVD | 0 | 0 | 5.16 | 0.09 | 8.22 |
| | | 4 | 5.19 | 0.09 | 8.21 |
| | | 8 | 5.20 | 0.09 | 8.21 |
| | | 16 | 5.21 | 0.09 | 8.21 |
| | | 24 | 5.21 | 0.09 | 8.21 |
| Randomized SVD | 1 | 0 | 5.17 | 0.09 | 8.17 |
| | | 4 | 5.19 | 0.09 | 8.16 |
| | | 8 | 5.20 | 0.09 | 8.16 |
| | | 16 | 5.21 | 0.09 | 8.16 |
| | | 24 | 5.21 | 0.09 | 8.16 |
| Randomized SVD | 2 | 0 | 5.17 | 0.09 | 8.16 |
| | | 4 | 5.20 | 0.09 | 8.16 |
| | | 8 | 5.20 | 0.09 | 8.15 |
| | | 16 | 5.21 | 0.09 | 8.16 |
| | | 24 | 5.22 | 0.09 | 8.16 |

2011; Martinsson & Tropp, 2020). Concretely, with $d=3072$, $r=64$, and $p \in \{0,\dots,24\}$, we have $(r+p)/d \leq 88/3072 \approx 2.9\%$, so the RSVD term $(q+1)\,d^2(r+p)$ scales roughly like a few percent of $d^3$ up to constant factors, matching the observed order-of-magnitude reduction in runtime.

**Effect of $q$ and $p$.** Runtime varies only weakly across $p \in \{0, 4, 8, 16, 24\}$ and $q \in \{0, 1, 2\}$ ($5.16\,\mathrm{s} \to 5.22\,\mathrm{s}$). This is expected because the dominant RSVD cost is the matrix-block multiplies $\mathbf{M}\Omega$, $\mathbf{M}^\top(\cdot)$; increasing $p$ from 0 to 24 changes $(r+p)$ from 64 to 88 (only $\sim 38\%$), and the extra $q$ passes add a small multiple of the same GEMM cost. The lower-order term $d(r+p)^2$ is negligible at this scale. In short, the linear dependence on $(r+p)$ and on $(q+1)$ predicted by

$$\mathcal{O}\big((q+1)\,d^2(r+p) + d(r+p)^2\big)$$

manifests as a near-flat runtime curve because $d \gg r+p$ and GEMM kernels saturate the device (Halko et al., 2011; Martinsson & Tropp, 2020).

**Accuracy.** Perplexity stays essentially unchanged: Exact = 8.16; RSVD is 8.22 at $(q=0, p=0)$ and improves to 8.15–8.16 for $q \geq 1$ (with small $p$ already sufficient). This aligns with randomized SVD theory: even a single power iteration ($q=1$) sharpens separation between leading and trailing singular directions and yields a right subspace that is effectively indistinguishable (for a rank-$r$ objective) from Exact SVD in downstream perplexity (Halko et al., 2011; Musco & Musco, 2015; Martinsson & Tropp, 2020).

The empirical results agree with the stated complexity: Exact SVD on $\mathbf{M}$ incurs $\mathcal{O}(d^3)$ time, while RSVD retrieves the leading right subspace in $\mathcal{O}\big((q+1)\,d^2(r+p)\big)$ time (plus a minor $d(r+p)^2$ term) (Golub & Van Loan, 2013; Halko et al., 2011; Martinsson & Tropp, 2020). In practice, $q=1$ with a modest $p$ (e.g., $p \in [4, 16]$) delivers near-Exact perplexity at $\sim 8\times$ lower wall time, and increasing $p$ further yields diminishing returns (Halko et al., 2011; Martinsson & Tropp, 2020).

D.3.2 POWER ITERATION & OVERSAMPLING DIFFERENCE

Table 12: Randomized SVD hyperparameters on WikiText-2, measured on LLaMA-3.2-3B. We sweep (a) oversampling $p$ *(fixed $q = 2$)* and (b) power iterations $q$ *(fixed $p = 16$)* and report perplexity (lower is better).

(a) Oversampling $p$ sweep *(fixed $q = 2$)*.

| Method | $p$ | PPL $\downarrow$ |
|---|---|---|
| LLaMA 3.2-3B | 10 | 8.16 |
| | 12 | 8.16 |
| | 16 | 8.16 |
| | 24 | 8.16 |
| LLaMA 3.1-8B | 10 | 6.59 |
| | 12 | 6.59 |
| | 16 | 6.59 |
| | 24 | 6.58 |
| Qwen 3-8B | 10 | 9.90 |
| | 12 | 9.89 |
| | 16 | 9.88 |
| | 24 | 9.89 |
| Qwen 3-14B | 10 | 8.81 |
| | 12 | 8.80 |
| | 16 | 8.81 |
| | 24 | 8.81 |

(b) Power iterations $q$ sweep *(fixed $p = 16$)*.

| Method | $q$ | PPL $\downarrow$ |
|---|---|---|
| Llama 3.2-3B | 0 | 8.21 |
| | 1 | 8.16 |
| | 2 | 8.16 |
| Llama 3.1-8B | 0 | 6.63 |
| | 1 | 6.59 |
| | 2 | 6.59 |
| Qwen 3-8B | 0 | 9.97 |
| | 1 | 9.87 |
| | 2 | 9.88 |
| Qwen 3-14B | 0 | 8.79 |
| | 1 | 8.81 |
| | 2 | 8.81 |

Table 12 contrasts oversampling $p$ (with $q$=2 fixed; subtable 12a) and power iterations $q$ (with $p$=16 fixed; subtable 12b). Empirically, increasing $p$ from 10 to 24 leaves PPL essentially unchanged across models (differences of $\leq 0.01$), whereas raising $q$ from 0 to 1 yields small but consistent gains (most visibly on QWEN3–8B), after which improvements saturate by $q$=2.

This pattern aligns with the standard analysis of randomized SVD (RSVD). Oversampling enlarges the sketch dimension to $\ell = r + p$, which reduces the probability of missing near-rank-$r$ directions but ultimately does not change the target truncation rank $r$. Once $r$ already captures the dominant subspace and the spectral gap is reasonable, the marginal benefit of additional $p$ is small; theory predicts only a mild reduction of the residual as $p$ grows (e.g., with expected error bounds that degrade roughly as $\sqrt{r/(p-1)}$), so practical guidance typically recommends $p \approx$ 5–10 (Halko et al., 2011; Martinsson & Tropp, 2020).

By contrast, $q$ directly amplifies spectral separation via the power scheme. Forming $\mathbf{Y} = (\mathbf{A}\mathbf{A}^\top)^q \mathbf{A}\Omega$ effectively reweights singular values as $\sigma_i^{2q+1}$, which boosts the ratio between $\sigma_r$ and the tail $\{\sigma_{j>r}\}$ and thereby reduces leakage beyond rank $r$. As a result, the sampled subspace aligns better with the true top-$r$ subspace, often yielding noticeable gains from $q$=0 to $q$=1, with diminishing returns thereafter; $q \in \{1, 2\}$ is commonly recommended in practice (Halko et al., 2011; Ma & Ma, 2024; Martinsson & Tropp, 2020).

When $r$ already captures the dominant energy, increasing $p$ beyond a modest buffer offers little accuracy benefit, while a single power iteration ($q$=1) can materially improve approximation for matrices with slowly decaying spectra. In our experiments, this theoretical expectation manifests as flat PPL curves across $p$ and consistent but saturating improvements across $q$.

E COMPATIBILITY ACROSS QUANTIZATION DATATYPES

We apply weight-only quantization to Mistral-7B and evaluate on the WikiText-2 test set across both integer and floating-point-like datatypes (Table 13). For the integer settings (INT2/INT3/INT4), we use uniform weight-only quantization with shared scales within each weight group. For the floating-point-like settings (MXFP4, MXFP6, NVFP4), we adopt microscaling-style formats in

Table 13: WikiText-2 test perplexity (↓) for different datatypes.

| Method | FP16 | INT | | | Floating-point-like | | |
|---|---|---|---|---|---|---|---|
| | | INT2 | INT3 | INT4 | MXFP4 | MXFP6 | NVFP4 |
| Quant only | 5.32 | 1015.39 | 6.16 | 5.51 | 8.05 | 5.36 | 6.09 |
| Quant + GlowQ | | 24.23 | 5.84 | 5.41 | 6.10 | 5.32 | 5.63 |

which weights are first normalized within a small block and then encoded using low-bit floating-point codes. Concretely, MXFP4 and MXFP6 follow the block-wise microscaling design of MX+ and the OCP MX specification, using a shared scale per block and 4-bit or 6-bit element codes, respectively Lee et al. (2025); Open Compute Project (2023). NVFP4 follows NVIDIA's reference design with a microscaled FP4 representation for weights, as described in their low-precision inference guidelines Alvarez et al. (2025). These configurations allow us to test GlowQ not only on conventional integer quantization, but also on recent microscaling-based floating-point-like formats.

Layering GlowQ on top of the quant-only baselines reduces perplexity by -991.16 on INT2, -0.32 on INT3, -0.10 on INT4, -1.95 on MXFP4, -0.04 on MXFP6, and -0.46 on NVFP4, relative to the corresponding quant-only settings. Improvements hold across all six evaluated datatypes, indicating that GlowQ behaves as an orthogonal, plug-and-play low-rank correction rather than a mechanism tied to a single integer format or precision; in particular, it remains compatible with recent floating-point-like microscaling formats while providing consistent accuracy gains.

# F  LONGBENCH RESULTS

Table 14: The results of Llama-3.1-8B-Instruct on LongBench. The model is evaluated on the 15 English subsets using the official LongBench evaluation protocol, with up to 4K input tokens as context.

| Method | NarrativeQA | Qasper | MultiFieldQA | HotpotQA | MuSiQue | 2WikiMQA | GovReport | QMSum |
|---|---|---|---|---|---|---|---|---|
| Baseline | 18.26 | 12.01 | 25.96 | 13.76 | 7.87 | 14.95 | 32.79 | 21.43 |
| W4A4+GlowQ | 14.68 | 10.80 | 24.95 | 14.21 | 8.39 | 14.20 | 32.01 | 22.01 |
| W4A8+GlowQ | 15.56 | 11.77 | 23.71 | 14.39 | 8.41 | 14.92 | 32.00 | 21.19 |
| W4A16+GlowQ | 15.46 | 11.82 | 23.68 | 14.39 | 7.77 | 14.53 | 32.32 | 21.20 |

| | MultiNews | LCC | RepoBench-P | TriviaQA | SAMSum | TRec | PR | Avg |
|---|---|---|---|---|---|---|---|---|
| Baseline | 26.95 | 51.93 | 47.00 | 87.76 | 44.72 | 70.00 | 37.50 | 34.19 |
| W4A4+GlowQ | 26.43 | 47.50 | 37.51 | 85.54 | 42.05 | 69.00 | 36.36 | 32.38 |
| W4A8+GlowQ | 27.03 | 51.50 | 35.97 | 84.10 | 42.62 | 68.50 | 37.08 | 32.58 |
| W4A16+GlowQ | 26.86 | 50.46 | 35.59 | 84.30 | 42.67 | 68.50 | 37.17 | 32.45 |

Table 14, 15 shows that across both 4K and 8K context settings on the English LongBench benchmark (Bai et al., 2023b), applying W4 weight quantization with GlowQ (W4A4/8/16+GlowQ) leads to only small differences from the original LLaMA-3.1-8B-Instruct on the 15 English LongBench tasks. On most tasks, the scores remain within a few points of the baseline, and the relative difficulty and ranking among tasks are largely preserved. This indicates that, even under aggressive quantization of both weights and activations, the low-rank correction in GlowQ keeps the overall performance stable.

When we extend the context length from 4K to 8K, both the baseline and the GlowQ models improve their average scores by a similar margin. In other words, in scenarios that benefit from longer context, the GlowQ models track the same performance trends as the full-precision model, without a collapse in reasoning ability in the long-context regime. Overall, GlowQ enables 4-bit quantization while preserving LLaMA-3.1-8B-Instruct's performance not only in standard contexts but also in long-context settings.

Table 15: The results of Llama-3.1-8B-Instruct on LongBench. The model is evaluated on the 15 English subsets using the official LongBench evaluation protocol, with up to 8K input tokens as context.

| Method | NarrativeQA | Qasper | MultiFieldQA | HotpotQA | MuSiQue | 2WikiMQA | GovReport | QMSum |
|---|---|---|---|---|---|---|---|---|
| Baseline | 23.50 | 13.54 | 27.87 | 16.83 | 10.94 | 16.44 | 34.27 | 22.87 |
| W4A4+GlowQ | 23.45 | 12.20 | 27.41 | 15.34 | 9.21 | 16.15 | 33.87 | 22.78 |
| W4A8+GlowQ | 25.38 | 12.61 | 25.71 | 15.37 | 9.93 | 15.30 | 34.07 | 22.67 |
| W4A16+GlowQ | 25.36 | 12.61 | 25.62 | 15.13 | 9.82 | 15.20 | 34.00 | 22.59 |

| | MultiNews | LCC | RepoBench-P | TriviaQA | SAMSum | TRec | PR | Avg |
|---|---|---|---|---|---|---|---|---|
| Baseline | 26.87 | 52.81 | 48.04 | 90.77 | 43.94 | 71.00 | 73.13 | 38.19 |
| W4A4+GlowQ | 26.39 | 48.73 | 38.83 | 88.78 | 42.43 | 70.50 | 70.52 | 36.44 |
| W4A8+GlowQ | 27.14 | 52.06 | 38.55 | 88.49 | 43.60 | 71.00 | 72.73 | 36.97 |
| W4A16+GlowQ | 26.96 | 51.12 | 38.84 | 88.67 | 43.44 | 71.00 | 73.50 | 36.92 |

## G  SELECTIVE RESTORATION ACROSS MODEL FAMILY

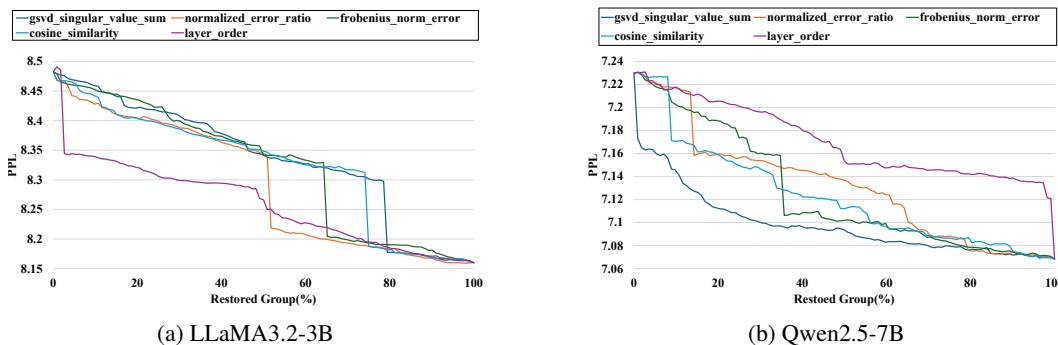

(a) LLaMA3.2-3B  (b) Qwen2.5-7B

Figure 12: Perplexity versus fraction of restored groups for different restoration metrics. For each metric, we sort the groups according to its score (GSVD singular-value sum, normalized error ratio, Frobenius-norm error, cosine similarity, or simple layer order), progressively restore groups back to full precision, and record the resulting perplexity.

**Importance metric selection.** The performance of GlowQ-S depends on the policy used to rank groups for restoration. In Fig. 12, we evaluate five saliency metrics from quantization and pruning literature. These include: (1) gsvd singular value sum, our $g_{ec}$ score (Eq. 9), which measures the captured error "energy" ($\|A\|_F^2$) in the low-rank factors and follows the standard practice of using singular-value energy to summarize PCA components (Jolliffe & Cadima, 2016; Halko et al., 2011); (2) normalized error ratio, our $g_{ner}$ score ( Eq. 10), a widely used PTQ-style proxy based on relative weight error $\|E_g\|_F/\|W_g\|_F$ (Nagel et al., 2021; Gholami et al., 2021; Krishnamoorthi, 2018); (3) frobenius norm error, the absolute error $\|E_g\|_F$ (Nagel et al., 2021; Pouransari et al., 2020; Zhao et al., 2025); (4) cosine similarity, measuring angular deviation between pre- and post-quantization weights or activations, which has been shown to be a strong pruning/quantization proxy (Mason-Williams & Dahlqvist, 2024; Chang et al., 2023); and (5) layer order as a simple baseline. The results show that gsvd singular value sum and normalized error ratio are consistently the most effective, yielding the steepest perplexity reduction. However, as noted in Sec. 3.3, no single metric is universally optimal. Therefore, our final policy (Sec. 4.6) pragmatically evaluates both $g_{ec}$ and $g_{ner}$ for a given model and selects the one that performs best, providing a robust, data-driven approach.

**Results on Fig. 13.** Across the four panels, LLaMA models exhibit a clear knee: perplexity drops steeply once a relatively small fraction of groups is restored, then plateaus. In contrast, Qwen and OPT show a gradual, near-linear descent as the restored fraction increases. The two evaluation curves in each subplot (ppl_gsvd vs. ppl_NER) track each other closely, differing mainly in the sharpness of the early descent.

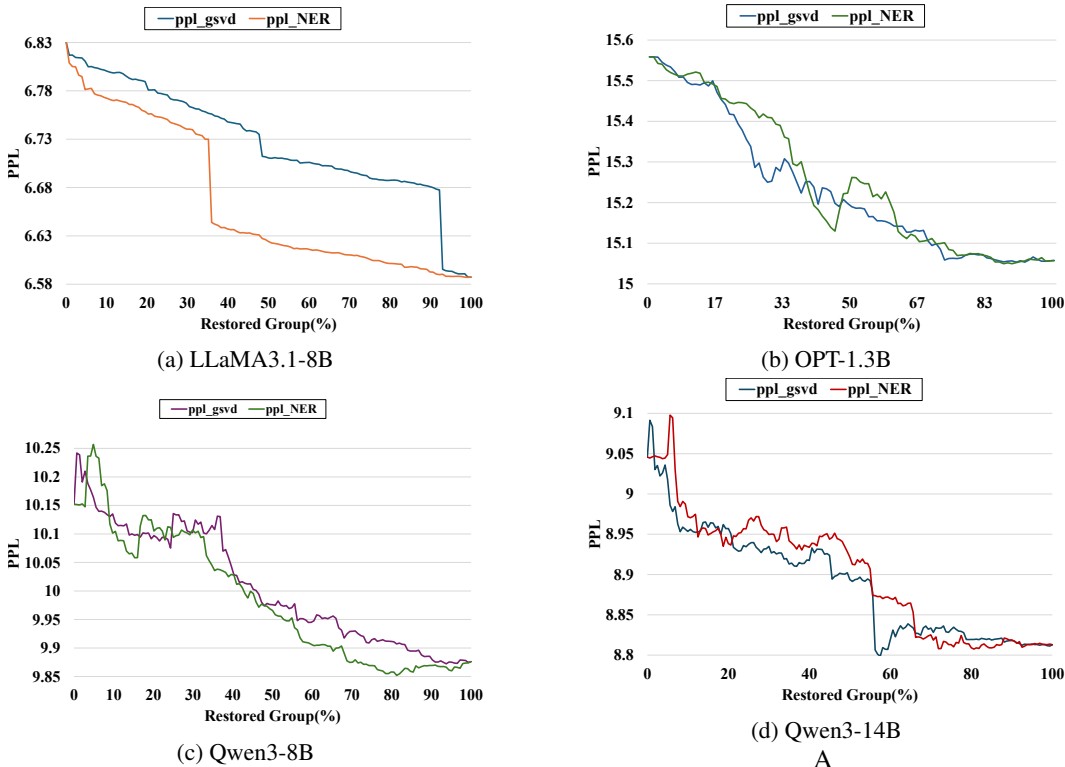

Figure 13: Perplexity as a function of restored group percentage for dif- ferent model families (LLaMA 3.1-8B, Qwen 3-8B, Qwen 3-14B, OPT-1.3B). We compare GSVD- based restoration (ppl gsvd) against NER-based restoration (ppl NER).

These curves suggest selecting the error-recovery metric per model family: outlier/energy ranking with small budgets for knee-shaped profiles, and Hessian-/loss-weighted ranking with broader budgets for diffuse profiles. This family-aware policy aligns with known outlier, anisotropy, and curvature phenomena in modern LLMs.

Table 16: Zero-shot results on LLaMA 3.2-3B.

| Method | Rank | PIQA | ARC-C | ARC-E | HellaS | WinoG | BoolQ | LAMBADA | C4 | AVG |
|---|---|---|---|---|---|---|---|---|---|---|
| | | Acc ↑ | Acc ↑ | Acc ↑ | Acc-norm ↑ | Acc ↑ | Acc ↑ | Acc ↑ | word PPL ↓ | Acc ↑ |
| FP16 | - | 72.33 | 39.33 | 72.33 | 63.67 | 70.33 | 77.00 | 71.00 | 10.30 | 67.14 |
| ZeroQuant-V2 | | 75.33 | 39.33 | 73.33 | 60.67 | 68.67 | 74.67 | 65.67 | 11.45 | 65.38 |
| QERA | | 76.67 | 38.67 | 72.33 | 61.67 | 68.67 | 72.33 | 64.67 | 11.04 | 65.48 |
| L2QER | 64 | 75.33 | 40.00 | 71.67 | 64.00 | 68.33 | 73.33 | 68.33 | 11.04 | 66.19 |
| GlowQ | | 77.67 | 39.67 | 72.00 | 64.00 | 70.33 | 74.33 | 70.33 | 10.98 | 66.90 |
| GlowQ-S | | 77.33 | 39.67 | 71.67 | 64.00 | 69.67 | 71.67 | 70.33 | 11.07 | 66.33 |

Table 17: Zero-shot results on LLaMA 3.1-8B.

| Method | Rank | PIQA | ARC-C | ARC-E | HellaS | WinoG | BoolQ | LAMBADA | C4 | AVG |
|---|---|---|---|---|---|---|---|---|---|---|
| | | Acc ↑ | Acc ↑ | Acc ↑ | Acc-norm ↑ | Acc ↑ | Acc ↑ | Acc ↑ | word PPL ↓ | Acc ↑ |
| FP16 | - | 78.67 | 51.67 | 80.67 | 67.67 | 74.67 | 80.67 | 79.00 | 9.00 | 73.29 |
| ZeroQuant-V2 | | 78.00 | 51.33 | 81.67 | 68.67 | 76.00 | 84.33 | 74.33 | 9.87 | 73.48 |
| QERA | | 77.00 | 51.33 | 80.33 | 69.00 | 74.33 | 82.67 | 75.33 | 9.68 | 72.86 |
| L2QER | 64 | 79.67 | 49.33 | 80.67 | 66.67 | 74.33 | 80.33 | 76.00 | 9.63 | 72.43 |
| GlowQ | | 79.67 | 51.00 | 81.33 | 66.00 | 74.33 | 82.00 | 79.00 | 9.59 | 73.33 |
| GlowQ-S | | 79.00 | 50.33 | 81.67 | 66.33 | 72.00 | 82.00 | 77.00 | 9.78 | 72.62 |

Table 18: Zero-shot results on Qwen 3-8B.

| Method | Rank | PIQA | ARC-C | ARC-E | HellaS | WinoG | BoolQ | LAMBADA | C4 | AVG |
|---|---|---|---|---|---|---|---|---|---|---|
| | | Acc ↑ | Acc ↑ | Acc ↑ | Acc-norm ↑ | Acc ↑ | Acc ↑ | Acc ↑ | word PPL ↓ | Acc ↑ |
| FP16 | - | 77.33 | 53.00 | 83.00 | 63.67 | 68.67 | 87.00 | 67.67 | 14.52 | 71.48 |
| ZeroQuant-V2 | | 75.67 | 52.33 | 80.33 | 63.00 | 71.00 | 85.33 | 63.67 | 15.00 | 70.19 |
| QERA | | 76.33 | 51.33 | 79.00 | 62.33 | 69.67 | 85.67 | 64.67 | 14.78 | 69.86 |
| L2QER | 64 | 75.67 | 51.33 | 79.33 | 62.67 | 67.67 | 85.33 | 64.67 | 14.82 | 69.52 |
| GlowQ | | 76.67 | 52.33 | 80.33 | 64.67 | 71.00 | 86.33 | 63.67 | 14.60 | 70.71 |
| GlowQ-S | | 76.33 | 50.67 | 80.67 | 63.33 | 70.67 | 85.00 | 65.33 | 14.77 | 70.29 |

Table 19: Zero-shot results on Qwen 3-14B.

| Method | Rank | PIQA | ARC-C | ARC-E | HellaS | WinoG | BoolQ | LAMBADA | C4 | AVG |
|---|---|---|---|---|---|---|---|---|---|---|
| | | Acc ↑ | Acc ↑ | Acc ↑ | Acc-norm ↑ | Acc ↑ | Acc ↑ | Acc ↑ | word PPL ↓ | Acc ↑ |
| FP16 | - | 78.33 | 59.33 | 80.33 | 66.67 | 75.67 | 92.00 | 66.33 | 13.08 | 74.10 |
| ZeroQuant-V2 | | 78.33 | 59.33 | 78.00 | 65.67 | 73.00 | 92.00 | 62.00 | 13.79 | 72.62 |
| QERA | | 76.98 | 57.67 | 79.33 | 67.00 | 74.00 | 92.00 | 65.00 | 13.29 | 73.14 |
| L2QER | 64 | 78.33 | 56.33 | 79.67 | 66.67 | 75.33 | 91.67 | 64.67 | 13.80 | 73.24 |
| GlowQ | | 77.67 | 56.67 | 80.00 | 68.87 | 75.67 | 91.33 | 66.67 | 13.26 | 73.84 |
| GlowQ-S | | 77.67 | 57.00 | 79.33 | 67.67 | 74.33 | 91.33 | 65.33 | 13.48 | 73.24 |

Table 20: Zero-shot results on Vicuna-7B.

| Method | Rank | PIQA | ARC-C | ARC-E | HellaS | WinoG | BoolQ | LAMBADA | C4 | AVG |
|---|---|---|---|---|---|---|---|---|---|---|
| | | Acc ↑ | Acc ↑ | Acc ↑ | Acc-norm ↑ | Acc ↑ | Acc ↑ | Acc ↑ | word PPL ↓ | Acc ↑ |
| FP16 | - | 76.00 | 41.33 | 70.33 | 66.00 | 68.00 | 80.33 | 72.33 | 8.70 | 67.76 |
| ZeroQuant-V2 | | 75.67 | 43.00 | 71.00 | 65.00 | 65.33 | 80.00 | 68.33 | 9.07 | 66.90 |
| QERA | | 76.00 | 42.67 | 70.67 | 67.00 | 67.33 | 81.00 | 69.67 | 8.91 | 67.76 |
| L2QER | 64 | 76.33 | 42.33 | 70.00 | 66.00 | 67.00 | 80.67 | 68.00 | 8.93 | 67.19 |
| GlowQ | | 75.67 | 41.67 | 70.00 | 66.67 | 67.00 | 80.33 | 69.67 | 8.87 | 67.29 |
| GlowQ-S | | 76.00 | 43.67 | 69.33 | 66.00 | 66.67 | 82.00 | 70.00 | 8.99 | 67.67 |

Table 21: Zero-shot results on Vicuna-13B.

| Method | Rank | PIQA | ARC-C | ARC-E | HellaS | WinoG | BoolQ | LAMBADA | C4 | AVG |
|---|---|---|---|---|---|---|---|---|---|---|
| | | Acc ↑ | Acc ↑ | Acc ↑ | Acc-norm ↑ | Acc ↑ | Acc ↑ | Acc ↑ | word PPL ↓ | Acc ↑ |
| FP16 | - | 77.33 | 49.33 | 73.67 | 67.33 | 74.33 | 86.00 | 74.33 | 7.76 | 71.76 |
| ZeroQuant-V2 | | 77.67 | 46.67 | 76.00 | 66.33 | 75.00 | 86.00 | 74.00 | 7.86 | 71.67 |
| QERA | | 78.00 | 47.33 | 76.33 | 67.33 | 75.33 | 86.00 | 73.00 | 7.88 | 71.90 |
| L2QER | 64 | 77.67 | 48.67 | 75.67 | 67.00 | 74.00 | 84.67 | 74.00 | 7.79 | 71.67 |
| GlowQ | | 78.33 | 47.67 | 75.67 | 67.33 | 74.67 | 85.33 | 74.00 | 7.85 | 71.86 |
| GlowQ-S | | 77.67 | 57.00 | 79.33 | 67.67 | 74.33 | 91.33 | 65.33 | 7.86 | 73.24 |

## LLM USAGE DISCLOSURE

### WRITING POLISH

After completing the full draft, we used a large language model (LLM) purely to aid proofreading and light copy-editing. Specifically, the LLM suggested fixes for grammar, spelling, punctuation, typographical errors, and minor wording for clarity and consistency.

### RETRIEVAL AND DISCOVERY

We also used an LLM as a literature discovery assistant to broaden our search beyond papers we had already identified. The LLM helped generate alternative keywords and surface potentially relevant works.

