# OpenReview forum: "GlowQ: Group-Shared LOw-Rank Approximation for Quantized LLMs"
_ICLR.cc/2026/Conference — ICLR 2026 Poster_

### Official Review · Reviewer_HaNH · 2025-10-29

**Soundness:** 3
**Presentation:** 4
**Contribution:** 2
**Rating:** 4
**Confidence:** 4

**Summary:**

This paper proposes GlowQ, a group-shared low-rank correction method for quantized LLMs that improves both accuracy and efficiency by sharing a single right-factor matrix across input-sharing modules (e.g., Q/K/V), caching the projection to avoid redundant computation, and aligning the correction subspace with data usage via covariance-weighted optimization. Combined with a selective restoration strategy (GlowQ-S), it reduces TTFB by up to 23.4% and increases throughput by 37.4% with minimal accuracy loss, outperforming existing baselines in perplexity and downstream task performance while remaining compatible with standard PTQ pipelines.

**Strengths:**

1. Efficient Group-Shared Correction via Caching: GlowQ reduces redundant computation by sharing a single low-rank right factor across input-sharing modules (e.g., Q/K/V projections), enabling one-time computation and reuse of the projection R=BsharedXR=Bshared​X, which significantly lowers computational overhead during inference.
2. Data-Aware Subspace Alignment: By incorporating input covariance into the low-rank approximation objective, GlowQ aligns the correction subspace with directions that are most frequently activated in practice, enhancing recovery accuracy without increasing rank or parameters.
3. Scalable and Deployment-Friendly Design: The method combines a QR-reduced randomized SVD solver for efficient training with a selective restoration strategy (GlowQ-S), achieving substantial latency reduction (up to 23.4% lower TTFB) and throughput improvement (up to 37.4%) while maintaining compatibility with existing post-training quantization pipelines.

**Weaknesses:**

1. While the group-sharing idea is technically sound and well-executed, the central concept is an adaptation of well-known collective matrix factorization and SVD-sharing across blocks, now framed in the context of quantized LLMs. The extension to covariance alignment is also a known trick, though its synergy with group correction is effective.
2. While the paper compares GlowQ extensively to prior PTQ and error-corrected LLM quantization baselines (e.g., AWQ, GPTQ, L2QER, QERA, ZeroQuant), it omits direct comparison or numerical discussion of very recent approaches, such as rotation-based saliency-aware quantization methods (e.g., ROSAQ), vector quantization for KV cache (CommVQ, AnTKV), and loss-guided PTQ (GuidedQuant).
3. This method requires an additional calibration phase to estimate the compensation matrix, but the paper does not seem to list the time and memory costs for the training/calibration phase. Additionally, storing intermediate activations and error matrices for larger models (e.g., 30B+) could lead to significant memory pressure.
4. The paper claims the use of "custom CUDA W4A16 kernels," but it does not describe the implementation details or whether key techniques such as operator fusion were used in the low-rank correction modules. Therefore, it is unclear whether the comparisons between different methods are made at the same level of optimization.
5. Critically Limited and Potentially Misleading Experimental Scope (W4A16 only): The paper's most significant flaw is that all experiments are conducted in a W4A16 setting. This is an unrealistic and overly forgiving scenario that ignores the primary challenge in modern PTQ: activation quantization. By using FP16 activations, the main computation path and the input to the correction module remain noise-free, which dramatically simplifies the problem. The paper's claims of improved accuracy and efficiency are therefore unsubstantiated in any practical low-bit setting (e.g., W8A8, W4A4). It is entirely possible that the proposed group-sharing benefits would vanish or even become detrimental once activation quantization noise is introduced.
6. format error: line339 -> BitsAndBytes ?, line218,268,290,293 -> Missing formula number.

**Questions:**

1. The performance of GlowQ-S is highly dependent on the importance scoring functions (such as gec and gner). I am curious if there are other metrics that can indirectly reflect the effectiveness of these scoring functions. In other words, can these two metrics be proven to be globally optimal across a range of models?
2. The method relies on the assumption that modules within an input-sharing group can share a common right factor ( B_{\text{shared}} ), but it does not systematically verify the validity of this assumption across all layers or architectures. For example, when the functions of matrix projections like Q and V are different, it is unclear whether enforcing this sharing introduces bias.
3. The confinement to W4A16 is a major limitation. Can you provide any results or analysis for a W4A8 or W4A4 setting? How does your group-sharing approximation hold up when the input X to the correction path is also quantized, introducing another layer of error?

---

> ### Author Response · Authors · 2025-11-19
> **Weakness 1**
>
> We deeply appreciate the reviewer's careful reading and the perceptive comments that have helped us clarify and refine our work. Below, we address each of the points raised in your review.
>
> > While the group-sharing idea is technically sound and well-executed, the central concept is an adaptation of well-known collective matrix factorization and SVD-sharing across blocks, now framed in the context of quantized LLMs. The extension to covariance alignment is also a known trick, though its synergy with group correction is effective.
>
> We thank the reviewer for the insightful comment. We agree that our method builds on known tools such as collective matrix factorization and covariance-aligned low-rank approximation. Our contribution is not the reinvention of these tools but the new formulation and analysis that arise when these ideas are applied to quantization error in LLMs, where modules share inputs and activations are highly anisotropic. We highlight three domain-specific findings that, to our knowledge, do not appear in prior low-rank PTQ work:
>
> (i) **Optimality of a group-shared right factor.** Prior correctors (LQER, ASER, QERA, etc.) treat each projection independently. In contrast, in GlowQ, which formalizes input-sharing groups (Q/K/V, MLP gate/up), we show that the shared-$B$ structure is an optimal consequence of the error-reconstruction objective under shared inputs. Specifically, the natural least-squares problem over their quantization errors admits a single shared right factor $B$ as the global minimizer since separate $B_i$ offer no additional expressive power because differences can be absorbed into the left factors $A_i$.
>
> (ii) **A principled method to minimize quantization risk.** While whitening is a known technique, our key claim is that the covariance alignment is an imperative step for the optimal transformation under the actual activation distribution of LLMs. Concretely, the output error of any correction depends on how the residual interacts with the input covariance. Because LLM activations exhibit strong anisotropy, directions with high variance amplify remaining quantization error, i.e., the expected output error $\mathbb{E}||Mx||_2^2$ is exactly equivalent to the Frobenius error of the right-weighted residual $||(E{\mathrm{cat}} - AB)\Sigma_x^{1/2}||_F^2$. Our analysis shows that the correct objective must therefore weight residual errors according to the activation covariance to ensure the learned basis captures directions that dominate the model’s behavior.
>
> (iii) **Analysis of a shared covariance-aligned right subspace.** Using this formulation, we empirically find that the covariance-aligned shared subspace is nearly identical to each module’s own SVD subspace as shown in our cross-basis heatmaps. At the same time, group-shared GlowQ matches layer-wise low-rank correction in perplexity across 13 models and even in the MoE setting, despite using a single shared $B$ per group. After whitening, the dominant quantization-error directions are effectively shared across modules that see the same input tensor. To our knowledge, this phenomenon (dominant quantization-error directions becoming shared once activations are whitened) has not been characterized in prior work on low-rank error correction.

---

> ### Author Response · Authors · 2025-11-19
> **Weakness 2**
>
> > While the paper compares GlowQ extensively to prior PTQ and error-corrected LLM quantization baselines (e.g., AWQ, GPTQ, L2QER, QERA, ZeroQuant), it omits direct comparison or numerical discussion of very recent approaches, such as rotation-based saliency-aware quantization methods (e.g., ROSAQ), vector quantization for KV cache (CommVQ, AnTKV), and loss-guided PTQ (GuidedQuant).
>
> We would like to clarify our position on these methods (ROSAQ [1], GuidedQuant [2], CommVQ [3], AnTKV [4]), as we believe they are complementary or orthogonal to our proposed GlowQ, rather than direct baselines.
>
> The primary goal of GlowQ is to serve as an efficient error correction technique for weight quantization. The core problem we address is that existing low-rank error correction methods add significant latency and memory overhead by applying independent modules to every decoder block. GlowQ's main contribution is to drastically reduce this overhead by introducing group-shared factors and a cache-and-reuse mechanism.
>
> The methods you suggested address different aspects of model compression.
>
> 1.  Foundational PTQ Algorithms (e.g., ROSAQ [1], GuidedQuant [2]):
> $ $
>      * These methods focus on *how* to generate a better quantized weight, $A$ , to minimize the initial quantization loss. They are foundational PTQ algorithms.
>     * In our paper, we already include AWQ and GPTQ  (Table 1) as strong, representative baselines for this category.
>     * GlowQ is not a foundational PTQ algorithm itself, but rather a plug-and-play correction module that operates on top of an existing PTQ pipeline. This is clearly demonstrated in Table 4, where we successfully apply GlowQ on top of both GPTQ and BnB to achieve further performance gains.
>     * Therefore, we view ROSAQ [1] and GuidedQuant [2] as complementary, just like AWQ/GPTQ. They could serve as the base quantizer, and GlowQ could then be applied to efficiently correct their residual errors.
>
> 2.  KV Cache Quantization (e.g., CommVQ [3], AnTKV [4]):
>     * GlowQ's goal is to correct the error from weight quantization ($W \approx W_q + AB$). Its method is fundamentally activation-aware, as it uses input activation statistics ($\Sigma_x$) to find the optimal, data-preferred correction subspace.
>     * While our work involves activation quantization (e.g., W4A16, W4A8, etc.), this refers to the activations used *during the linear projection* for the correction $A(BX)$.
>     * This is fundamentally different from methods like CommVQ [3] and AnTKV [4]. They are not focused on the weights or the projection operation itself. Instead, they focus on compressing the KV Cache activations (i.e., the *outputs* of the K and V projections) that are *stored in memory* for subsequent decoding steps.
>     * Therefore, the two tasks remain orthogonal. GlowQ corrects the weight error during the projection, while CommVQ/AnTKV compress the resulting activations for memory savings. The two could be combined.
>
> For these reasons, we believe our current set of baselines which includes direct error-correction competitors (e.g., L2QER, QERA) and representative base PTQ methods (e.g., AWQ, GPTQ) is the most appropriate for evaluating GlowQ's core contribution: efficient and high-performance error correction.
>
> Nevertheless, we believe that it is a promising path for future research. We agree that integrating GlowQ with advanced PTQ methods like ROSAQ [1] or combining it with orthogonal techniques like CommVQ [3] could lead to even greater synergies in LLM compression. We include a new discussion for this potential in our Section 4.5.
>
> Thank you once again for your constructive feedback, which has helped us clarify the position and contribution of our work.

---

> ### Author Response · Authors · 2025-11-19
> **Weakness 3**
>
> > This method requires an additional calibration phase to estimate the compensation matrix, but the paper does not seem to list the time and memory costs for the training/calibration phase. Additionally, storing intermediate activations and error matrices for larger models (e.g., 30B+) could lead to significant memory pressure.
>
> GlowQ introduces an additional offline calibration stage. To clarify these overhead costs, we now profile the OPT-6.7B/13B/30B models with peak GPU/CPU memory, error/covariance tensor sizes, and calibration/decomposition time, listing these in the below table to capture costs associated with storing intermediate output for error statistics.
>
> Even for OPT-30B, the peak GPU memory during calibration fits within a single 80GB GPU, and the CPU memory usage, while higher due to storing statistics, occurs only in this one-time offline stage. We additionally include these results in Appendix D.1.2. of the revised version of manuscript.
>
> | model              | gpu_peak_GB | cpu_RAM_GB | err_size_GB | cov_size_GB | nsamples | calib_time_sec | decomp_time_sec |
> |--------------------|------------:|------------:|------------:|------------:|---------:|---------------:|----------------:|
> | facebook/opt-6.7b  |       14.68 |      114.66 |       24.00 |       38.06 |       32 |        1252.40 |          403.00 |
> | facebook/opt-6.7b  |       14.68 |      114.66 |       24.00 |       38.06 |       64 |        2389.30 |          403.64 |
> | facebook/opt-6.7b  |       14.68 |      114.66 |       24.00 |       38.06 |      128 |        4663.73 |          403.68 |
> | facebook/opt-13b   |       27.41 |      224.31 |       46.88 |       74.32 |       32 |        2388.55 |          952.88 |
> | facebook/opt-13b   |       27.41 |      224.31 |       46.88 |       74.32 |       64 |        4557.40 |          952.54 |
> | facebook/opt-13b   |       27.41 |      224.31 |       46.88 |       74.32 |      128 |        8895.11 |          952.24 |
> | facebook/opt-30b   |       77.91 |      527.31 |      110.25 |      174.75 |       32 |        5752.80 |         2335.14 |
> | facebook/opt-30b   |       77.91 |      527.31 |      110.25 |      174.75 |       64 |       10976.47 |         2335.34 |
> | facebook/opt-30b   |       77.91 |      527.31 |      110.25 |      174.75 |      128 |       21423.80 |         2335.94 |

---

> ### Author Response · Authors · 2025-11-19
> **Weakness 4**
>
> > The paper claims the use of "custom CUDA W4A16 kernels," but it does not describe the implementation details or whether key techniques such as operator fusion were used in the low-rank correction modules. Therefore, it is unclear whether the comparisons between different methods are made at the same level of optimization.
>
> We plan to release our code publicly for the CUDA W4A16 kernels which we already submitted as a supplementary with our original manuscript. To make this point explicit in the modified version, we have updated Section 4.3 to include the following clarification:
>
> > “This setup ensures a fair comparison, as both the Layerwise baseline and GlowQ utilize the identical custom *CUDA W4A16* kernels compiled with the same optimization level, isolating the algorithmic impact of our caching strategy.”
>
> While the custom kernel is necessary for an efficient implementation, its engineering details are not central to the paper’s scientific contributions; for completeness, we provide the key implementation aspects below.
>
>
> #### 1. Scope of the custom W4A16 kernel
>
> The “custom CUDA W4A16 kernel” has a single responsibility: computing the main quantized matrix–vector product ($W_q x$) in the decode regime.
>
> Concretely, it is a W4A16 GEMV kernel (4-bit weights, 16-bit activations) optimized for small-batch decoding (e.g., ($M \leq 128$)). The kernel fuses the on-the-fly dequantization of the 4-bit packed weights with per-group 16-bit scales and zero-points directly into the dot product with fp16 activations. That is, it loads an `int32` word, unpacks 8 4-bit values, applies the corresponding per-group scale and zero-point, and immediately accumulates the dequantized values into fp32 partial sums. This avoids any separate dequantization kernel or intermediate fp16 weight tensor.
>
> This custom kernel is used only for the small-batch decode path. For the large-batch prefill stage we use a standard “dequantize-then-GEMM” implementation backed by PyTorch/cuBLAS, where quantized weights are first dequantized to floating point and then multiplied.
>
> #### 2. Low-rank correction: no fusion into the W4A16 kernel
>
> Importantly, we do not fuse the low-rank correction term ($+A(Bx$)) into the custom W4A16 kernel. The low-rank part is always applied as a sequence of separate PyTorch operations after the ($W_q x$) computation has finished.
>
> The forward pass in our implementation proceeds as:
>
> 1. Compute the main quantized matvec ($W_q x$) using the custom CUDA W4A16 kernel.
> 2. Compute (R = Bx) using a separate call to `F.linear` (standard fp16 GEMM).
> 3. Compute (A R) using another separate `F.linear` call.
> 4. Add the two results using a separate in-place `torch.add_`.
>
> Thus, the full cost of two additional matrix multiplications and the element-wise addition required for the low-rank correction is explicitly present in our latency measurements. None of this work is hidden inside a fused CUDA epilogue that would give GlowQ an unfair advantage over other variants.
>
> #### 3. Fairness of the latency comparisons (Tables 3 and 8)
>
> With this in mind, we clarify what is being compared in Tables 3 and 8.
>
> Our latency evaluation does not compare against external PTQ baselines whose kernel implementations may differ. Instead, it compares different error-correction strategies implemented within the same codebase and on the same backend, namely:
>
> * **Layerwise**: a standard layer-wise low-rank correction ($A_i(B_i x)$) applied per module on top of the custom W4A16 kernel for ($W_q x$).
> * **GlowQ**: our proposed group-shared correction ($A_i(B_{\text{shared}} x)$), which reuses a shared (B) within a group of layers.
> * **GlowQ-S**: a lightweight variant of GlowQ that keeps the same shared-(B) structure but reduces the correction cost (e.g., effective rank).
>
> All three configurations share exactly the same custom CUDA W4A16 kernel for the core quantized matvec ($W_q x$). The low-rank term ($A(Bx)$) is implemented in all cases using the same pattern of PyTorch `F.linear` calls described above; the only difference is how often ($Bx$) must be recomputed and how the low-rank factors are structured (per-layer vs shared).
>
> Consequently, the latency differences in Tables 3 and 8 are not due to more aggressive kernel fusion for GlowQ, but instead reflect the algorithmic benefit of reusing ($R = B_{\text{shared}} x$) across multiple layers in GlowQ / GlowQ-S, compared to repeatedly computing ($B_i x$) in the Layerwise baseline.
>
> GlowQ-S is implemented on the same W4A16 and low-rank computation stack as GlowQ, so its additional speedup is purely due to its lighter correction configuration.

---

> ### Author Response · Authors · 2025-11-19
> **Weakness 6**
>
> > format error: line339 -> BitsAndBytes ?, line218,268,290,293 -> Missing formula number.
>
> We thank you for catching these formatting issues. All typos (BitsAndBytes) and missing formula numbers have been corrected in the revised manuscript.

---

> ### Author Response · Authors · 2025-11-19
> **Q1**
>
> > The performance of GlowQ-S is highly dependent on the importance scoring functions (such as gec and gner). I am curious if there are other metrics that can indirectly reflect the effectiveness of these scoring functions. In other words, can these two metrics be proven to be globally optimal across a range of models?
>
> As the reviewer pointed out, the importance scoring functions may affect the final quality of the error correction in GlowQ; we acknowledge that it is likely infeasible to derive any single metric  as 'globally optimal' in practice  across the full spectrum of models and tasks.
>
> In fact, during the GlowQ development, we investigated several established saliency metrics, which have been widely used in quantization [5, 6, 7], pruning [8, 9], and low-rank approximation [10, 11] literature, to verify their effectiveness for measuring layer importance in the error correction. However, we observed that different model families have distinct properties: for example, some models are more sensitive to weight magnitude, while others are more sensitive to outlier features.
>
> The metrics we analyzed (https://imgur.com/a/EFqIyrB) include:
>
> 1.  GSVD Energy Capture : This corresponds to our $g_{ec}$ score (Eq. 9). It measures the sum of squared singular values (or $||A||_F^2$), identifying groups where our low-rank approximation captures the most quantization error "energy." This is a standard approach in principal component analysis for measuring variance capture [10].
> 2.  Normalized Error Ratio : This corresponds to our $g_{ner}$ score (Eq. 10). It measures the Frobenius norm of the quantization error relative to the norm of the original weights ($||E_g||_F / ||W_g||_F$). This is a widely-used saliency metric in post-training quantization (PTQ) to find layers with the most *relative* sensitivity [5,6].
> 3.  Frobenius Norm of Error : An unnormalized version of $g_{ner}$ that measures the absolute magnitude of the quantization error ($||E_g||_F$). This is a core component of many quantization error minimization objectives [12,13].
> 4.  Cosine Similarity Error : Measures the change in *direction* of the weight vectors after quantization (i.e., $1 - \cos(\mathbf{w}, \mathbf{w_q})$). This captures errors not just in magnitude but also in angular deviation, a concept explored in model pruning [8].
> 5.  Layer Order : A simple baseline that restores layers sequentially, which we also noted as a fallback.
>
> The results show that the relative performance of these metrics varies by model family, e.g., LLaMA models show a distinct "knee," while Qwen models show a more gradual degradation. Given this model-dependent behavior, we selected the two metrics that provided the most robust and effective performance across our test suite: GSVD Energy Capture (`gec`) and Normalized Error Ratio (`gner`).
>
> As stated in Section 4.6, our final GlowQ-S policy evaluates both and adopts the one that yields the better result for a given model. This provides a practical, high-performance solution without claiming universal optimality.
>
> To provide further transparency on this process, the revised manuscript includes the new ablation study of all investigated metrics to Appendix G with the new results in Figure 12.

---

> ### Author Response · Authors · 2025-11-19
> **Q2**
>
> > The method relies on the assumption that modules within an input-sharing group can share a common right factor ( B_{\text{shared}} ), but it does not systematically verify the validity of this assumption across all layers or architectures. For example, when the functions of matrix projections like Q and V are different, it is unclear whether enforcing this sharing introduces bias.
>
> In our original (and revised) manuscript, we have shown the validity of the $B_{\text{shared}} $, especially for functionally different modules like Q and V,  both theoretically and empirically. Here, we summarize our main claim as follows (while the related discussion is included in Section 3.1.1, 3.1.2, Appendix B of our revised manuscript):
>
> ### 1. Theoretical Optimality for the Joint Problem
>
> The reviewer's concern is valid if we were approximating the functions of Q and V themselves. However, as noted in the Related Work (Section 2) and Method (Section 3), our method's goal is to approximate their quantization error matrices ($E_Q$, $E_V$, etc.). We frame this as a joint low-rank approximation problem.
>
> This approach is theoretically grounded in the established principle of Collective Matrix Factorization (CMF) [14], where multiple matrices sharing a common dimension are factored jointly. As we demonstrate in our paper (Sec. 3.1.1, Appx. A.1), the joint least-squares problem ($min_{A,B} ||E_{cat} - AB||F^2$) for the vertically stacked error matrix $E_{cat}$ is optimally solved by a single shared right factor $B$. By the Eckart-Young-Mirsky theorem, this shared factor simply spans the top-r right-singular subspace of the entire stacked matrix, and allowing individual $B_i$ factors adds no extra expressivity (as noted in our Observation 1).
>
> ### 2. Addressing Bias via Covariance Alignment
>
> Your concern about bias is correct for the naive SVD. LLM activations are known to be highly anisotropic [15,16], meaning the data is strongly concentrated in a few directions, as we discuss in our paper (Sec. 3.1.2). A naive SVD, which ignores this data distribution, would misalign the shared subspace and indeed introduce the very bias you mentioned.
>
> This is precisely why our method adopts a covariance-aligned objective ($min_{A,B}||(E_{cat}-AB)\Sigma_{x}^{1/2}||_{F}^{2}$) (Sec. 3.1.2). This is a principled method, consistent with established work on Weighted Low-Rank Approximations [17,18], which "whitens" the error and steers the shared subspace toward the data-preferred directions.
>
> ### 3. Systematic Empirical Verification
>
> We did not just assume this alignment; we systematically verified it. As shown in Appendix B (Figures 5 and 6) of our paper, we directly address this question. These heatmaps show the alignment between the computed shared basis ($B_{\text{shared}}$) and the individual bases for $E_Q$, $E_K$, $E_V$, etc.
>
> * **Evidence:** The "no whiten" (naive) plots show chaotic, unaligned subspaces. In contrast, the "whitened" (our method's) plots show a strong diagonal structure. This is the core empirical proof: once covariance-aligned, the data-weighted error subspaces of Q, K, and V *are* highly aligned and can be shared.
> * **Performance Proof:** The ultimate proof is in the performance. If our assumption introduced harmful bias, our method would underperform a baseline that uses individual $B_i$ factors ("Layer-wise"). As shown in our Appendix B.1 (Table 7), the perplexity of our method and the "LAYERWISE" baseline are "essentially identical" (mean gap +0.001 PPL) across 13 models.
>
> In summary, the $B_{\text{shared}}$ assumption is justified because: (1) it is theoretically optimal for the joint problem (as shown in our Appx. A.1), (2) we use a principled covariance-alignment method (our Sec. 3.1.2) to correct for data anisotropy [15,16], and (3) we have empirically verified in our paper both the subspace alignment (Appx. B) and the final performance (Table. 7), confirming no bias is introduced.

---

> ### Author Response · Authors · 2025-11-19
> **W5 & Q3**
>
> >Critically Limited and Potentially Misleading Experimental Scope (W4A16 only): The paper's most significant flaw is that all experiments are conducted in a W4A16 setting. This is an unrealistic and overly forgiving scenario that ignores the primary challenge in modern PTQ: activation quantization. By using FP16 activations, the main computation path and the input to the correction module remain noise-free, which dramatically simplifies the problem. The paper's claims of improved accuracy and efficiency are therefore unsubstantiated in any practical low-bit setting (e.g., W8A8, W4A4). It is entirely possible that the proposed group-sharing benefits would vanish or even become detrimental once activation quantization noise is introduced.
>
> > The confinement to W4A16 is a major limitation. Can you provide any results or analysis for a W4A8 or W4A4 setting? How does your group-sharing approximation hold up when the input X to the correction path is also quantized, introducing another layer of error?
>
> While our primary contribution focuses on correcting weight quantization error, as suggested by the reviewer, we conducted new experiments applying GlowQ in weight-and-activation (W&A) quantization settings. We found that our method remains effective when activations are also quantized.
>
>
> The table below presents the perplexity on WikiText-2 for both W4A4 and W4A8 configurations across a diverse range of models.
>
> | | Q config | LLaMA2 7B | LLaMA2 13B | LLaMA 3.2 3B | LLaMA 3.1 8B | Qwen 2.5 7B | Qwen 2.5 14B | Qwen 3 8B | Qwen 3 14B | Mistral 7B | OPT 1.3B | OPT 6.7B |
> | :--- | :---: | :---: | :---: | :---: | :---: | :---: | :---: | :---: | :---: | :---: | :---: | :---: |
> | FP16 | - | 5.48 | 4.90 | 7.81 | 6.24 | 6.86 | 5.29 | 9.73 | 8.64 | 5.32 | 14.62 | 10.85 |
> | GlowQ | W4A4 | 5.90 | 5.20 | 9.21 | 7.42 | 8.03 | 6.55 | 10.66 | 9.33 | 5.74 | 26.35 | 11.31 |
> | GlowQ-S | W4A4 | 5.92 | 5.20 | 9.25 | 7.45 | 8.05 | 6.61 | 10.72 | 9.37 | 5.79 | 27.42 | 11.33 |
> | GlowQ | W4A8 | 5.59 | 4.97 | 8.20 | 6.63 | 7.12 | 5.71 | 10.08 | 8.85 | 5.43 | 14.85 | 10.97 |
> | GlowQ-S | W4A8 | 5.60 | 4.97 | 8.24 | 6.64 | 7.13 | 5.77 | 10.10 | 8.92 | 5.48 | 14.99 | 10.99 |
>
> Specifically, in the W4A8 setting, GlowQ demonstrates exceptional stability, incurring only a marginal perplexity increase of approximately 2–4% compared to the FP16 baseline across the LLaMA and Mistral families. Even under the more aggressive W4A4 quantization, the performance degradation is effectively contained, generally staying within a 7–10% range for most modern architectures. Thus, we conclude that our data-driven correction method is robust and its benefits are preserved in combined W&A quantization scenarios.
>
> In the revised version of our manuscript, we incorporated these results into Section 4.2. In this updated section, we also included a comparison of GlowQ against other baseline error correction methods under these same W4A4 and W4A8 settings.

---

> ### Author Response · Authors · 2025-11-19
> **References**
>
> [1] Yoon, J., Lee, G., Jeon, D., Kang, I., & Na, S. H. (2025). ROSAQ: Rotation-based Saliency-Aware Weight Quantization for Efficiently Compressing Large Language Models. arXiv preprint arXiv:2506.13472.
>
> [2] Kim, J., El Halabi, M., Park, W., Schaefer, C. J., Lee, D., Park, Y., Lee, J. W., & Song, H. O. (2025). GuidedQuant: Large Language Model Quantization via Exploiting End Loss Guidance. arXiv preprint arXiv:2505.07004. (ICML 2025).
>
> [3] Li, J., Zhang, Y., Hassan, M. Y., Chafekar, T., Cai, T., Ren, Z., ... & Gan, C. (2025). CommVQ: Commutative Vector Quantization for KV Cache Compression. arXiv preprint arXiv:2506.18879.
>
> [4] Zhang, Y., et al. (2025). AnTKV: Anchor Token-Aware Sub-Bit Vector Quantization for KV Cache in Large Language Models. arXiv preprint arXiv:2506.19505.
>
> [5] Nagel, M., et al. (2021). A White Paper on Neural Network Quantization. arXiv:2106.08295.
>
> [6] Gholami, A., et al. (2021). A Survey of Quantization Methods for Efficient Neural Network Inference. arXiv:2103.13630.
>
> [7] Krishnamoorthi, R. (2018). Quantizing Deep Convolutional Networks for Efficient Inference: A Whitepaper. arXiv:1806.08342.
>
> [8] Mason-Williams, G., & Dahlqvist, F. (2024). What Makes a Good Prune? Maximal Unstructured Pruning for Maximal Cosine Similarity. ICLR 2024.
>
> [9] Chang, J., et al. (2023). Iterative Clustering Pruning for Convolutional Neural Networks. Knowledge-Based Systems, 277, 110825.
>
> [10] Jolliffe, I. T. (2002). Principal Component Analysis, 2nd ed. Springer.
>
> [11] Halko, N., Martinsson, P. G., & Tropp, J. A. (2011). Finding Structure with Randomness: Probabilistic Algorithms for Constructing Approximate Matrix Decompositions. SIAM Review, 53(2), 217-288.
>
> [12] Pouransari, H., & Boufounou, P. (2020). Least Squares Binary Quantization of Neural Networks. CVPR Workshops.
>
> [13] Zhao, W., et al. (2025). ASER: Activation Smoothing and Error Reconstruction for Large Language Model Quantization. AAAI 2025.
>
> [14] A. P. Singh and G. J. Gordon. "Relational learning via collective matrix factorization." In Proceedings of KDD, 2008.
>
> [15] K. Ethayarajh. "How contextual are contextualized word representations? Comparing the geometry of BERT, ELMo, and GPT-2 embeddings." In Proceedings of EMNLP-IJCNLP, 2019.
>
> [16] N. Godey, É. de la Clergerie, and B. Sagot. "Anisotropy is inherent to self-attention in transformers." arXiv preprint, 2024.
>
> [17] N. Srebro and T. Jaakkola. "Weighted low-rank approximations." In Proceedings of ICML, 2003.
>
> [18] G. H. Golub and C. F. Van Loan. "Matrix Computations." Johns Hopkins University Press, 4th edition, 2013.
>
> ---
>
> We would like to thank you again for your time and expertise. Your detailed feedback has been invaluable in refining our manuscript.

---

> ### Comment · Reviewer_HaNH · 2025-11-25
>
> Thank you for providing such a detailed and comprehensive explanation in response to the questions raised. I will revisit and reconsider my previous score of this work.

---

> > ### Author Response · Authors · 2025-11-25
> >
> > Dear Reviewer HaNH,
> >
> > Thank you very much for your positive feedback and for taking the time to reconsider our work. We truly appreciate your engagement with our paper.
> >
> > Authors

---

### Official Review · Reviewer_DXnt · 2025-10-31

**Soundness:** 3
**Presentation:** 2
**Contribution:** 3
**Rating:** 4
**Confidence:** 5

**Summary:**

This paper introduces GlowQ, a group shared low-rank approximation method, that a single rightside projection is used for a group with same input. A BX caching method is proposed to make it deployable. A selective method is proposed to further reduce the memory and latency.

**Strengths:**

- This paper is technically sound.
- The experimental results show the effectiveness of the proposed method.

**Weaknesses:**

- Writing and presentation need refinement. For instance, the symbol E in Eq. 1 is identical to E_cat in Eq. 2—consistent notation should be used. In Fig. 1 (inference-path sub-figure), the chosen colors are too similar to be distinguished; a more discernible palette is required.
- For GlowQ-S, the fraction of restored groups are not provided, should be provided for each experiment

**Questions:**

- In Fig.3, as PPL is lower the better, what is the definition of the percentage of PPL?
- In Tab.4, No Caching means GlowQ without caching? Or a layerwise method? If it means GlowQ without caching, the authors should provide performance comparisons with layerwise methods.
- Modern GPUs already natively support other narrow-bit formats such as MXFP4, NVFP4 and MXFP6, whose peak throughput is significantly higher than that of INT4. The paper should therefore clarify:
a) Is GlowQ directly applicable to these FP4/FP6 formats, or does its low-rank projection rely on integer-only operators?
b) If applicable, how does GlowQ’s accuracy compare with simply running the network in MXFP4/NVFP4/MXFP6 without any additional compression?

---

> ### Author Response · Authors · 2025-11-19
> **Weakness 1**
>
> We are grateful for the reviewer's insightful feedback and detailed examination of our work, which have helped us to significantly improve the paper. We address the valuable points you raised in the following responses.
>
> >  Writing and presentation need refinement. For instance, the symbol E in Eq. 1 is identical to E_cat in Eq. 2—consistent notation should be used. In Fig. 1 (inference-path sub-figure), the chosen colors are too similar to be distinguished; a more discernible palette is required.
>
> Upon request, we have thoroughly revised the entire manuscript. In addition, we have corrected the notation for consistency and updated the color palette in Figure 1 for better distinguishability.
> > $E_{cat}:=[E^T_1 ... E^T_m]$

---

> ### Author Response · Authors · 2025-11-19
> **Weakness 2**
>
> > For GlowQ-S, the fraction of restored groups are not provided, should be provided for each experiment.
>
> Based on your feedback, we clarified the fraction of restored groups for GlowQ-S in the caption of Table 2 as follows:
>
> > "GlowQ-S restores 51% of layers for LLaMA 3.2-3B, while all other models use 50% restoration."

---

> ### Author Response · Authors · 2025-11-19
> **Q1**
>
> > In Fig.3, as PPL is lower the better, what is the definition of the percentage of PPL?
>
> To illustrate the trade-off, we normalized the results for both metrics. For the PPL axis, the highest (worst) PPL value from our experiments was set as the 100% baseline, and the lowest (best) PPL value was set to 0%. The TTFB values were normalized in the same manner.
>
> We understand that this normalization, while intended to show the relationship between the metrics, could cause confusion. To improve clarity, we have revised Figure 3 by adding a secondary y-axis on the right side to display the exact PPL values.

---

> ### Author Response · Authors · 2025-11-19
> **Q2**
>
> >  In Tab.4, No Caching means GlowQ without caching? Or a layerwise method? If it means GlowQ without caching, the authors should provide performance comparisons with layerwise methods.
>
> In Table 4, the row we labeled "No Caching" was intended to represent the standard layerwise baseline, where quantization is applied sequentially layer by layer without our caching strategy.
>
> > To resolve the potential misleading, we have updated Table 4 by renaming the "No Caching" label to "Layerwise." We have also revised the corresponding descriptions in the revised version.

---

> ### Author Response · Authors · 2025-11-19
> **Q3**
>
> > Modern GPUs already natively support other narrow-bit formats such as MXFP4, NVFP4 and MXFP6, whose peak throughput is significantly higher than that of INT4. The paper should therefore clarify: a) Is GlowQ directly applicable to these FP4/FP6 formats, or does its low-rank projection rely on integer-only operators? b) If applicable, how does GlowQ’s accuracy compare with simply running the network in MXFP4/NVFP4/MXFP6 without any additional compression?
>
> **a) Is GlowQ directly applicable to these FP4/FP6 formats?**
>
> Yes, GlowQ is directly applicable. Our method's low-rank projection is an empirical, data-driven correction of the quantization error (the delta between \(W\) and \(Q(W)\)). This approach is agnostic to the specific data type; it learns to correct the error statistics whether \(Q(W)\) is an integer or a low-bit float. The method does not rely on integer-only operators. Please refer to the answer for Reviwer CZae.
>
> **b) How does GlowQ’s accuracy compare with simply running the network in MXFP4/NVFP6/NVFP4?**
>
> We conducted experiments on the specific data types mentioned. We used the Mistral 7B model, and the results are reported as perplexity (PPL) measured on the Wikitext-2 dataset. The low-bit floating-point data types were implemented following recent specifications and research [1, 2, 3].
>
> The “Quant only” row represents the baseline accuracy of running the network in that format. The “GlowQ” row shows the accuracy after applying our correction.
>
> |               | **FP16 (Baseline)** | **MXFP4** | **MXFP6** | **NVFP4** |
> | :-----------: | :-----------------: | :-------: | :-------: | :-------: |
> | **Quant only** | 5.32               | 8.05      | 5.36      | 6.09      |
> | **GlowQ**      | 5.32               | 6.10      | 5.32      | 5.63      |
>
> Relative to the quant-only baselines, GlowQ reduces perplexity by (−1.95) on MXFP4, (−0.04) on MXFP6, and (−0.46) on NVFP4. Thus, even when the model is already using modern low-bit floating-point formats, GlowQ provides additional accuracy gains on top of the native microscaling baselines.
>
> These experiments demonstrate that even though the quantization error’s data statistics differ qualitatively across data types, GlowQ’s data-driven correction approach is robust to these variations and generalizes well. Therefore, GlowQ is not only compatible with modern FP formats but also provides a measurable accuracy improvement over their native use by successfully correcting their associated quantization errors. We clarified this in the revised manuscript by adding Section E, Compatibility Across Quantization Datatypes, where we analyze GlowQ’s behavior on both integer and microscaling floating-point formats.

---

> ### Author Response · Authors · 2025-11-19
> **References**
>
> [1] Lee, J., Park, J., Cha, S., Cho, J., & Sim, J. (2025). *MX+: Pushing the Limits of Microscaling Formats for Efficient Large Language Model Serving*. arXiv preprint arXiv:2510.14557.
>
> [2] Open Compute Project. (2024). *OCP Microscaling Formats (MX) Specification Version 1.0*. Archived from the original on 2024-02-24.
>
> [3] NVIDIA Technical Blog. (2025, June 24). *Introducing NVFP4 for Efficient and Accurate Low-Precision Inference*.
>
> ---
>
> We thank you once again for your time and feedback, which have significantly helped us improve the quality and clarity of our paper.

---

> ### Author Response · Authors · 2025-11-26
>
> Dear reviewer DXnt,
>
> Thank you again for your thoughtful feedback and constructive comments. As the rebuttal period is nearing its end, we would kindly like to ask if you could take a moment to review our responses and the updated manuscript to check whether we have addressed your concerns adequately. If you have any further questions, remarks, or requests, we would be very happy to clarify or provide additional results. We sincerely appreciate your time and consideration.
>
> Authors

---

### Official Review · Reviewer_CZae · 2025-10-31

**Soundness:** 3
**Presentation:** 2
**Contribution:** 3
**Rating:** 6
**Confidence:** 5

**Summary:**

This work describes an efficient low-rank compensation technique for weight-quantized LLMs.
The method finds an optimal shared down-projection factor in a data-driven manner, with whitening and grouping treatment.

**Strengths:**

+ The idea is sound, with adequate proof.
+ The method is relatively well described.
+ There is potential practical significance.

**Weaknesses:**

- The empirical, data-driven, low-rank correction of quantization error is dependent on the data type and its precision.  As most of the results presented are with INT4, there is a lack of demonstration on the effectiveness for different data types and precisions--do the data statistics differ qualitatively under those conditions?
- As activation quantization is also important in practice, it is not clear how quantization of the activations, in combination with weight quantization, change the story.
- Quantization errors tend to accumulation over time across tokens as well, which is particularly relevant in long-context reasoning settings.  Reporting accuracy in language modeling perplexity is insufficient in addressing this issue.

**Questions:**

I have listed major questions in the Weaknesses section above.  Here are a few minor questions in addition.

* The empirical statistical analysis of the quantization error is essential to this paper.  Do you have any results on any empirical scaling laws of the statistics, in addition to simple qualitative descriptions such as long tail?  It might be helpful to apply logarithmic scale in Figure 2 to expose certain power laws.
* Is there any results on how and how well this method could be applied to MoE FFN layers?
* In addition to full-precision error correction, other orthogonal methods use mixed-precision, how is the choice among these methods subject to practical tradeoff?

---

> ### Author Response · Authors · 2025-11-19
> **Weakness 1**
>
> We sincerely thank you for your careful review and valuable feedback, which have significantly contributed to the improvement of our paper. We provide the following responses to address the questions and concerns you raised.
>
> >The empirical, data-driven, low-rank correction of quantization error is dependent on the data type and its precision. As most of the results presented are with INT4, there is a lack of demonstration on the effectiveness for different data types and precisions--do the data statistics differ qualitatively under those conditions?
>
> To address the concern raised in W1, we conducted experiments to verify that GlowQ's effectiveness is not limited to INT4 but generalizes robustly to other data types and precisions.
>
> We experimented with various integer precisions (INT2, INT3, INT4) and low-bit floating-point data types (MXFP4, MXFP6, NVFP4) using the Mistral 7B model. For our methodology, the integer precisions were quantized using a group size of 128 (G128). The low-bit floating-point data types were implemented following recent specifications and research [1, 2, 3]. The results are reported as perplexity (PPL) measured on the Wikitext-2 dataset.
>
> The results are presented in the table below.
>
> | | **FP16 (Baseline)** | **INT2** | **INT3** | **INT4** | **MXFP4** | **MXFP6** | **NVFP4** |
> | :--- | :---: | :---: | :---: | :---: | :---: | :---: | :---: |
> | **Quant only** | 5.32 | 1015.39 | 6.16 | 5.51 | 8.05 | 5.36 | 6.09 |
> | **GlowQ** | 5.32 | 24.23 | 5.84 | 5.41 | 6.10 | 5.32 | 5.63 |
>
>
>
> As shown in the table, our results demonstrate that GlowQ’s empirical, data-driven correction method remains consistently effective across diverse quantization settings. Its behavior does not depend on a particular data type or precision, even though the statistical characteristics of quantization error vary qualitatively across formats.
>
> Therefore, the effectiveness of GlowQ is not limited to INT4. It successfully corrects quantization errors and restores model performance across a broad spectrum of scenarios, including ultra-low-bit integers (like INT2) and various floating-point formats. We will incorporate these results and discussions into the revised manuscript as a new Section E, Compatibility Across Quantization Datatypes, to explicitly analyze GlowQ’s behavior across integer and microscaling formats.

---

> ### Author Response · Authors · 2025-11-19
> **Weakness 2**
>
> > As activation quantization is also important in practice, it is not clear how quantization of the activations, in combination with weight quantization, change the story.
>
> While our primary contribution focuses on correcting weight quantization error, as suggested by the reviewer, we conducted new experiments applying GlowQ in weight-and-activation (W&A) quantization settings. We found that our method remains effective when activations are also quantized.
>
>
> The table below presents the perplexity on WikiText-2 for both W4A4 and W4A8 configurations across a diverse range of models.
>
> | | Q config | LLaMA2 7B | LLaMA2 13B | LLaMA 3.2 3B | LLaMA 3.1 8B | Qwen 2.5 7B | Qwen 2.5 14B | Qwen 3 8B | Qwen 3 14B | Mistral 7B | OPT 1.3B | OPT 6.7B |
> | :--- | :---: | :---: | :---: | :---: | :---: | :---: | :---: | :---: | :---: | :---: | :---: | :---: |
> | FP16 | - | 5.48 | 4.90 | 7.81 | 6.24 | 6.86 | 5.29 | 9.73 | 8.64 | 5.32 | 14.62 | 10.85 |
> | GlowQ | W4A4 | 5.90 | 5.20 | 9.21 | 7.42 | 8.03 | 6.55 | 10.66 | 9.33 | 5.74 | 26.35 | 11.31 |
> | GlowQ-S | W4A4 | 5.92 | 5.20 | 9.25 | 7.45 | 8.05 | 6.61 | 10.72 | 9.37 | 5.79 | 27.42 | 11.33 |
> | GlowQ | W4A8 | 5.59 | 4.97 | 8.20 | 6.63 | 7.12 | 5.71 | 10.08 | 8.85 | 5.43 | 14.85 | 10.97 |
> | GlowQ-S | W4A8 | 5.60 | 4.97 | 8.24 | 6.64 | 7.13 | 5.77 | 10.10 | 8.92 | 5.48 | 14.99 | 10.99 |
>
> Specifically, in the W4A8 setting, GlowQ demonstrates exceptional stability, incurring only a marginal perplexity increase of approximately 2–4% compared to the FP16 baseline across the LLaMA and Mistral families. Even under the more aggressive W4A4 quantization, the performance degradation is effectively contained, generally staying within a 7–10% range for most modern architectures. Thus, we conclude that our data-driven correction method is robust and its benefits are preserved in combined W&A quantization scenarios.
>
> In the revised version of our manuscript, we incorporated these results into Section 4.2. In this updated section, we also included a comparison of GlowQ against other baseline error correction methods under these same W4A4 and W4A8 settings.

---

> ### Author Response · Authors · 2025-11-19
> **Weakness 3**
>
> >  Quantization errors tend to accumulation over time across tokens as well, which is particularly relevant in long-context reasoning settings. Reporting accuracy in language modeling perplexity is insufficient in addressing this issue.
>
> | Method    | NarrativeQA | Qasper | MultiFieldQA | HotpotQA | MuSiQue | 2WikiMQA | GovReport | QMSum | MultiNews | LCC    | RepoBench-P | TriviaQA | SAMSum | TREC  | PR    | Avg   |
> |-----------|-------------|--------|--------------|----------|---------|----------|-----------|-------|-----------|--------|-------------|----------|--------|-------|-------|-------|
> | fp16      | 18.26       | 12.01  | 25.96        | 13.76    | 7.87    | 14.95    | 32.79     | 21.43 | 25.95     | 51.93  | 47.00       | 87.76    | 44.72  | 70.00 | 37.50 | 34.19 |
> | W4A4+AB   | 14.68       | 10.80  | 24.95        | 14.21    | 8.39    | 14.20    |32.01         | 22.01     | 26.43         | 47.50     | 37.51          | 85.54        | 42.05      | 69.00     | 36.36     | 32.38     |
> | W4A8+AB   | 15.56       | 11.77  | 23.71        | 14.39    | 8.41    | 14.92    | 32.00     | 21.19 | 27.03     | 51.50  | 35.97       | 84.10    | 42.62  | 68.50 | 37.08 | 32.58 |
> | W4A16+AB  | 15.46       | 11.82  | 23.68        | 14.39    | 7.77    | 14.53    | 32.32     | 21.20 | 26.86     | 50.46  | 35.59       | 84.30    | 42.67  | 68.50 | 37.17 | 32.45 |
>
>
> | Method    | NarrativeQA | Qasper | MultiFieldQA | HotpotQA | MuSiQue | 2WikiMQA | GovReport | QMSum | MultiNews | LCC    | RepoBench-P | TriviaQA | SAMSum | TREC  | PR    | Avg   |
> |-----------|-------------|--------|--------------|----------|---------|----------|-----------|-------|-----------|--------|-------------|----------|--------|-------|-------|-------|
> | fp16      | 23.50       | 13.54  | 27.87        | 16.83  | 10.94    | 16.44    | 34.27    | 22.87 | 26.87    | 52.81  | 48.04       | 90.77   | 43.94  | 71.00 | 73.13 | 38.18 |
> | W4A4+AB   |23.45     | 12.20  | 27.41       | 15.34    | 9.21    | 16.15    | 33.87         |22.78     | 26.39         | 48.73     | 38.83           | 88.78        | 42.43     | 70.50     |70.52     | 36.44     |
> | W4A8+AB   | 25.38     | 12.61  | 25.71        | 15.37   | 9.93   | 15.30    | 34.07         | 22.67     | 27.14        | 52.06      | 38.55           | 88.49        | 43.60      | 71.00    | 72.73    | 36.97     |
> | W4A16+AB  | 25.36       | 12.61  | 25.62      | 15.13   | 9.82   | 15.20    | 34.00     | 22.59 | 26.96    | 52.12  | 38.84       | 88.67   | 43.44  | 71.00 | 73.50 | 36.92 |
>
>
> To verify the model's reasoning capabilities in long-context settings and address the concern regarding the accumulation of quantization errors across tokens, we moved beyond simple perplexity measurement and conducted additional experiments using the comprehensive long-context benchmark, LongBench ([https://github.com/THUDM/LongBench.git](https://github.com/THUDM/LongBench.git)).
>
> The experiments were performed on the Llama-3.1-8B-Instruct model using both 4K and 8K input context settings. As presented in the tables above (Table 1, Table 2), the models applying GlowQ (W4A4, W4A8, W4A16) maintain superior performance across all context lengths, exhibiting no significant degradation compared to the FP16 Baseline model.
>
> Notably, even in long sequences such as 8K, the W4A8+GlowQ (Avg: 36.97) result is highly competitive with the Baseline (Avg: 38.18). This demonstrates that GlowQ effectively manages the potential issue of quantization error accumulation that can occur over extended token lengths. We have included these detailed experimental results in Appendix F.

---

> ### Author Response · Authors · 2025-11-19
> **Q1**
>
> > The empirical statistical analysis of the quantization error is essential to this paper. Do you have any results on any empirical scaling laws of the statistics, in addition to simple qualitative descriptions such as long tail? It might be helpful to apply logarithmic scale in Figure 2 to expose certain power laws.
>
> Following this suggestion, we have updated Figure 2 (https://imgur.com/a/srLNOUI) by adding a new panel (b), which plots the empirical eigenvalue spectrum of the input covariance in $(\log_{10}\lambda_r,\ \log_{10} r)$ coordinates for both the QKV and MLP groups. In this log–log plot, the tail region is well approximated by a straight line over roughly one to two decades in $r$, indicating an approximate power-law decay $\lambda_r \propto r^{-\alpha}$. A least-squares fit in this linear regime yields exponents $\alpha_{\mathrm{MLP}} \approx 0.77$ and $\alpha_{\mathrm{QKV}} \approx 1.19$, which quantitatively confirms that the activation statistics exhibit heavy-tailed structure.
>
> For completeness, with panels (c)-(d), we show that the induced quantization-error matrices also have low effective rank. Most of their Frobenius energy is captured by a relatively small rank compared to the full dimension, also confirming that the use of a low-rank error-correction subspace is effective in the error correction.

---

> ### Author Response · Authors · 2025-11-19
> **Q2**
>
> > Is there any results on how and how well this method could be applied to MoE FFN layers?
>
> We conducted experiments to directly apply the core idea of our proposed method, GlowQ which is to group errors from layers that share the same input to Mixture-of-Experts (MoE) architectures.
>
>
>  We used the representative MoE model, Qwen1.5-MoE-A2.7B, as our benchmark. In an MoE structure, the expert FFN layers and the shared MLP layer all share the same input before routing. Following the GlowQ methodology, we grouped these components together to apply quantization error correction.
>
> The Wikitext-2 perplexity (PPL) results for the Qwen1.5-MoE-A2.7B model are as follows:
>
> | Model | Qwen1.5-MoE-A2.7B (PPL) |
> | :--- | :--- |
> | FP16 (Original) | 7.22 |
> | 4-bit Quant only | 7.70 |
> | GlowQ (Proposed) | 7.41 |
> | Layerwise (Baseline) | 7.39 |
>
>  The result shows that 4-bit uniform quantization (Quant only) degrades the PPL from 7.22 to 7.70, a loss of 0.48.
>  GlowQ recovers 0.29 of this PPL loss, achieving a PPL of 7.41 (recovering ~60% of the loss). The 'Layerwise' baseline (which applies a separate correction module to every expert) recovers 0.31, achieving a PPL of 7.39 (recovering ~65% of the loss). Furthermore, our GlowQ method achieves nearly identical accuracy to the Layerwise method, with only a +0.02 PPL difference.
>
> This offers a significant advantage in reducing computation overhead for the the error correction. For example, the Layerwise method requires a separate error-correction matrix for every single expert; in contrast, the GlowQ method uses a single shared right-hand matrix ($B_{\text{shared}}$ ) across all experts and the shared MLP within the group. For Qwen1.5-MoE-A2.7B, we measured that GlowQ achieves nearly equivalent accuracy while reducing the memory footprint required for the low-rank correction by approximately 63% compared to the Layerwise approach.
>
>
> It is noteworthy that the reason this efficient sharing is possible can be seen in the alignment heatmaps in Figures 7,8 and 9 (https://imgur.com/a/LENoJim). The heatmaps visually demonstrate that the error subspaces of individual experts are well-aligned with the shared right-hand subspace ($B_{\text{shared}}$) identified by GlowQ. This confirms that a single shared $B$ matrix can effectively correct the quantization errors of multiple experts.
>
> In the revised manuscript, we added the detailed analysis to Section 4.5 (Compatibility with PTQ Methods and Generalization to MoE) of the paper.

---

> ### Author Response · Authors · 2025-11-19
> **Q3**
>
> > In addition to full-precision error correction, other orthogonal methods use mixed-precision, how is the choice among these methods subject to practical tradeoff?
>
> • Compression vs. FP budget. Mixed-precision improves robustness by raising the bit-width of some layers from 4-bit to 8/16-bit. However, this directly reduces the effective compression ratio. GlowQ keeps the backbone fully 4-bit and uses a small FP16 low-rank head for error correction. In our observation, GlowQ outperforms the state-of-the-art mixed precision based quantization method, e.g., AWQ, up to 0.53 ppl points, as shown in Table 1. As our method introduces only negligible FP16 overhead, we believe that GlowQ offers a competitive solution in scenarios requiring strict 4-bit compression with tightly constrained floating-point computation.
>
> • Memory efficiency. Mixed-precision needs to keep salient components, e.g., channels and/or layers, in a high-precision mode, In contrast, GlowQ keeps the 4-bit storage for the main backbone while only maintaining minimal high-precision weights, typically less than 5% as shown in Figure 4.

---

> ### Author Response · Authors · 2025-11-19
> **References**
>
> [1] Lee, J., Park, J., Cha, S., Cho, J., & Sim, J. (2025). *MX+: Pushing the Limits of Microscaling Formats for Efficient Large Language Model Serving*. arXiv preprint arXiv:2510.14557.
>
> [2] Open Compute Project. (2024). *OCP Microscaling Formats (MX) Specification Version 1.0*. Archived from the original on 2024-02-24.
>
> [3] NVIDIA Technical Blog. (2025, June 24). *Introducing NVFP4 for Efficient and Accurate Low-Precision Inference*.
>
> ---
>
> Once again, we thank you for your insightful review and constructive suggestions for improving our work.

---

> ### Author Response · Authors · 2025-11-26
>
> Dear reviewer CZae,
>
> Thank you again for your thoughtful feedback and constructive comments. As the rebuttal period is nearing its end, we would kindly like to ask if you could take a moment to review our responses and the updated manuscript to check whether we have addressed your concerns adequately. If you have any further questions, remarks, or requests, we would be very happy to clarify or provide additional results. We sincerely appreciate your time and consideration.
>
> Authors

---

### Author Response · Authors · 2025-12-01
**Summary of Revisions**

Dear Area Chair,

In this comment, we summarize how our revision addresses the reviewers’ main concerns and highlight the key changes in the updated manuscript.

1. Concern about being restricted to INT4 and W4A16 (datatype generality and W&A quantization)

Reviewers CZae (W1, W2), HaNH (W5 and Q3), and DXnt (Q3) wondered if GlowQ is limited to an INT4 + W4A16 setting and if it remains effective when activations are also quantized. To respond, the revision includes new experiments and clarifications.

(1) Generality across data types (new Appendix E)

* We added experiments on Mistral-7B using integer precisions INT2, INT3, INT4, and low-bit floating-point formats MXFP4, MXFP6, and NVFP4.
* For each format, we compared “Quant only” and “Quant only + GlowQ” on WikiText-2 perplexity. We found that GlowQ consistently improves perplexity over the quant-only baselines in all formats.
* This shows that GlowQ’s low-rank correction is not limited to INT4 but is a data-type-agnostic error-correction method applicable across integer and microscaling formats.
* These results are summarized in the new Appendix E: Compatibility Across Quantization Datatypes.

(2) Weight-and-Activation (W&A) quantization (Section 4.2)

* We included experiments on important W4A4 and W4A8 settings across 11 models (LLaMA-2/3, Qwen-2.5/3, Mistral, OPT) evaluated on WikiText-2 perplexity.
* Both GlowQ and GlowQ-S maintain small accuracy gaps compared to FP16 even under W4A4 and W4A8, and are competitive with existing mixed-precision baselines.
* These results are integrated into the main body in Section 4.2 to emphasize that GlowQ is not limited to W4A16.
* We also added comparisons against another error-correction baseline (L2QER) using the same W4A4/W4A8 setups.

2. Quantization error accumulation in long-context settings (LongBench results)

Reviewer CZae (W3) pointed out that short-context perplexity is not enough to assess error accumulation in long-context reasoning, and requested an evaluation on a long-context benchmark. We therefore added the following experiments:

* We evaluated Llama-3.1-8B-Instruct on 15 English tasks from LongBench with context lengths of 4K and 8K tokens.
* When comparing FP16 with W4A4, W4A8, and W4A16 + GlowQ variants, we found that across both 4K and 8K contexts, GlowQ-based models maintain task scores very close to the FP16 baseline. In particular, at 8K, the average score of W4A8 + GlowQ stays near the FP16 baseline.
* This indicates that GlowQ preserves perplexity on short sequences and effectively manages quantization error accumulation over long token sequences.
* These results are summarized in the new Appendix F: LongBench Results.

3. Shared-B assumption, covariance alignment, and scaling-law analysis

Reviewers HaNH (W1, Q2) and CZae (Q1) asked whether projections with different functions (such as Q and V) can safely share a single $B_{shared}$, whether we can go beyond qualitative “long tail” descriptions to more quantitative scaling laws, and whether the role of covariance alignment is justified. The revision strengthens these points as follows.

(1) Covariance-weighted objective (Section 3.1)

* We define quantization risk as the output error under the actual activation distribution $Σ_x$ and formulate GlowQ’s objective as minimizing the Frobenius norm of $(E_{cat} − AB) Σ_x^{1/2}$, a covariance-weighted low-rank approximation.
* This clarifies that GlowQ does not perform a basic SVD, instead, it addresses a weighted low-rank problem that takes anisotropic LLM activations into account.

(2) Scaling law of input covariance (revised Figure 2)

* Following reviewer CZae’s suggestion, we added a log–log panel to Figure 2 that shows the eigenvalue spectrum of the input covariance.
* Plotting the spectra of QKV and MLP groups in ($log_{10} λ_r, log_{10} r$) coordinates reveals that the tail region is well represented by a straight line, indicating a power-law decay $λ_r ∝ r^{−α}$. We estimate the exponents α for each group via least-squares fitting.
* This gives quantitative evidence that the activation statistics have a heavy-tailed, anisotropic structure.

(3) Empirical validation of the shared-B assumption (Appendix B)

* We expanded the cross-basis heatmap analysis in Appendix B. Before whitening, the error subspaces of different modules are poorly aligned. After covariance alignment, the alignment with the shared basis becomes nearly diagonal.
* This shows that even for functionally different projections like Q/K/V and gate/up, the data-weighted main error directions converge to a shared subspace once whitening is applied.
* Furthermore, Appendix B.1 (Table 7) shows that, across 13 models, the perplexity gap between the Layerwise baseline and GlowQ is only about +0.001 on average, providing quantitative evidence that the shared-B assumption does not introduce harmful bias.

---

> ### Author Response · Authors · 2025-12-01
> **Summary of Revisions 2**
>
> 4. Positioning GlowQ relative to ROSAQ, GuidedQuant, CommVQ, and AnTKV
>
> Reviewers HaNH (W2) and CZae (Q3) asked for a clearer explanation of GlowQ’s relation to recent methods like rotation-based saliency-aware PTQ (ROSAQ), loss-guided PTQ (GuidedQuant), and KV-cache compression methods (CommVQ, AnTKV). In the revision, we added the following discussion in Section 4.5:
>
> * ROSAQ, GuidedQuant, AWQ, GPTQ, and related methods are base PTQ algorithms that aim to produce better quantized weights. In contrast, GlowQ is a plug-and-play error-correction module that works on top of these base quantizers.
> * As shown in Table 5, GlowQ can be applied on top of GPTQ and BitsAndBytes, providing extra performance gains beyond what the base PTQ method achieves.
> * CommVQ and AnTKV compress stored K/V activations (the KV cache), while GlowQ corrects weight errors in the projection step $W_q x$; these are separate problem settings.
> * Therefore, these methods are complementary rather than competing, and combining GlowQ with advanced PTQ or KV-cache compression methods represents a promising future direction.
>
> 5. Extension to MoE architectures and comparison to Layerwise correction
>
> Reviewer CZae (Q2) asked how GlowQ works on Mixture-of-Experts (MoE) FFN layers. The revision includes the following:
>
> * For Qwen1.5-MoE-A2.7B, we observe that expert FFNs and the shared MLP share the same input before routing. We group them as a single input-sharing group and apply GlowQ.
> * Under 4-bit uniform quantization, PPL degrades from 7.22 to 7.70. GlowQ, using a single shared $B_{shared}$ across experts, recovers about 60% of this loss and reaches 7.41, while a Layerwise baseline with per-expert $B_i$ recovers about 65% of the loss (PPL 7.39). Accuracy is nearly identical.
> * Meanwhile, the Layerwise approach must maintain a separate error-correction module for each expert. We measure that GlowQ achieves similar accuracy while reducing low-rank correction memory by about 63% in this MoE setting.
> * These results are summarized in Section 4.5. Appendix B (Figures 7, 8, and 9) further shows, via alignment heatmaps, that the error subspaces of individual experts are well aligned with the shared right-hand basis $B_{shared}$ identified by GlowQ, confirming that a single shared B can effectively correct quantization errors across multiple experts.
> * This shows that the shared-B idea naturally extends to MoE architectures.
>
> 6. Calibration and memory overhead, CUDA kernel fairness, and GlowQ-S scoring metric
>
> Reviewers HaNH (W1, W3, W4,Q1) raised several practical questions:
>
> * What are the time and memory overheads of the calibration stage?
> * Are the latency comparisons fair given the custom CUDA W4A16 kernel?
> * Does GlowQ-S depend too heavily on a particular saliency metric?
>
> The revision addresses these points as follows.
>
> (1) Calibration and statistics overhead profiling (new Appendix D.1.2)
>
> * We profiled OPT-6.7B, OPT-13B, and OPT-30B, reporting GPU peak memory, CPU RAM usage, error/covariance tensor sizes, calibration time, and decomposition (SVD/RSVD) time in a table.
> * The results show that even for OPT-30B, calibration fits within a single 80GB GPU, and the higher CPU memory usage happens only during a one-time offline statistics collection stage.
>
> (2) CUDA W4A16 kernel and low-rank correction implementation (Section 4.5)
>
> * The custom CUDA W4A16 kernel handles only W4A16 GEMV in the small-batch decode path, combining dequantization of 4-bit weights and the dot product into a single kernel.
> * Importantly, the low-rank correction term +A(Bx) is never fused into this kernel and is always done as two separate PyTorch F.linear calls plus an add operation.
> * Layerwise, GlowQ, and GlowQ-S all use the same W4A16 kernel and the same approach for applying A and B. The only difference is whether $Bx$ is recomputed per layer or shared and cached across a group.
> * Therefore, the latency differences in Tables 3 and 8 reflect purely algorithmic differences (caching/sharing strategy), not kernel-level optimization. This clarification is added to Section 4.5.
>
> (3) GlowQ-S scoring metric and ablations (new Appendix G)
>
> * During the design of GlowQ-S, we tested several saliency metrics. Appendix G now gathers these and presents ablations over five metrics: GSVD energy capture (our g_ec), normalized error ratio (g_ner), Frobenius norm of error, cosine-similarity-based metric, and a simple layer-order baseline (Fig. 12).
> * While the best-performing metric varies somewhat across model families, g_ec and g_ner proved to be the most robust and effective overall. Accordingly, the final GlowQ-S policy evaluates both metrics and chooses whichever works better for each model.
> * We do not claim that any single metric is the best across all models. Instead, we present the ablation results to clarify the selection process.

---

> ### Author Response · Authors · 2025-12-01
> **Summary of Revisions 3**
>
> 7. Summary of concrete changes in the revision
>
> Connecting the above points to the paper structure, the main changes in the revision are:
>
> (1) New experiments and appendices
>
> * Appendix E: New experiments on Mistral-7B with INT2/3/4, MXFP4, MXFP6, NVFP4 (Quant only vs Quant + GlowQ).
> * Appendix F: LongBench 4K/8K results for Llama-3.1-8B-Instruct.
> * Appendix D.1.2: Calibration and statistics collection overhead (memory and time) for OPT-6.7B/13B/30B.
> * Appendix G: Ablation of saliency metrics for GlowQ-S (five metrics, Fig. 12).
>
> (2) Main text sections
>
> * Section 4.2: Updated from W4A16-only to include W4A4/W4A8 + GlowQ/GlowQ-S results, and comparisons with other error-correction baselines using the same setups.
> * Section 4.5: Clarified the relationship between GlowQ and ROSAQ, GuidedQuant, CommVQ, AnTKV; added the Qwen1.5-MoE-A2.7B experiments and shared-B memory savings; detailed the W4A16 kernel and low-rank application structure.
>
> (3) Theory and analysis
>
> * Section 3.1: Clearly defined quantization risk as a covariance-weighted low-rank problem and connected $(E_{cat} − AB) Σ_x^{1/2}$ to weighted low-rank literature.
> * Figure 2: Added a log–log eigenvalue spectrum panel for the input covariance, including power-law decay and estimated exponents α.
> * Appendix B: Clarified the justification for shared-B and subspace alignment using pre-/post-whitening cross-basis heatmaps.
>
> (4) Clarity and formatting
>
> * Unified notation for $E$ and $E_{cat}$.
> * Updated the color palette in Figure 1 for better clarity.
> * Added GlowQ-S restoration ratios (about 50%) to the caption of Table 2.
> * Renamed “No Caching” in Table 4 to “Layerwise” to reflect its accurate meaning.
> * Fixed typos (e.g., BitsAndBytes) and missing equation numbers, and generally revised the manuscript text for clarity and readability.
>
>
>
> We hope these revisions address the main concerns raised by the reviewers and that this summary is helpful for your meta-review.
>
> Sincerely,
> The authors

---

### Meta-Review · Area_Chair_Uesf · 2026-01-12

**Summary:**

The paper introduces GlowQ an approach for error correction in quantised models. There is a single shared right-factor matrix across modules. The projection is cached for efficiency.

Reviewer CZae, Reviewer HaNH and Reviewer DXnt raised concerns whether the proposed method is only effective in INT4 + W4A16 setting which compromises the majority of the experiments. This was the main issue in my opinion. The authors comprehensively addressed this concern by adding experiments for new INT precisions and low-bit FP formats, they also added experiments on the W4A4 and W4A8 settings. These results show that the proposed method is still effective. In any case, I think the authors should address the limitation of their experimental analysis and be explicit about the settings where their method has been convincingly shown to be most effective. Especially because the W4A16 setting is still useful in practice, e.g. if we mostly care about data movement and not necessarily reducing flops. However, it's worth pointing out that here other baselines may also perform equally well or better. Still, since this approach is reasonably orthogonal to other efforts in the field, I believe GlowQ is a useful contribution. Relatedly, regarding concerns about how the method is situated in the literature and related work, the authors added more discussion which clarifies that GlowQ is orthogonal and complementary. This should be empirically validated in future work.

The authors sufficiently addressed the concern from Reviewer CZae that error may accumulate for long-context tasks by including results from the LongBench suite.

The reviewers were also wondering about the "optimality" of the single shared $B$. The authors provided evidence that, empirically, covariance alignment is what makes the sharing possible without too much degradation in performance. Moreover, the gap between the Layerwise baseline and GlowQ seems to be small indicating the shared $B$ is good enough in practice.

In response to Reviewer CZae the authors added some results on MoEs which show that GlowQ still provides some benefits.

Questions regarding fair comparison (tuned CUDA kernels) and practical aspects (memory overhead, latency) were sufficiently addressed by the authors.

Overall, I believe the authors thoroughly and adequately addressed the reviewers' concerns. In this instance, the peer review process significantly improved the paper. Coupled with the strengths highlighted by the reviewers (technically sound idea and paper, potential practical significance, scalable and deployment-friendly design) I recommend acceptance.

**Reviewer Concerns:**

Almost all concerns by the reviewers were adequately addressed in a very comprehensive rebuttal by the authors.

**Reviewer Scores:**

There is a good chance that Reviewer HaNH and Reviewer DXnt would have increased their score form 4 to 6. Reviewer CZae would have likely kept their positive score.

---

### Decision · Program_Chairs · 2026-01-26

Accept (Poster)